# ON MEMORIZATION OF LARGE LANGUAGE MODELS IN LOGICAL REASONING

## ABSTRACT

Large language models (LLMs) achieve good performance on challenging reasoning benchmarks, yet could also make basic reasoning mistakes. This contrasting behavior is puzzling when it comes to understanding the mechanisms behind LLMs' reasoning capabilities. One hypothesis is that the increasingly high and nearly saturated performance on common reasoning benchmarks could be due to the memorization of similar problems. In this paper, we systematically investigate this hypothesis with a quantitative measurement of memorization in reasoning tasks, using a dynamically generated logical reasoning benchmark based on Knights and Knaves (K&K) puzzles. We found that LLMs could interpolate the training puzzles (achieving near-perfect accuracy) after fine-tuning, yet fail when those puzzles are slightly perturbed, suggesting that the models heavily rely on memorization to solve those training puzzles. On the other hand, we show that while fine-tuning leads to heavy memorization, it also consistently improves generalization performance. In-depth analyses with perturbation tests, cross difficulty-level transferability, probing model internals, and fine-tuning with wrong answers suggest that the LLMs learn to reason on K&K puzzles despite training data memorization. This phenomenon indicates that LLMs exhibit a complex interplay between memorization and genuine reasoning abilities. Finally, our analysis based on a per-sample memorization score sheds light on how LLMs switch between reasoning and memorization when solving logical puzzles.

## 1 INTRODUCTION

Modern Large Language Models (LLMs) show impressive reasoning capabilities that allow them to solve a wide range of challenging problems including commonsense reasoning and mathematical reasoning. In the meantime, LLMs also make mistakes on some of the most basic problems, such as comparing which number is bigger—13.11 or 13.8 (Lin, 2024), and counting the number of sisters that Alice's brother has (Nezhurina et al., 2024). This contrast is puzzling when it comes to understanding how exactly LLMs solve reasoning tasks. This question is important both scientifically and practically: understanding how LLMs reason could shed light on their learning and generalization behaviors. It is also crucial for real-world applications where robust reasoning is required due to safety and trustworthiness concerns (Wang et al., 2023a; Wallace et al., 2024; Lee et al., 2024; Wei et al., 2024a).

One hypothesis is that LLMs could be relying on *memorization* when solving those reasoning tasks, especially when measured by popular benchmarks that could be accidentally leaked into various massive internet-crawled pre-training datasets. Previous work (Tirumala et al., 2022; Carlini et al., 2023) show that LLMs could indeed memorize the training data, which may lead to potential privacy (Carlini et al., 2021) or copyright (Karamolegkou et al., 2023; Wei et al., 2024b) concerns. Additional evidences of potential memorization come from extensive studies on data contamination in LLMs (Magar & Schwartz, 2022; Balloccu et al., 2024; Shi et al., 2024; Xu et al., 2024; Oren et al., 2024). To mitigate the issue of benchmark saturation potentially due to memorization, some papers focus on designing dynamic benchmarks (Roberts et al., 2023; Zhu et al., 2024; Srivastava et al., 2024; Jain et al., 2024; Wu et al., 2024a) or alternative evaluation protocols (Zeng et al., 2023; Zhang et al., 2024; Xu et al., 2024; Srivastava et al., 2024).

In this paper, we take a direct approach to quantify the memorization behaviors of LLMs in reasoning tasks within a controlled setting. Specifically, we seek to understand: (i) whether LLMs rely on

Figure 1: Illustration of the definition of Local Inconsistency based Memorization Score, LiMem. High level of memorization occurs when the model shows high accuracy in solving sense problems but fails to consistently solve those problems under local perturbations that require similar underlying reasoning principles.

memorization to solve reasoning tasks, and (ii) whether memorization is only detrimental to learning to reason. Both questions are inspired by human behavior. For instance, when a student works hard on the preparation material for an exam, the preparation could help them get familiarized with the problems, and their ability to solve new problems could usually improve with enough exercises. However, without genuinely understanding the principles, they might fail when the same problem is slightly changed despite doing well on prepared problems. Our metric of memorization LiMem, illustrated in Fig. 1, is based on this intuition, that an LLM shows a high level of memorization when it solves reasoning problems with high accuracy but struggles to consistently solve those problems under local perturbations requiring similar mathematical principles (i.e., low consistency). We note that a similar perturbation (mostly at language-level) idea has been used in previous work, especially in detecting contamination (Golchin & Surdeanu, 2023; Yang et al., 2023; Xu et al., 2024). However, given our focus on understanding memorization in logical reasoning tasks, we further consider problem-level perturbation that slightly changes the mathematical structure of a puzzle, in addition to language-level perturbations. To facilitate our study, we propose a new logical reasoning benchmark that supports automatic problem-level perturbation. With this tool, we evaluate 11 off-the-shelf models, and fine-tuned Llama3-8B and GPT4o-mini to quantify memorization in reasoning tasks, and reveal interesting interplay between memorization and reasoning: while models indeed tend to memorize many logical puzzles, they develop genuine reasoning capabilities during fine-tuning (even directly on question-answer pairs without reasoning steps), and the reasoning performance improves as the memorization level increases.

In the following, we provide an outline of the paper and summarize our key contributions:

- To quantify memorization in reasoning tasks, we define a memorization score based on the notions of performance inconsistency under local perturbation, inspired by human behavior (§ 2.1).
- To facilitate the measurement, we propose a new logical reasoning benchmark based on the *Knights and Knaves* (K&K, Smullyan, 1978; Johnson-Laird & Byrne, 1990) puzzles, that can generate new puzzles at different difficulty levels, locally perturb existing puzzles, and automatically synthesize detailed reasoning steps to solve a given puzzle (§ 2.2).
- We show that K&K puzzles are challenging, and only the most advanced LLMs could solve them well. Moreover, our analysis suggests those models exhibit some level of memorization (§ 3).
- By fine-tuning on K&K samples, we confirm that modern LLMs are capable of memorizing a large collection of puzzles, and reach high memorization score when interpolating (i.e., fitting, Belkin et al., 2018) the training set. We observe that the models' generalization accuracies continue to improve as memorization grows (§ 4).
- We design various in-depth analyses (§ 4.1∼§ 4.3) to verify that the models developed genuine reasoning capabilities after fine-tuning even with only question-answer pairs and wrong answers, via local perturbation tests, cross difficulty-level transferability, and model internal probing.
- We show that fine-tuning with detailed reasoning steps can further boost the generalization on K&K puzzles, even when fine-tuned with wrong reasoning steps (§ 5).
- To analyze the interplay between memorization and reasoning, we measure per-sample memorization and study how LLMs switch between memorization and reasoning to solve a puzzle (§ 6).

## 2 HOW TO MEASURE MEMORIZATION IN REASONING TASKS

### 2.1 MEMORIZATION METRICS FOR REASONING TASKS

Memorization of LLMs has been studied in various contexts such as privacy (Carlini et al., 2023), copyright (Carlini et al., 2021; Karamolegkou et al., 2023; Wei et al., 2024b; He et al., 2024), and

Figure 2: K&K data generation framework employs abstract and natural language modules to generate question answer pair and synthetic CoTs for each K&K sample, based on the problem specification: number of persons ($N$), tree width ($W$), and depth ($D$). Perturbers in these modules can alter the math structure and language description, respectively.

solving knowledge intensive tasks (Hartmann et al., 2023). In this paper, we are specifically interested in measuring the level of memorization when solving reasoning tasks, by borrowing intuition from human behavior. For example, when preparing for an exam, a student may not be able to fully digest the underlying principles due to various reasons or constraints. But when (luckily) facing the same problem the student had prepared for, they would still be able to solve it. A key characteristic of this type of memorization is: (A) high accuracy on observed problems and (B) low accuracy when the problem is slightly changed. Based on this intuition, for a dataset $\mathcal{D}$ of reasoning puzzles, we combine the following two quantities to measure memorization:

1. For (A), we measure the accuracy of a target model $f$ on $\mathcal{D}$, denoted as $\mathsf{Acc}(f;\mathcal{D})$. We are especially interested in measuring on the set of *observed puzzles*, i.e., the training set, $\mathsf{Acc}(f;\mathsf{Tr})$. We say $f$ *interpolates* (Belkin et al., 2018; Muthukumar et al., 2020; Belkin, 2021; Bartlett et al., 2021) the training puzzles if $\mathsf{Acc}(f;\mathsf{Tr}) \approx 100\%$.

2. For (B), we measure a *consistency ratio* $\mathsf{CR}(f;\mathcal{D})$ between the number of *consistently solved puzzles* after some *local perturbations*, and the number of solved puzzles (without perturbation). We are interested in local perturbations that make minimal changes to the puzzle and maintain the same underlying principle for solving it, and a similar difficulty level (to be specified in § 2.2).

We combine the two factors to define a *Local Inconsistency-based Memorization Score*:

$$\mathsf{LiMem}(f;\mathcal{D}) = \mathsf{Acc}(f;\mathcal{D}) \cdot (1 - \mathsf{CR}(f;\mathcal{D})). \tag{1}$$

When there is no ambiguity, we simply call it the memorization score. $\mathsf{LiMem}(f;\mathcal{D}) \in [0,1]$ and a larger score provides stronger evidence of memorization. Specifically, a high $\mathsf{LiMem}(f;\mathsf{Tr})$ matches the characteristic behavior of human memorizing observed puzzles, and in this case we say $f$ *memorized* the training puzzles. Furthermore, we also measure $\mathsf{LiMem}(f;\mathsf{Tst})$ on test examples, to study if the generalization accuracy is due to reasoning or memorization. Note the $\mathsf{Acc}(f;\mathcal{D})$ factor is simple but necessary, as there are three types of behaviors: (i) solving by memorization, (ii) solving by reasoning, (iii) not solving (e.g., random guessing). A high $\mathsf{LiMem}(f;\mathcal{D})$ indicates (i), but a low $\mathsf{LiMem}(f;\mathcal{D})$ would only indicate (ii) if we separately check that $\mathsf{Acc}(f;\mathcal{D})$ is high.

To effectively measure the memorization score $\mathsf{LiMem}(f;\mathcal{D})$, we need a principled way to (1) perform a local perturbation that changes the puzzle while maintaining its difficulty level; (2) compute the new correct answer after perturbation. Towards this goal, we design and implement a functional dataset based on the Knights and Knaves puzzles (Smullyan, 1978; Johnson-Laird & Byrne, 1990).

## 2.2 Knights and Knaves Logical Reasoning Benchmark

*Knights and Knaves (K&K)* is a type of logical puzzle where some characters can only answer questions truthfully, and others only falsely. The goal is to infer each character $i$'s truthfulness $B_i$ (Boolean value) according to its statement $S_i$. Fig. 2 shows an example puzzle and its answer.

The principle underlying K&K is the Boolean satisfiability problem (SAT) (Boolean satisfiability problem, 2024). SAT was the first problem proven to be NP-complete and many well-known problems can be translated into SAT, such as hardware and software verification and theorem proving (SAT solver, 2024). Hence, the performance of a model on SAT (i.e., K&K puzzles) can be important indicative of its reasoning capabilities. Specifically, consider a K&K puzzle involving $N$ people, a possible solution assign a Boolean value to $N$ variables $B_1, B_2, \ldots, B_N$, where the truth

value of $B_i$ indicates whether the $i$th person is telling the truth. By definition, the $i$th person is telling the truth if and only if their statement $S_i$ is true. Therefore, a valid solution to a K&K puzzle is a Boolean assignment for $B_1, B_2, \ldots, B_N$ such that the following formula is true.

$$(B_1 \Leftrightarrow S_1) \wedge (B_2 \Leftrightarrow S_2) \wedge \cdots \wedge (B_N \Leftrightarrow S_N). \tag{2}$$

Based on the K&K puzzle, we design a *dynamic* benchmark that supports generating new puzzles and perturbing existing puzzles. Our library automatically solves the K&K puzzles and generates detailed reasoning steps and solutions for evaluation and training. Moreover, to support measuring memorization, we also provide a procedure to perturb a given puzzle and recompute the new solution after the perturbation. Specifically, our benchmark consists of two modules:

**The Abstract Module** has four components that can generate and manipulate K&K puzzles in an abstract form (see § C.1). (1) The **Generator** creates a random K&K puzzle given a specification $(N, D, W)$ that determines the difficulty level. Specifically, it generates a puzzle with $N$ people, and for each person, a statement that consists of a random tree of maximum width $W$ and depth $D$, where each node is sampled uniformly at random from the following candidate sets. The *leaf* node can be a claim that a specific person is lying (i.e., knave) or truth-telling (i.e., knight), and the *branching* node can be *and*, *or*, *not*, *imply*, and *equivalence*. (2) The **Solver** finds the solution to a given puzzle. In our study, we ignore puzzles with no or more than one solution. So we implement the solver by converting the puzzle to a Boolean satisfiability problem and enumerate all Boolean assignments so that we can easily obtain a list of all valid solutions to filter out unwanted puzzles. (3) The **Reasoner** generates a step-by-step reasoning procedure that leads to the solution. We design it to mimic the reasoning steps used by humans and some LLMs: instead of enumerating all Boolean assignments, it examines each person sequentially, makes an assumption (knight/knave) and checks if it leads to a contradiction with statements from people with specific identities assumed. If not, it continues to examine the next person; otherwise it will try an alternative assumption or backtrack to a previously examined person (details in § C.3). (4) The **Perturber**, given a puzzle, generates a locally perturbed version that is (superficially) similar to the original puzzle, and solvable with the same underlying principle and at a similar difficulty level. The Perturber replaces either an entire statement or a leaf node in a statement with a newly sampled one. The process is rerun until the perturbed puzzle has a unique solution different from the original puzzle, or until a maximum number of attempts is reached. This rarely happens for $N \geq 3$ people puzzles. When it happens we skip the puzzle in our perturbation analysis. See Tab. 1 for concrete examples.

**The Natural Language Module** has three components that operate in natural language space. (1) The **NL-Generator** takes an abstract K&K puzzle and formats it in natural language. It is template-based, with randomly sampled person names and a random template for making claims, and it uses a few heuristics to convert tree-structured logical statements to natural language. (2) The **NL-Reasoner** converts the reasoning steps computed by the abstract Reasoner to the natural language format in a similar manner. See Fig. 15 for dataset length distributions. (3) The **NL-Perturber** generates perturbed puzzles by keeping the abstract puzzle intact and manipulating the language-level descriptions as follows (See § C.7): (i) replace character names with uncommon names; (ii) replace knight/knave with other similar pairs of role names, e.g., saint/sinner; (iii) reorder the statements from the characters; (iv) flip the role name from knight/knave to knave/knight. Note that the flipped role perturbation is somewhat adversarial as it goes against the common intuition that a good character would tell the truth and a bad one would lie, so we include it mostly for reference purposes.

## 3 QUANTIFYING LLM MEMORIZATION IN REASONING TASKS

We measure memorization for off-the-shelf models (§ 3.1) and fine-tuned models (§ 3.2).

### 3.1 OFF-THE-SHELF MODELS

**Evaluation setup.** To generate our K&K benchmark (§ 2.2), we use the max tree width $W = 2$ and depth $D = 2$, and create 100 test puzzles for each $N$-people task ($N \in \{2, 3, \ldots, 8\}$). Then, we generate perturbed versions for each puzzle under 6 perturbation types introduced in § 2.2: {*perturbed statement, perturbed leaf node, random role-pair name, uncommon person name, reordered statement, flipped role*}. We utilize 0-shot direct prompting with task-specific instructions for open-ended question-answering (details in § D.2). Note that even under direct prompting, capable LLMs can generate Chain of Thought (Wei et al., 2022, CoT). Our evaluation mainly considers the 0-shot setting, excluding potential biases introduced by in-context examples (Zhao et al., 2021). We defer

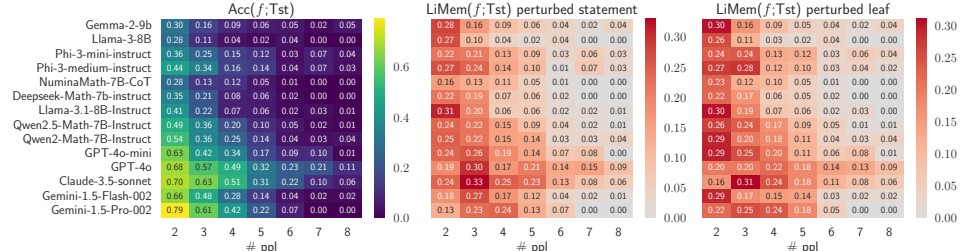

Figure 3: Test accuracy $\mathsf{Acc}(f; \mathsf{Tst})$ of off-the-shelf models under 0-shot direct prompting drops with increasing puzzle complexity (left). $\mathsf{LiMem}(f; \mathsf{Tst})$ on test examples under statement perturbation (middle) and leaf perturbation (right) is large for specific models, indicating signs of memorization in solving these puzzles.

the results under CoT prompting (i.e., explicitly adding a CoT trigger "Let's think step by step"), and 1-shot prompting to § E. To evaluate the accuracy of the model's output, we use keyword matching for full-puzzle correctness. A response is considered correct if every person's true identity is included in the conclusion part of the model's response.

**Off-the-shelf models do not perform well on K&K tasks.** We evaluate 14 models that are shown to perform competitively on common reasoning benchmarks[1]. As shown in Fig. 3, K&K benchmark poses a challenging logical reasoning task for all the models. Even for the easiest puzzles involving only 2 persons, the best models still achieve $\leq 70\%$ accuracy. And the performance drops significantly as the complexity increases (the best accuracy is only $11\%$ for 8-people puzzles).

**Off-the-shelf models are sensitive to locally perturbed test samples.** To quantify LLMs' memorization of the logical reasoning task, we employ the score proposed in Eq. (1). Since the training data for the off-the-shelf models is unknown, we focus on measuring the memorization score $\mathsf{LiMem}(f; \mathsf{Tst})$ on the test set here. We observe (Fig. 3 right) that for the cases where a model has relatively high accuracy, the memorization scores under local perturbation are generally high. For example, $\mathsf{LiMem}(\text{Claude-3.5-Sonnet}; \mathsf{Tst}) > 0.3$ on 3-people puzzles under both statement and leaf perturbation. Those measurements indicate signs of memorization when solving these puzzles.

## 3.2 FINE-TUNED MODELS

Here, we study a model's memorization behavior when fine-tuned on K&K puzzles.

**Fine-tuning setup.** We take Llama3-8B and GPT4o-mini and run *supervised fine-tuning* (SFT) on a set of K&K training puzzles disjoint from the test set. We consider two fine-tuning paradigms: (1) Fine-tuning on detailed CoT steps (**CoT FT**): during SFT, the model observes the concatenation of the question, synthetic CoT steps, and the answer for each puzzle; the loss is computed on the CoT steps and the answer part. (2) Fine-tuning on the answers (**Direct FT**) where the model observes the question-answer pair for each puzzle, and the loss is only computed on the answer part. Examples of CoT FT/Direct FT training instances are provided in § D.2.2. We fine-tune the models for each $N$-people task separately, with $n_{\text{train}} = 1,000$ for $3 \leq N \leq 8$, and $n_{\text{train}} = 200$ for 2-people task due to limited number of combinations. We fine-tune Llama3-8B for 50 epochs[2] and GPT4o-mini for 5 epochs (due to budget constraints) via the OpenAI fine-tune API (details in § D.2). During the evaluation, we follow the same prompting paradigm as FT paradigm, i.e., direct/CoT prompting for direct/CoT-FTed model, which is shown effective in § E.2.

**LLMs interpolate K&K training puzzles**. In Fig. 4, we present the training accuracy of models trained on each task on the $x$-axis (each dot represents a training epoch). We find that models exhibit high training accuracy in tasks such as $3/5$-people puzzles. The higher capacity GPT4o-mini nearly achieves interpolation ($\mathsf{Acc}(f; \mathsf{Tr}) \approx 100\%$) using both Direct FT and CoT FT.

**Interpolating LLMs have large memorization scores on training examples**. From Fig. 5, **(1)** we observe high $\mathsf{LiMem}(f; \mathsf{Tr})$ memorization score on training samples (e.g., $\sim 50\%$ on 8-people task) under various perturbations. It shows significant gaps between accuracy on the original sample and the consistent accuracy under perturbation, suggesting a heavy reliance on memorization. **(2)** $\mathsf{LiMem}(f; \mathsf{Tr})$ is higher for more difficult tasks (e.g., $5/8$-people), which could mirror human behavior, where memorization is often used to tackle challenging tasks that people do not fully understand. **(3)** More capable model GPT4o-mini, in general, show lower memorization scores than Llama3-8B.

---

[1]We are not evaluating the OpenAI o1 model because API access is limited to only the highest-tier users.

[2]We fine-tune Llama3-8B for max 100 epochs in Fig. 21 and find that it typically converges at 50 epochs.

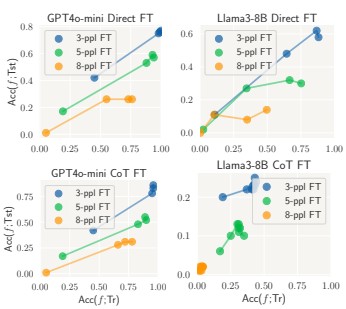

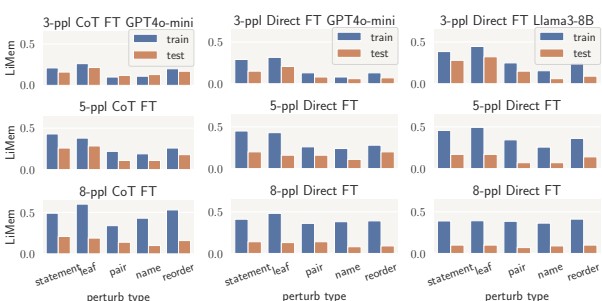

Figure 4: Train & test accuracy increases over the epochs. FTed LLMs can achieve interpolation ($\approx$ 100% train accuracy) for easy tasks, e.g., 3/5-people puzzles. Llama3-8B struggles with CoT FT, likely due to its limited capacity.

Figure 5: Fine-tuned LLMs exhibit high memorization score on the training set under different perturbations, especially for hard tasks. The score on the test set can be smaller than on the training set. Models show stronger memorization under math-level perturbations compared to language-level perturbations. We report consistency ratio separately in Fig. 19, and memorization score of combined math & language-level perturbations in Fig. 20.

**Ablation on local perturbations.** Comparing different perturbations in Fig. 5, we find that **(1)** LLMs exhibit a higher memorization score when evaluated with math-level perturbations (e.g., statement/leaf) compared to language-level, which indicates that LLMs can compose the language understanding capability to solve the same puzzle in alternative phrasing. **(2)** LLMs get nearly zero accuracy on role-flipped samples (e.g., when a knight, typically viewed as truthful, is defined as always lying), and memorization score $\mathsf{LiMem}(f; \mathsf{Tr})$ under role-flipping for Llama3-8B is $\sim 80\%$ as shown in Fig. 6. This could be due to an internal bias or commonsense understanding that knights are inherently good characters (e.g., truthful), and thus LLMs disregard the altered puzzle statement.

## 4 LLMs Learn to Reason by Fine-tuning With Answers Only

§ 3 shows that both off-the-shelf and fine-tuned models exhibit memorization when solving K&K reasoning tasks. Does it mean that those models do not have reasoning capabilities at all? As we will show, it turns out that the models can do both, and interestingly the reasoning capability consistently improves as the memorization level increases when the models are fine-tuned on K&K puzzles.

We focus on analyzing Direct FT in this section and discuss CoT FT in § 5. For humans, solving K&K tasks without understanding the underlying logic is difficult. However, after observing the step-by-step reasoning steps, people can understand the procedure and solve the puzzles more easily. Similarly, compared to CoT FT, learning from only answers (Direct FT) without detailed reasoning steps is intuitively more challenging for LLMs, as the models need to come up with the reasoning procedures on their own. Therefore, the models might be more likely to rely on memorization in this case. Surprisingly, from Fig. 5, we did not observe Direct FTed GPT4o-mini models exhibiting consistently higher memorization score than CoT FTed ones. It turns out that models can learn to reason K&K puzzles well directly from observing only question-answer pairs, as we will show in § 4.1. To better understand what the model actually learns through Direct FT, we conduct a probing analysis on model internals in § 4.3 and an ablation study with incorrect answers fine-tuning in § 4.3.

### 4.1 Reasoning capabilities of Direct FT-ed model

**Generalization performance increases with memorization level**. As shown in Fig. 6 , the test accuracy ($y$-axis) of fine-tuned LLMs on the unseen test set continues to increase over the epochs, despite that the memorization score $\mathsf{LiMem}(f; \mathsf{Tr})$ on training samples also increases.

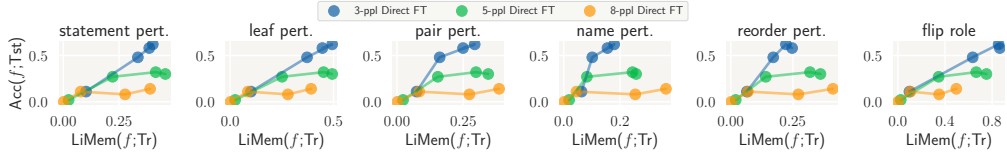

Figure 6: Test accuracy of fine-tuned Llama3-8B increases over epochs, despite the stronger memorization of training data as reflected by larger $\mathsf{LiMem}(f; \mathsf{Tr})$. See Fig. 18 for results on GPT4o-mini.

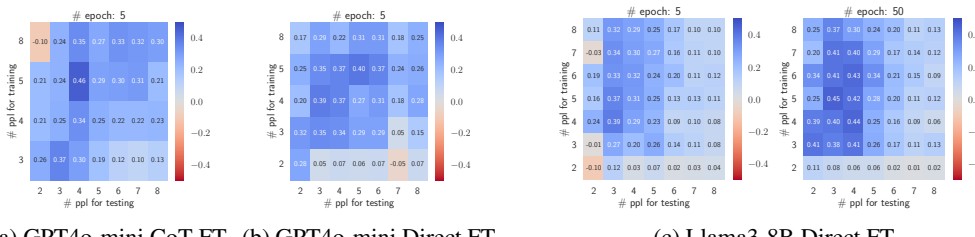

(a) GPT4o-mini CoT FT    (b) GPT4o-mini Direct FT        (c) Llama3-8B Direct FT

Figure 7: Test accuracy improvement on $N$-people problems for LLMs fine-tuned on $M$-people problems, compared to the unfine-tuned model, under 0-shot direct prompting. Most grid values are above 0, indicating transferability and enhanced reasoning abilities across unseen tasks. Results for more epochs are in § E.2.

$\mathsf{LiMem}(f; \mathsf{Tst})$ **on test samples is smaller than** $\mathsf{LiMem}(f; \mathsf{Tr})$ **on train samples** in Fig. 5, indicating that models are less likely to rely on memorization when solving test samples compared to training samples. Since a low memorization score can result from either solving by reasoning or failing to solve (e.g., low accuracy), we further analyze the model's reasoning ability on test samples through a transferability study below.

**Fine-tuned model generalizes across different difficulty levels**. We evaluate LLMs' transferability by fine-tuning on $M$-people puzzles and testing on $N$-people puzzles. When $M \neq N$, the testing is out-of-distribution compared to training and solving it requires reasoning. The $N \times M$ test accuracy improvement grid (compared to the un-FTed model) in Fig. 7 shows: **(1)** Training on any $M$-people puzzle generally improves test accuracy on any $N$-people puzzles, suggesting that the model learns general task-solving rules after FT (to reason and solve both easier and harder unseen puzzles). **(2)** More training epochs (e.g., 50 vs. 5) improve results, especially for Llama3-8B. **(3)** Accuracy gains are larger for $N \leq 6$ puzzles, though improvements on harder tasks remain possible.

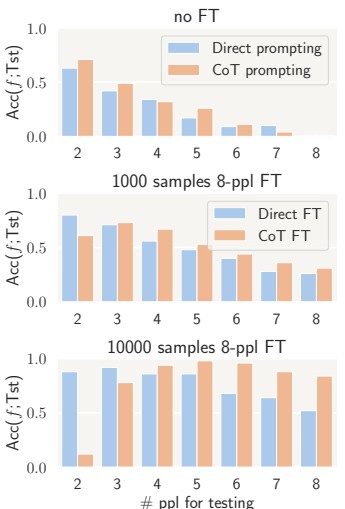

Figure 8: Transferability of $1k/10k$ 8-ppl FTed GPT4o-mini. Llama3-8B results are in Fig. 24.

**Fine-tuning with 10k 8-people puzzles.** Given the significant performance improvement from fine-tuning, a natural question arises: can brute-force fine-tuning on a very large number of puzzles eventually solve the K&K puzzles, by observing/memorizing a variety of combinations of persons' claims and their corresponding answers? We Direct FT GPT4o-mini on $1k/10k$ of the most challenging 8-people puzzles for 5 epochs. Fig. 8 shows that **(1)** $10k$-FT significantly outperforms $1k$-FT across all tasks, reaching $\sim 90\%$ test accuracy on moderately difficult 4/5-people puzzles. **(2)** CoT FT is generally more effective than Direct FT with $10k$ samples, likely due to the guidance provided by reasoning steps. **(3)** An exception is the 2-people task, where the training and testing distribution gap causes the CoT FTed model to occasionally get stuck in a loop of listing assumptions and contradictions, resulting in long, repetitive responses without reaching a conclusion[3]. **(4)** Direct FT with $10k$ puzzles achieves surprisingly high test accuracy on all tasks, e.g., 52% on 8-people tasks, where the un-FTed model scores near 0. Notably, the models do not see reasoning steps during training and rely solely on memorizing answers. We also observe high transferability for $10k$ Direct FTed Llama3-8B in Fig. 24, e.g., 87% test accuracy on 3-people puzzles.

## 4.2 PROBING DIRECT FTED MODELS

To investigate whether Direct FTed models develop internal understanding of the skills necessary to solve K&K puzzles when learning only from the answers, we use probing techniques (Adi et al., 2017; Conneau et al., 2018; Hewitt & Liang, 2019; Ye et al., 2024) to analyze their internal representations. Specifically, we study whether a Direct FTed model's intermediate outputs provide evidence that it can distinguish between correct and incorrect statements for a given K&K puzzle, which is essential for solving the puzzle via reasoning. For a given model, we extract intermediate outputs from all transformer blocks for 200 correct and 200 incorrect statements, then check whether these

---

[3]We observe similar accuracy drop on 2-people task for Llama3-8B (see Fig. 24) when it is Direct FTed for overly long epochs. We provide more examples and discussions in § E.2.2.

outputs form distinct clusters by measuring the training accuracy of a logistic regression model fit on them (see § D.2.3 for details). For each $N$-people K&K puzzle, we report the per-layer probing accuracy averaged across seven Direct FTed models, each Direct FTed on an $M \in \{2, 3, \ldots, 8\}$-people task.

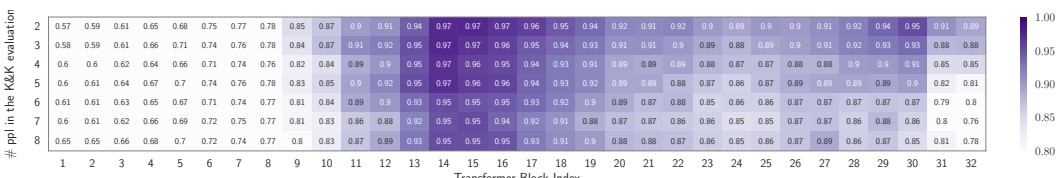

Figure 9: Probing accuracy of K&K puzzles with different number of people in testing puzzles across different layers of the Llama3-8B transformer model. Results for un-FTed models are shown in Fig. 33 in § E.

Fig. 9 shows **(1)** a clear trend of higher probing accuracy in deeper layers, peaking at around the 14th/15th layer. The near-perfect peak accuracy suggests that the model's internal representations have a clear distinction between true/false statements about a given puzzle. **(2)** The probing accuracy is much higher than the un-FTed model (Fig. 33 in § E), suggesting that such representations are learned from the question-answer pairs during Direct FT. **(3)** Puzzles with more people seem to demand more internal computation, as evidenced by the point where probing accuracy surpasses 85% shifting to later transformer blocks.

### 4.3 DIRECT FT WITH WRONG ANSWERS

To further explore what could the models learn from the question-answer pairs without detailed reasoning steps, we consider an extreme scenario of learning with incorrect answers: for each $N$-people training puzzle, we randomly select $\tilde{N}$ from $[1, N]$ and flip the knight/knave identities of $\tilde{N}$ randomly chosen individuals. Surprisingly, Fig. 10 shows that Direct FT with incorrect answers still leads to non-trivial improvements for Llama3-8B. These improvements occur gradually over more epochs, suggesting that the model progressively developed reasoning skills during fine-tuning.

Note that in this case the improved test accuracy could not have come from memorization because 100% of the training examples are incorrectly labeled. However, since in each wrong answer of a $N$-people puzzle, there are still $N - \tilde{N}$ correct role assignments where the random $\tilde{N} \geq 1$. The model might have learned to reason from those partially correct role assignments in the wrong answer.

However, as shown in Fig. 11, when applied to more capable GPT4o-mini models, Direct FT on 5-people puzzles where 100% training examples have corrupted answers does not lead to improvement. Moreover, the negative effects transfer to other tasks, notably easier ones (2/3/4-people). Nevertheless, as the percentage of corrupt-answer training examples reduces ($\leq 50\%$), the model could gain improved reasoning capabilities that generalize across different $N$-people tasks. We provide GPT4o-mini results under more epochs in Fig. 32 and Llama3-8B results for partially wrong answer FT in Figs. 30 and 31.

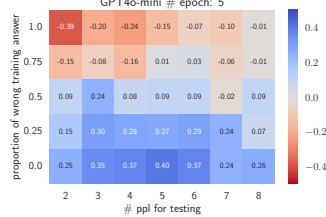

Figure 11: Direct FT w/ various wrong training answer proportions on 5-ppl task.

### 5 LLMs LEARN TO REASON BY FINE-TUNING WITH CoTs

Here we measure models' reasoning capabilities after fine-tuning with detailed reasoning steps.

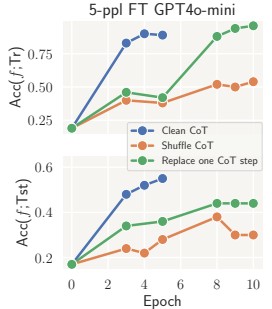

Figure 12: Wrong CoTs FT.

**Model learns to reason on CoT when model capacity is large enough**. As shown in Fig. 4, **(1)** training with reasoning steps as guidance improves test accuracy ($y$-axis) on unseen puzzles. **(2)** However, Llama3-8B struggles with CoT FT, likely due to its limited capacity to effectively learn CoT skills with $\leq$1K training samples. **(3)** Similar to Direct FT results in § 4, in CoT FT, memorization of training data is higher than test data (Fig. 5), yet test accuracy improves despite that the memorization score increases over training (Fig. 18), and the fine-tuned models show positive transferability to easier/harder tasks (Fig. 7). **(4)** Though models can generalize surprisingly well under

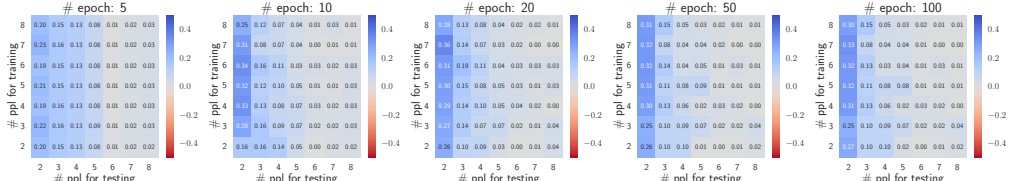

Figure 10: Test accuracy improves on $N$-people puzzles for Llama3-8B fine-tuned on $M$-people puzzles **with completely wrong answers**, compared to the unfine-tuned model. This evaluation uses 1-shot direct prompting (see Fig. 29 for results under different prompting setups).

Direct FT, CoT FT could lead to much higher test accuracy, especially with a larger training set (Fig. 8).

**Fine-tuning with wrong CoTs.** The CoT training data includes both reasoning steps and answers. To understand the role of the CoT component in improving model generalization, we fine-tune GPT4o-mini with two types of incorrect CoT data: **(a)** randomly shuffled CoT steps, disrupting the logic of the reasoning steps; and **(b)** CoTs with a single incorrect step, simulating genuine mistakes that people would sometimes make, where one step is randomly replaced with another puzzle's CoT step (adjusting names to fit the current context). The results in Fig. 12 show that **(1)** fine-tuning with a 100% corrupted CoT dataset can still enhance test accuracy over the epochs, suggesting that the model learns to reason (potentially from the correct answers) despite CoT errors. **(2)** Altering one CoT step slows convergence and reduces test accuracy compared to clean CoT. **(3)** Shuffling CoT steps further harms both convergence and generalization. These also suggest that using correct logical chains in CoT can help LLMs to more effectively learn to reason.

## 6    DISTINGUISHING MEMORIZATION FROM REASONING

The findings above show that models' reasoning capabilities continue to improve as they memorize more training examples. In other words, the models use both memorization and reasoning to solve the puzzles. How do models decide which example to memorize or reason about? We can use our framework to study this question by extending the memorization score to a per-example metric.

Specifically, consider measuring Eq. (1) on a 1-point dataset $\mathcal{D} = \{x\}$. We skip the examples where $\mathsf{Acc}(f; \{x\}) = 0$ as the consistency ratio $\mathsf{CR}(\{x\})$ is NaN in this case. Then $\mathsf{LiMem}(f; \{x\}) \in \{0, 1\}$ is a binary indicator: 0 indicates $x$ is consistently solved after local perturbation; 1 indicates otherwise. We would like to see if there is a clear rule that can separate the two types of puzzles.

**Setup.** We collect training samples $\{x_i\}$ on which the targeted LLM makes correct predictions, and assign a binary categorical label as either "consistently solved" (i.e., solved by reasoning) puzzle or "not consistently solved" (i.e., solved by memorization) puzzle. We split this dataset into random disjoint 80%/20% training/test sets, and train a simple logistic regression model to solve this binary classification problem, in order to study the question: Is there a simple indicator that determines whether a model would solve a given puzzle by reasoning or memorization?

**Puzzle-based indicators.** We consider the following features: (1) TF-IDF; (2) Bag-of-Words; (3) Word Length; (4) Character Length; (5) concatenation of all. Each feature can be extracted from one of the following fields: (1) question; (2) synthetic CoT reasoning steps; (3) model response[4]; (4) concatenation of the above fields. The training and test performance (measured with AUC as the dataset

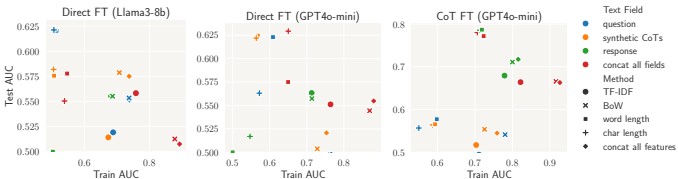

Figure 13: AUC for classifying 3-people puzzles under leaf perturbation based on puzzle-based indicators. Results under more tasks and perturbations are in Fig. 34.

can be unbalanced) are shown in Fig. 13. We observe a test AUC of 0.629/0.787 for Direct/CoT FT-ed GPT4o-mini, and 0.627 for Direct FT-ed Llama3-8B. This indicates that the puzzle-based indicators could be informative, though not perfect, at determining which examples are reasoned vs. memorized.

---

[4]Strictly speaking this is a model-based indicator feature.

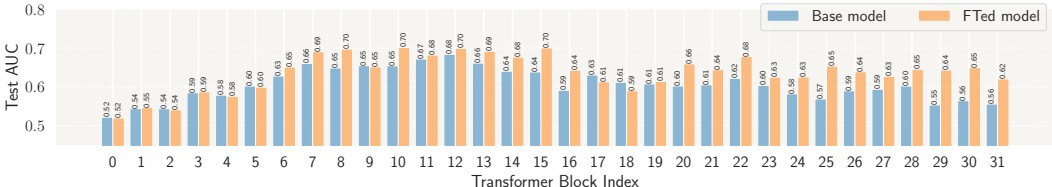

Figure 14: Test AUC for predicting 3-people puzzles based on whether they are consistently solved under leaf perturbation by the Llama3-8B model Direct-FTed. The embeddings across different layers of the fine-tuned Llama3-8B provide more distinguishable signals than those of the un-FTed model, leading to 0.7 AUC at the middle layers. Results under more tasks and perturbations are in Fig. 35.

**Model-based indicators.** Here we study model-based indicators to test whether the internal activations of the fine-tuned model are informative for this categorization. Since we do not have access to the model internals of GPT4o-mini, we conduct the experiment on Llama3-8B. Specifically, we feed each puzzle question to the FT-ed model, collect the average embedding at each layer as features, and then train a linear classifier based on the features of each layer. The test AUCs are shown in Fig. 14, where we compare the results based on a not-FTed base Llama3-8B model, to see how much of the feature expressiveness comes from the fine-tuning process. We observe that **(1)** the features from the bottom layers are too low level to classify memorization vs. reasoning, but the test AUCs improve at higher layers. **(2)** The features from the FTed model are consistently more informative than the un-FTed model, suggesting that the model's decision regarding memorization vs. reasoning on specific samples likely stems from the fine-tuning process. **(3)** The best model embedding-based indicator provides stronger signals than the puzzle-based indicator (Fig. 13 left) for Llama3-8B, with 0.70 and 0.627 test AUC on 3-people puzzles, respectively.

## 7    RELATED WORK

**Memorization in LLMs.** Previous work on LLM memorization primarily focused on near-verbatim training text regurgitation from the perspective of privacy or copyright concerns (Carlini et al., 2021; Lee et al., 2022; Carlini et al., 2023; Lukas et al., 2023; Biderman et al., 2024; Prashanth et al., 2024). In contrast, we focus on quantifying the memorization behavior of LLMs when solving reasoning tasks, using a metric computed with the help of local perturbation of reasoning puzzles.

**Benchmark Contamination and Logical Reasoning Evaluation.** Recent research has revealed LLMs' significant performance decline when faced with altered versions of popular reasoning benchmarks (Oren et al., 2024; Xu et al., 2024; Yang et al., 2023; Yao et al., 2024; Zhang et al., 2024; Srivastava et al., 2024), suggesting potential benchmark contamination. Various synthetic benchmarks have been developed to evaluate LLMs' logical reasoning capabilities, allowing for dynamic and scalable generation of samples with different configurations and difficulty levels (Clark et al., 2020; Giadikiaroglou et al., 2024; Parmar et al., 2024; Dziri et al., 2024; Lin et al., 2024; Kazemi et al., 2024; Mondorf & Plank, 2024). TruthQuest (Mondorf & Plank, 2024) is the most similar task to our work, which provides K&K-type of 3-6 person puzzles and answers. Our work provides more comprehensive dynamic set of K&K puzzles that support the automatic generation of perturbations, solutions, and detailed reasoning steps. Moreover, we define and measure memorization, and reveal its intricate relation to reasoning.

We refer the readers to § B for a more comprehensive discussion of related work.

## 8    CONCLUSION

We propose a memorization metric LiMem based on the inconsistency when solving a locally perturbed logical reasoning puzzle, and quantitatively characterize the amount of memorization and reasoning. Through an in-depth analysis based on local perturbation, transferability, intermediate outputs probing, and fine-tuning with wrong answers, we find that LLMs learn to reason as they memorize more training examples. Furthermore, we study input and model-based signals that determine which puzzles are solved by reasoning vs by memorization. To support these studies, we create a feature-rich dynamic logical reasoning benchmark that not only enables our memorization study, but could also be useful for future studies related to LLM logical reasoning. We defer more discussion and future work to § A.

## REPRODUCIBILITY STATEMENT

In this paper, we have taken steps to ensure the reproducibility of our results: (1) The source code for data generation, LLM evaluation and fine-tuning is provided in the supplementary material. We have provided the README and scripts to replicate the experiments in the paper. (2) Detailed information about K&K data generation can be found in § C. This includes descriptions of each component in the abstract puzzle module and the natural language module. (3) We described the experimental setups in each section (§ 3-§ 6). Additional details on models, hyperparameters, LLM evaluation, fine-tuning, and computational resources are provided in Appendix § D.

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

APPENDIX

## A    DISCUSSION AND FUTURE WORK

Our results reveal intricate phenomena of the interplay between reasoning and memorization, but challenging questions remain open: (i) While a model's reasoning capabilities improve during fine-tuning as it memorizes more training puzzles, it is unclear exactly how those capabilities develop, especially when fine-tuned on only question-answer pairs without detailed reasoning steps. (ii) While the models' reasoning capabilities can be significantly improved after fine-tuning, they have not reached 100% test accuracy yet. Is it because the models only learned some "shortcut rules" that can only solve a specific subset of puzzles? If so, what are the shortcuts? (iii) Since some model-based indicators can approximately predict when the model is solving a specific puzzle by memorization vs by reasoning, can we further design intervention mechanisms to bias the model towards reasoning during inference or training time? Exploring the open questions in further research would deepen our understanding of this space.

## B    EXTENDED RELATED WORK

**Memorization in LLMs.** Prior work has explored training data memorization in LLMs, primarily in the contexts of *privacy* and *copyright* concerns (Carlini et al., 2021; Lukas et al., 2023; He et al., 2024), focusing on how LLMs may reproduce text near-verbatim to their training data (Lee et al., 2022; Carlini et al., 2023; Biderman et al., 2024). Recent work by Prashanth et al. (2024) further introduces a taxonomy for memorization, categorizing it into Recitation, Reconstruction, and Recollection. They investigate the memorization behaviors of the Pythia model (Biderman et al., 2024) on the Pile dataset (Gao et al., 2020). In contrast, we examine memorization in the *reasoning* context, and focus on analyzing whether LLMs can accurately solve problems encountered during training but struggle to solve slightly perturbed variants. This allows us to better investigate the extent to which LLMs truly understand and generalize the *underlying principles* of the reasoning problems they have been trained on, as opposed to merely memorizing the *text*.

Recent research discusses signs of LLMs memorization in reasoning tasks by evaluating them on counterfactual reasoning tasks. These counterfactual tasks demand similar abstract reasoning skills as the original tasks but are less common in the training data. For instance, tasks such as reversing a sequence of words (McCoy et al., 2024) show better performance on high-probability sequences than on low-probability sequences; shifting each letter by $n$ places in the alphabet (Rot-$n$) (Prabhakar et al., 2024; McCoy et al., 2024) demonstrates higher performance when $n = 13$ than for other values, likely because "Rot-13" is commonly used in online forums. Wu et al. (2024b) presents 11 counterfactual tasks (e.g., 1-indexing in Python, base-9 arithmetic) that show significant performance declines. Jiang et al. (2024) changes some tokens in the reasoning task descriptions which leads to significant performance drops, suggesting that models might depend on recognizing superficial patterns with strong token bias. Moreover, Razeghi et al. (2022) finds a strong correlation between the accuracy for a number on numerical reasoning tasks and its frequency in pretraining for GPT-J/GPT-Neo. In our study, we formally define a memorization score to quantify performance variance under task perturbations, covering both counterfactual alterations (e.g., switching the roles of knights and knaves) and standard perturbations on language level and problem structure level.

**Detecting benchmark contamination.** Recent work has shown that LLMs' performance drastically declines when faced with altered versions of popular reasoning benchmarks, suggesting potential contamination/memorization of these benchmarks. The benchmark variants include diverse forms such as altered multiple-choice questions formats (Wang et al., 2024b; Zong et al., 2024; Gupta et al., 2024; Zhou et al., 2024; Robinson & Wingate, 2023), rephrased or translated problems (Xu et al., 2024; Yang et al., 2023; Yao et al., 2024), shuffled example orderings (Oren et al., 2024), human-curated problems of comparable difficulty (Zhang et al., 2024), functional variants generating random instantiations (Srivastava et al., 2024; Mirzadeh et al., 2024), and problems beyond specific date cutoffs (Roberts et al., 2023; Jain et al., 2024). Previous work either focus on surface level language perturbations or require extensive expert-level annotations for math level variations. In contrast, our benchmark support automatic problem-level perturbation, solution and reasoning procedure generation, and easily scale to different difficult levels and dataset sizes without extra human efforts.

**Logical reasoning benchmarks.** To evaluate logical reasoning capabilities in LLMs, synthetic benchmarks have been developed to enable scalable generation of samples with varying configurations and difficulty levels (Clark et al., 2020; Giadikiaroglou et al., 2024; Parmar et al., 2024). For

instance, DyVal (Zhu et al., 2024) uses directed acyclic graphs to dynamically generate samples on reasoning tasks including deductive, Boolean, and abductive reasoning. Chen et al. (2024) focus on propositional logical problems involving definite clauses, and synthetically generate variations with different premise orders, such as forward, backward, and shuffled. Dziri et al. (2024) explore the limitations of LLMs in tasks requiring compositional reasoning, including multiplication, logic grid puzzles, and dynamic programming problems. ZebraLogic (Lin et al., 2024) is an extended benchmark that systematically tests logical reasoning capabilities. BoardgameQA (Kazemi et al., 2024) presents a question-answering dataset characterized by contradictory facts and rules in the questions. PRONTOQA (Saparov & He, 2023) is a synthetic question-answering dataset where each example is generated from a synthetic world model represented in first-order logic. This dataset enables parsing the generated chain of thought into symbolic proofs, facilitating formal analysis. TruthQuest (Mondorf & Plank, 2024) is the most similar task to our work, which provides evaluation samples based on K&K-type of puzzles involving 3-6 person. Our work provides more comprehensive *dynamic* set of K&K puzzles that support automatic generation of perturbations, solutions and detailed reasoning steps. Moreover, based on this benchmark, we define and measure memorization in reasoning tasks, revealing intricate interplay between memorization and reasoning in LLMs.

**Improving reasoning via fine-tuning.** Prior work has explored fine-tuning LLMs on synthetic reasoning data to enhance their performance on reasoning. DyVal (Zhu et al., 2024) shows that fine-tuning Llama2-13B-chat on their synthetic reasoning benchmark improves its performance on other popular reasoning benchmarks. BoardgameQA (Kazemi et al., 2024) find that fine-tuning BERT-large and T5-XXL on their training dataset with synthetic proofs outperforms few-shot CoT prompting using PaLM. Ye et al. (2024) pretrain GPT2 from scratch on synthetic math problems, synthetic CoT steps and solutions and show that model can solve problems from the same distribution and generalize to out-of-distribution (OOD) problems. However, Dziri et al. (2024) show that while GPT-3 fine-tuned on their compositional reasoning tasks with/without reasoning steps can solve in-distribution (ID) problems, it fails to generalize to OOD tasks with increased problem sizes. Besides using synthetic CoTs, there are work using model-generated CoTs to enhance the models' reasoning capabilities (Chung et al., 2024). STaR (Zelikman et al., 2022) uses model self-generated CoTs on correctly solved samples to iteratively fine-tune itself as a self-taught reasoner. A number of work (Puerto et al., 2024; Kim et al., 2023; Ho et al., 2023; Hsieh et al., 2023) leverage CoTs generated from teacher models to train smaller student models. Additionally, some recent efforts have focused on leveraging intermediate reasoning steps in CoT more implicitly. For instance, Deng et al. (2023) distill intermediate reasoning tokens into the network layers by representing reasoning steps as vectors and using them as targets; Deng et al. (2024) distill CoT by gradually removing the intermediate steps and fine-tuning the model to internalize these steps, predicting the answers based on partial CoT. Both studies show that full CoT fine-tuning may not be necessary for the model to achieve strong reasoning performance.

In our study, we employ both direct fine-tuning and CoT fine-tuning to achieve memorization on K&K training data. Notably, our findings show that the fine-tuned GPT4o-mini and Llama3-8B models can effectively generalize to unseen OOD and ID K&K problems, contributing new insights to the topic of LLM fine-tuning for reasoning.

Orthogonal to our work, inference-time techniques have been explored to enhance reasoning performance such as self-consistency (Wang et al., 2023b), self-verification (Weng et al., 2023), and integration with external symbolic solvers (Pan et al., 2023).

**Grokking**. Our findings are related to Grokking, first identified by Power et al. (2022) on a small algorithmic dataset, where validation accuracy suddenly improves from random chance to near-perfect generalization long after severe overfitting. Follow-up studies expanded the range of tasks where grokking occurs and proposed various explanations (Liu et al., 2022a; Murty et al., 2023; Liu et al., 2022b). Recently, Wang et al. (2024a) observed grokking in the domain of complex knowledge-based tasks, showing that implicit reasoning over parametric knowledge emerges only after extensive overfitting. In this work, we observe a related phenomenon but through the lens of memorization. Through novel (math & language-level) perturbation tests and transferability analyses, we verify that LLM reasoning skills emerge alongside memorization. Furthermore, our investigation focuses on logical reasoning, offering new insights into how LLMs acquire logical reasoning skills.

# C  DETAILS ON K&K BENCHMARK

## C.1  THE ABSTRACT REPRESENTATION

We use a simple internal representation using basic Python primitives (integer, string and tuple) to encode each K&K puzzle. This allows easy inter-operation with the json format to simplify saving and loading. Specifically, for a $N$-people puzzle, each person is represented by the integer $0, \ldots, N - 1$. Each person's statement is represented by a tuple (type, arguments, ...), where type indicate the statement type listed below:

- **Leaf Statements**: It can be either ('lying', i) or ('telling-truth', i), where i is an integer and this statement assert the ith person is lying or truthful.

- **Composite Statements**: It can take one or more statements as arguments, and has the following types:

  - Negation ('not', statement)
  - Conjunction ('and', statement1, statement2, ...)
  - Disjunction ('or', statement1, statement2, ...)
  - Implication ('->', statement1, statement2)
  - Equivalence ('<=>', statement1, statement2)

## C.2  THE ABSTRACT PUZZLE MODULE: GENERATOR

The Generator samples a problem based on a random seed and a difficulty level specification $(N, W, D)$, where $N$ indicates the number of people, $W$ indicates the max width of each statement, $D$ indicates the max depth of each person's statement. To instantiate the problem, we initialize a random number generator, and sample a statement for each person sequentially. We sample each statement type uniformly at random. For composite statement with variable number of sub-statements, we also randomize the number according to the max width $W$. We restrict the sampling to only leaf statements if the max depth is exhausted. We avoid (skip and resample) some invalid (e.g., asserting self is lying) or uninteresting cases (e.g., a and statement with identical sub-statements).

The following is an example K&K puzzle with 5 people in the abstract representation. We will use this example to illustrate various component in the rest of the section.

---

**Example puzzle of 5 people in the abstract representation**

```
(('and', ('lying', 3), ('telling-truth', 4)),
 ('<=>', ('lying', 3), ('telling-truth', 4)),
 ('telling-truth', 4),
 ('telling-truth', 0),
 ('<=>', ('telling-truth', 2), ('lying', 2)))
```

---

## C.3  THE ABSTRACT PUZZLE MODULE: SOLVER AND REASONER

Each K&K problem can be transformed and solved as a Boolean satisfiability problem. Specifically, consider a puzzle involving $N$ people, a possible solution assign a Boolean value to $N$ variables $B_1, B_2, \ldots, B_N$, where the truth value of $B_i$ indicates whether the $i$th person is telling the truth. By definition, the $i$th person is telling the truth if and only if their statement $S_i$ is true. Therefore, a valid solution to a K&K puzzle is a Boolean assignment for $B_1, B_2, \ldots, B_N$ such that the following formula evaluates to true.

$$(B_1 \Leftrightarrow S_1) \wedge (B_2 \Leftrightarrow S_2) \wedge \cdots \wedge (B_N \Leftrightarrow S_N). \tag{3}$$

We implement our Solver and Reasoner based on this reduction. We take two different approaches here, because we want to find *all* possible solutions in the Solver, and we want to generate *intuitive* intermediate steps for the Reasoner.

Specifically, we are primarily interested in evaluating K&K puzzles with a unique valid solution. Therefore, we design our Solver to use a simple brute-force search that enumerates all possible

Boolean assignments for $N$ people and count the number of assignments that evaluate Eq. (3) to true. In our dataset construction, we only include puzzles whose solution count is exactly one.

In the Reasoner, we are interested in procedurally generating intermediate reasoning steps that lead to the final solution. We note that when explaining the reasoning steps for K&K puzzles, human or off-the-shelf LLMs rarely use the brute-force assignment search approach adopted in our Solver. Instead, they tend to examine the statement from each person sequentially, construct a *partial* assignment for the people examined so far, and backtrack when a contradiction is found. We design our Reasoner following the same procedure.

Specifically, we maintain a queue of people to be examined next, and a partial assignment of knight / knave for people that have been examined so far. In each step, we examine the next person from the queue by adding to the partial assignment the assumed knight / knave role for this person. Given the newly proposed assignment, we go through the known statements and check if there is a contradiction. (A) If a contradiction is found, we record the statement of contradiction as the explanation, and start backtracking. Backtracking will put people back into the to-be-examined queue until we reach a person who has an alternative unexamined role assignment. If no such person is found during backtracking, this means there is no valid solution for this problem. (B) If a contradiction is not found, we can proceed to examine the next person in the queue. Here we also implement a mechanism to reorder the queue so that it may match the human behavior better. For example, if the current person's statement is "If Noah is a knight, then Lily is a knave." then we would bring Noah and Lily to the front of the to-be-examined queue, provided that they are in the queue (i.e., have not been previously examined).

The reasoning steps are generated and stored using a similar format as the abstract representation of the puzzle as described in § C.1. The following snippet shows an example of the generated reasoning steps for the example puzzle shown above:

> **Example of generated reasoning steps in the abstract representation**
>
> ```
> [('proposal', {'assignment': True, 'outcome': 'ok', 'person': 0}),
>  ('proposal', {'assignment': True, 'conflict_statement': (0, True), 'outcome': 'conflict', 'person': 3}),
>  ('proposal', {'assignment': False, 'conflict_statement': (3, False), 'outcome': 'conflict', 'person': 3}),
>  ('reconsider', {'exhausted': [3], 'person': 0}),
>  ('proposal', {'assignment': False, 'outcome': 'ok', 'person': 0}),
>  ('proposal', {'assignment': True, 'conflict_statement': (3, True), 'outcome': 'conflict', 'person': 3}),
>  ('proposal', {'assignment': False, 'outcome': 'ok', 'person': 3}),
>  ('proposal', {'assignment': True, 'conflict_statement': (0, False), 'outcome': 'conflict', 'person': 4}),
>  ('proposal', {'assignment': False, 'outcome': 'ok', 'person': 4}),
>  ('proposal', {'assignment': True, 'conflict_statement': (2, True), 'outcome': 'conflict', 'person': 2}),
>  ('proposal', {'assignment': False, 'outcome': 'ok', 'person': 2}),
>  ('proposal', {'assignment': True, 'conflict_statement': (1, True), 'outcome': 'conflict', 'person': 1}),
>  ('proposal', {'assignment': False, 'outcome': 'ok', 'person': 1}),
>  ('success', {'assignments': (False, False, False, False, False)})]
> ```

### C.4  THE ABSTRACT PUZZLE MODULE: PERTURBER

To support memorization measurement, the K&K Puzzle Perturber generate an perturbed version of a given puzzle. We design the perturbation with the following considerations:

- The perturbation should be "local", in the sense that the perturbed problem should be similar to the original problem when measured in some superficial ways, such as edit distance.
- The perturbation should keep the nature of the problem, i.e., the perturbed problem should be solvable using the same underlying principle, and the difficulty level should be roughly maintained.
- The perturbed puzzle should have a unique solution, which should be different from the solution of the original puzzle.

With those consideration, we support two different perturbations:

- Statement perturbation: randomly choose a person and sample a new statement tree for that person.
- Leaf perturbation: randomly choose a person, and from that person's existing statement tree, randomly choose one leaf node and change it.

## C.5 THE NATURAL LANGUAGE MODULE: NL-GENERATOR

The NL-Generator generate a K&K puzzle in natural language by converting a given abstract-form puzzle into the familiar text form described in natural language. For example, the puzzle presented (in the abstract representation) above can be materialized as follows:

---

**Example puzzle converted to natural language representation**

A very special island is inhabited only by knights and knaves. Knights always tell the truth, and knaves always lie. You meet 5 inhabitants: David, Zoey, Alexander, Aurora, and Isabella. In David's words: "Aurora is a knave and Isabella is a knight". Zoey expressed that Aurora is a knave if and only if Isabella is a knight. Alexander said, "Isabella is a knight." Aurora commented, "David is a knight". According to Isabella, "Alexander is a knight if and only if Alexander is a knave". So who is a knight and who is a knave?

---

Specifically, given a puzzle of $N$ people in the abstract representation, our natural language representation generator first sample $N$ human names, and then format each people's claim by plugging in the corresponding name mapping. We use some heuristics to make the conversion of the tree-structured logic statements to natural language sounds natural. Our current implementation randomly sample from 18 templates of making a statement claim and from the following common names — but this can be easily extended to include more.

```
COMMON_NAMES = ['Emma', 'Liam', 'Olivia', 'Noah', 'Ava', 'Ethan', 'Sophia',
                'Mason', 'Isabella', 'William', 'Mia', 'James', 'Charlotte',
                'Benjamin', 'Amelia', 'Lucas', 'Harper', 'Henry', 'Evelyn',
                'Alexander', 'Abigail', 'Michael', 'Emily', 'Daniel', 'Elizabeth',
                'Jacob', 'Sofia', 'Logan', 'Avery', 'Jackson', 'Ella', 'Sebastian',
                'Scarlett', 'Jack', 'Grace', 'Aiden', 'Chloe', 'Owen', 'Victoria',
                'Samuel', 'Riley', 'Matthew', 'Aria', 'Joseph', 'Lily', 'Luke',
                'Aurora', 'David', 'Zoey', 'Oliver', 'Penelope']
```

## C.6 THE NATURAL LANGUAGE MODULE: NL-REASONER

The NL-Reasoner generates detailed reasoning steps in natural language by converting the output from the abstract Reasoner to natural language descriptions using a similar approach as the NL-Generator. The following show the generated reasoning steps in natural language for the puzzle shown above:

---

**Reasoning steps generated by the Reasoner**

Let's think step by step, by considering whether each person is lying and if that leads to contradiction.

1. Assume David is a knight. No contradiction is found in their claim that Aurora is a knave and Isabella is a knight.

2. Aurora cannot be a knight, because this would contradict the claim of David that Aurora is a knave and Isabella is a knight.

3. Aurora cannot be a knave, because this would contradict the false claim of their own that David is a knight.

4. We have exhausted all possibilities for Aurora, so let us go back and reconsider David.

5. Assume David is a knave. No contradiction is found in their false claim that Aurora is a knave and Isabella is a knight.

6. Aurora cannot be a knight, because this would contradict the claim of their own that David is a knight.

7. Assume Aurora is a knave. No contradiction is found in their false claim that David is a knight.

8. Isabella cannot be a knight, because this would contradict the false claim of David that Aurora is a knave and Isabella is a knight.

9. Assume Isabella is a knave. No contradiction is found in their false claim that Alexander is a knight if and only if Alexander is a knave.

10. Alexander cannot be a knight, because this would contradict the claim of their own that Isabella is a knight.

11. Assume Alexander is a knave. No contradiction is found in their false claim that Isabella is a knight.

12. Zoey cannot be a knight, because this would contradict the claim of their own that Aurora is a knave if and only if Isabella is a knight.

13. Assume Zoey is a knave. No contradiction is found in their false claim that Aurora is a knave if and only if Isabella is a knight.

This leads to a feasible solution.

---

C.7  THE NATURAL LANGUAGE MODULE: NL-PERTURBER

The NL-Perturber generates perturbed puzzles at the language level. Note unlike in the perturbations generated by the abstract Perturber, NL-Perturber keep the underlying abstract puzzle intact and only modify the materialization in natural language. Therefore, the solution to the perturbed puzzle is identical to the solution to the original puzzle. Specifically, the NL-Perturber supports the following perturbations:

With those consideration in mind, we provide two family of perturbations:

- Uncommon name: replace the names of the characters with randomly sampled names from the set of uncommon names.

- Random role: change the role name from knight/knave to other pairs of role names. To avoid introducing bias, we sample from pairs of good/bad role names, including "*saint/sinner, hero/villain, angel/devil, altruist/egoist, sage/fool, pioneer/laggard*".

- Reorder statement: shuffle the order of presenting each person's statement.

- Flip role: change the role from knight/knave to knave/knight, i.e., knave will be telling the truth while knight will be lying.

The uncommon names are sampled from the following list:

```
UNCOMMON_NAMES = [
  'Zephyr', 'Elowen', 'Caspian', 'Isolde', 'Osiris', 'Vesper', 'Thaddeus', 'Ondine',
  'Lysander', 'Xanthe', 'Oberon', 'Calliope', 'Leander', 'Eulalia', 'Florian', 'Forsythe',
  'Nephele', 'Peregrine', 'Ianthe', 'Lazarus', 'Elodie', 'Cillian', 'Ottoline', 'Evander',
  'Saffron', 'Caius', 'Zora', 'Cyprian', 'Amaryllis', 'Theron', 'Perdita', 'Ignatius',
  'Zephyrine', 'Balthazar', 'Melisande', 'Zinnia', 'Sylvester', 'Cosima', 'Leocadio',
  'Percival', 'Oceane', 'Evanthe', 'Zenobia', 'Eurydice', 'Quillan', 'Aeronwen',
  'Thorsten', 'Xiomara', 'Zephyrus', 'Ysolde'
]
```

Note the flip role perturbation is somewhat adversarial as it goes against the common intuition that good role tends to tell the truth while bad role tends to lie. We indeed observe that the models would make a lot of mistakes under this perturbation, despite that the perturbed problem is perfect valid and unambiguous. However, the study of how model's bias impact its reasoning capability is not the main focus of this paper. So we keep this perturbation as reference but primarily focus on "benign" perturbations.

## C.8 DATASET GENERATION

**K&K dataset** During our data construction, we use the maximum width $W = 2$ and depth $D = 2$, and the number of persons in the puzzle $N = 2, 3, 4, 5, 6, 7, 8$.

We present the length distributions of K&K training dataset in Fig. 15. The length distributions of the test dataset are similar to those of the training dataset.

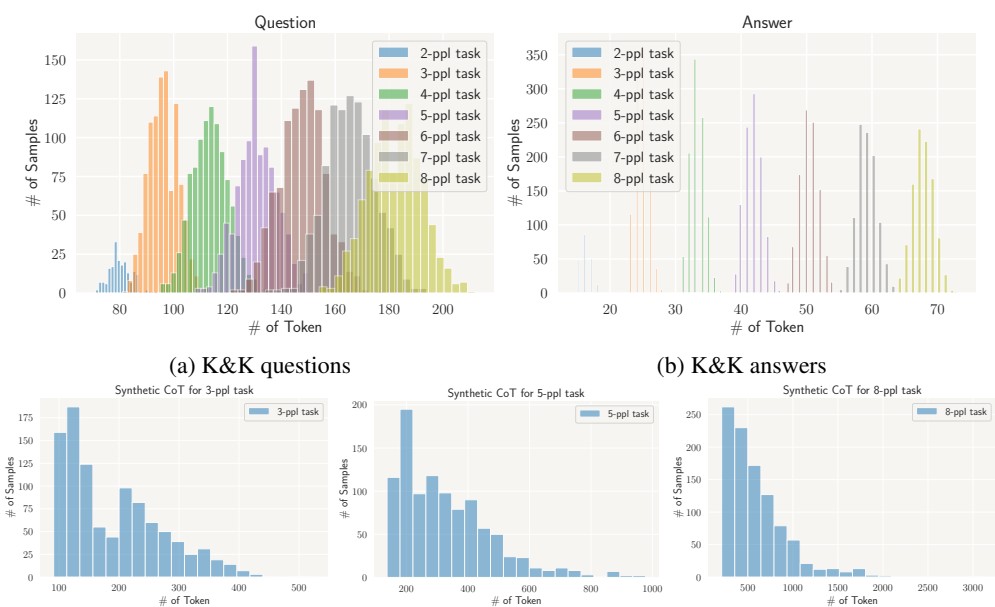

(a) K&K questions      (b) K&K answers

(c) 3-people K&K synthetic CoTs (d) 5-people K&K synthetic CoTs (e) 8-people K&K synthetic CoTs

Figure 15: Length distributions of K&K training data.

**Local perturbation** Tab. 1 presents the example knight (truth-teller) and knave (liar) scenario involving two people: Liam and Aria, with corresponding logical statements, and converted English statements, questions, and answers. It also shows three versions of the problems: an original example, a leaf-perturbed version, and a statement-perturbed version. Specifically, (1) leaf perturbation changes a "leaf" in the logical tree - a single truth value. In this case, it flipped Jacob's status in Oliver's statement from knave (liar) to knight (truth-teller) (2) Statement perturbation changes the entire structure of a statement. Here, it changed Oliver's statement entirely. Both perturbations re-

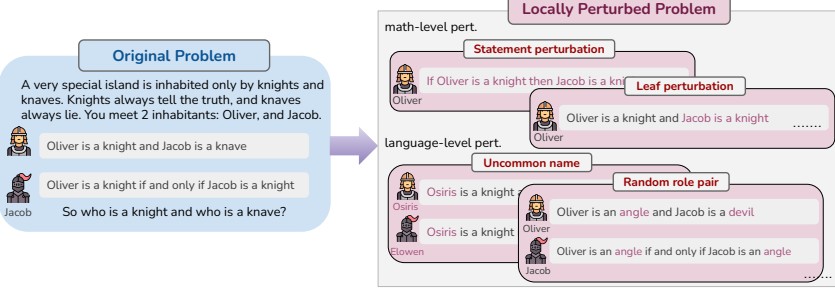

Figure 16: Comparison between different locally perturbed problems.

sult in changing the answer. The leaf perturbation creates a subtle change in one statement that flips the logical outcome, while the statement perturbation changes the entire one statement.

Table 1: 2-person puzzle generation with the knight (telling-truth) and knave (lying) and comparison between original sample, leaf-perturbed sample, and statement-perturbed sample.

| Type | Example | Leaf Perturbed Example | Statement Perturbed Example |
|---|---|---|---|
| person | Oliver (person index 0), Jacob (person index 1) | | |
| logical statement | Oliver: ('and', ('telling-truth', 0), ('lying', 1)) Jacob: ('⇔', ('telling-truth', 0), ('telling-truth', 1)) | Oliver: ('and', ('telling-truth', 0), (**'telling-truth', 1**)) Jacob: ('⇔', ('telling-truth', 0), ('telling-truth', 1)) | Oliver: (**'→', ('telling-truth', 0)**, (**'telling-truth', 1**)) Jacob: ('⇔', ('telling-truth', 0), ('telling-truth', 1)) |
| English statement | Oliver: Oliver is a knight and Jacob is a knave Jacob: Oliver is a knight if and only if Jacob is a knight | Oliver: Oliver is a knight and **Jacob is a knight** Jacob: Oliver is a knight if and only if Jacob is a knight | Oliver: **If Oliver is a knight then Jacob is a knight** Jacob: Oliver is a knight if and only if Jacob is a knight |
| question | A very special island is inhabited only by knights and knaves. Knights always tell the truth, and knaves always lie. You meet 2 inhabitants: Oliver, and Jacob. Oliver commented, "Oliver is a knight and Jacob is a knave". Jacob remarked, "Oliver is a knight if and only if Jacob is a knight". So who is a knight and who is a knave? | A very special island is inhabited only by knights and knaves. Knights always tell the truth, and knaves always lie. You meet 2 inhabitants: Oliver, and Jacob. Oliver commented, "Oliver is a knight and Jacob is a knight". Jacob remarked, "Oliver is a knight if and only if Jacob is a knight". So who is a knight and who is a knave? | A very special island is inhabited only by knights and knaves. Knights always tell the truth, and knaves always lie. You meet 2 inhabitants: Oliver, and Jacob. Oliver commented, "If Oliver is a knight then Jacob is a knight". Jacob remarked, "Oliver is a knight if and only if Jacob is a knight". So who is a knight and who is a knave? |
| answer | (1) Oliver is a knight (2) Jacob is a knave | (1) Oliver is a knight (2) **Jacob is a knight** | (1) Oliver is a knight (2) **Jacob is a knight** |

Moreover, we compare the math-level perturbation with language-level perturbation in Fig. 16.

As mentioned in § 2, the Perturber of the abstract puzzle module generates a perturbed puzzle with a *unique* solution that is *different* from the original puzzle, or until the maximum number of attempts is reached. We set this limit to 2000 attempts.

- For statement perturbation, the Perturber can always return a valid perturbed puzzle due to the large perturbation space.

- For leaf perturbation, since the process is restricted to a single leaf node, it may not always find a valid perturbed puzzle within the constraints of unique and different solution. Below are the detailed proportions of valid leaf perturbations on training samples (under 2000 max attempts for each sample): 76% valid for 2-person task; 93.4% valid for 3-person task; 95.4% valid for 4-person task; 98.8% valid for 5-person task; 99.5% valid for 6-person task; 100% valid for 7/8-person tasks.

# D EXPERIMENTAL SETUPS

## D.1 MODELS

Tab. 2 provides the details of the models evaluated in our study.

Table 2: HuggingFace links or endpoint specifications for evaluated models.

| Model | Link |
|---|---|
| Llama3-8B | https://huggingface.co/meta-llama/Meta-Llama-3-8B |
| Phi-3-mini | https://huggingface.co/microsoft/Phi-3-mini-4k-instruct |
| Phi-3-medium | https://huggingface.co/microsoft/Phi-3-medium-4k-instruct |
| NuminaMath-7B-CoT | https://huggingface.co/AI-MO/NuminaMath-7B-CoT |
| Deepseek-Math-7B | deepseek-ai/deepseek-math-7b-instruct |
| Claude-3.5-Sonnet | https://www.anthropic.com/news/claude-3-5-sonnet, claude-3-5-sonnet-20240620 endpoint |
| GPT4o-mini | https://platform.openai.com/docs/models/, gpt-4o-mini-2024-07-18 endpoint |
| GPT4o | https://platform.openai.com/docs/models/, gpt-4o-2024-05-13 endpoint |
| Gemini-1.5-Flash-002 | https://console.cloud.google.com/vertex-ai/model-garden, gemini-1.5-flash-002 endpoint |
| Gemini-1.5-Prof-002 | https://console.cloud.google.com/vertex-ai/model-garden, gemini-1.5-pro-002 endpoint |

## D.2 EXPERIMENTAL DETAILS

### D.2.1 EVALUATION

By default, we utilize zero-shot direct prompting with task-specific instructions for open-ended question-answering. We employ the following prompt:

---

**0-shot Direct Prompting**

Your task is to solve a logical reasoning problem. You are given set of statements from which you must logically deduce the identity of a set of characters.

You must infer the identity of each character. At the end of your answer, you must clearly state the identity of each character by following the format:

CONCLUSION:
(1) ...
(2) ...
(3) ...

### Question: {question}
### Answer:

---

In addition to the 0-shot direct prompting used in the main paper, we explore 0-shot Chain of Thought (CoT) prompting and 1-shot direct/CoT prompting and report the results in Appendix § E.

---

**0-shot CoT Prompting**

Your task is to solve a logical reasoning problem. You are given set of statements from which you must logically deduce the identity of a set of characters.

You must infer the identity of each character. First, explain your reasoning. At the end of your answer, you must clearly state the identity of each character by following the format:

CONCLUSION:
(1) ...
(2) ...
(3) ...

### Question: {question}
### Answer: Let's think step by step

---

In addition, we utilize a specific CoT prompting format for instruction-tuned models: DeepSeek-Math-7B and NuminaMath-7B-CoT, as recommended by their developers:

---

Please reason step by step, and put your final answer within \boxed{}.

---

This replaces the previous prompt, "Let's think step by step."

## 1-shot Direct Prompting

Your task is to solve a logical reasoning problem. You are given set of statements from which you must logically deduce the identity of a set of characters.

You must infer the identity of each character. At the end of your answer, you must clearly state the identity of each character by following the format:

CONCLUSION:
(1) ...
(2) ...
(3) ...

### Question: A very special island is inhabited only by knights and knaves. Knights always tell the truth, and knaves always lie. You meet 2 inhabitants: Jack, and Sophia. Jack tells you that Sophia is not a knave. Sophia says that If Jack is a knight then Sophia is a knight. So who is a knight and who is a knave?
### Answer:
CONCLUSION:
(1) Jack is a knight
(2) Sophia is a knight

### Question: {question}
### Answer:

---

**1-shot CoT Prompting**

Your task is to solve a logical reasoning problem. You are given set of statements from which you must logically deduce the identity of a set of characters.

You must infer the identity of each character. First, explain your reasoning. At the end of your answer, you must clearly state the identity of each character by following the format:

CONCLUSION:
(1) ...
(2) ...
(3) ...

### Question: A very special island is inhabited only by knights and knaves. Knights always tell the truth, and knaves always lie. You meet 2 inhabitants: Ella, and Penelope. In a statement by Ella: "Ella is a knight or Penelope is a knight". According to Penelope, "Ella is a knave if and only if Penelope is a knight". So who is a knight and who is a knave?
### Answer: Let's think step by step, by considering whether each person is lying and if that leads to contradiction. Assume Ella is a knight. Penelope cannot be a knight, because this would contradict the claim of their own. Penelope cannot be a knave, because this would contradict the false claim of their own. We have exhausted all possibilities for Penelope, so let us go back and reconsider Ella. Assume Ella is a knave. Penelope cannot be a knight, because this would contradict the false claim of Ella. Assume Penelope is a knave. This leads to a feasible solution.
CONCLUSION:
(1) Ella is a knave
(2) Penelope is a knave

### Question: {question}
### Answer: Let's think step by step

---

In our evaluation process, we use greedy decoding with temperature $t = 0$ for all models and a maximum token length of 2048.

To assess the correctness, we implement keyword matching: a response is considered correct if each person's ground truth identity appears in the conclusion part of the model's output.

### D.2.2   FINE-TUNING

**Training instance**   Each training instance in Direct FT includes the task instruction, question, and the correct conclusion. In CoT FT, each training instance includes the task instruction, question, synthetic reasoning steps, and the correct conclusion. Both formats are structured similarly to task instructions followed by a single demonstration used in 1-shot Direct Prompting or 1-shot CoT Prompting.

**Training loss**   In Direct FT, the loss for each training instance is computed on the tokens that appear directly after "### Answer:\n". In CoT FT, the loss is calculated on the tokens that appear directly after "### Answer: Let's think step by step".

**Training hyperparameters**   For Llama3-8B fine-tuning, we used LoRA fine-tuning with the following standard hyperparameters: a batch size of 4, gradient accumulation steps of 8, and 5e-5 learning rate. The LoRA configuration was set as follows: rank $r = 32$, scaling factor $\alpha = 32$, and dropout rate 0.05. No quantization techniques were used. We fine-tune for a maximum of 100 epochs. We primarily reported results before 50 epochs, as we found the model typically converged by then.

For GPT4o-mini fine-tuning, we utilized the default hyperparameters provided by the OpenAI fine-tuning API. The model was fine-tuned for 5 epochs to achieve high accuracy within reasonable budget.

**Reported Training accuracy**  For GPT4o-mini, the training accuracy for each $N$-people K&K task is calculated using 100 training samples due to budget constraints on API usage. For open-source Llama3-8B, the training accuracy is based on the full set of training samples.

### D.2.3  PROBING

As described in § 4.2, in the probing experiments, we train logistic regression models on the model's intermediate outputs from different transformer blocks, to distinguish between correct and incorrect statements. For each transformer block, we extract the MLP layer's output.

The correct/incorrect statements consist of a K&K puzzle and a conclusion about a character's role in the puzzle. For example, considering the following 2-people K&K puzzle:

> A very special island is inhabited only by knights and knaves. Knights always tell the truth, and knaves always lie. You meet 2 inhabitants: Oliver, and Ethan. Oliver told you that Oliver is a knight or Ethan is a knave. In a statement by Ethan: "Oliver is a knight". So who is a knight and who is a knave?

with the correct answer being

> Oliver is a knight, and Ethan is a knight.

We can generate two correct statements:

- A very special island is inhabited only by knights and knaves. Knights always tell the truth, and knaves always lie. You meet 2 inhabitants: Oliver, and Ethan. Oliver told you that Oliver is a knight or Ethan is a knave. In a statement by Ethan: "Oliver is a knight". So who is a knight and who is a knave? **Oliver is a knight.**

- A very special island is inhabited only by knights and knaves. Knights always tell the truth, and knaves always lie. You meet 2 inhabitants: Oliver, and Ethan. Oliver told you that Oliver is a knight or Ethan is a knave. In a statement by Ethan: "Oliver is a knight". So who is a knight and who is a knave? **Ethan is a knight.**

And two incorrect statements:

- A very special island is inhabited only by knights and knaves. Knights always tell the truth, and knaves always lie. You meet 2 inhabitants: Oliver, and Ethan. Oliver told you that Oliver is a knight or Ethan is a knave. In a statement by Ethan: "Oliver is a knight". So who is a knight and who is a knave? **Oliver is a knave.**

- A very special island is inhabited only by knights and knaves. Knights always tell the truth, and knaves always lie. You meet 2 inhabitants: Oliver, and Ethan. Oliver told you that Oliver is a knight or Ethan is a knave. In a statement by Ethan: "Oliver is a knight". So who is a knight and who is a knave? **Ethan is a knave.**

### D.2.4  DISTINGUISHING MEMORIZATION FROM REASONING

For GPT4o-mini and Llama3-8B, we calculate the memorization score for each training sample within each complete $N$-people K&K training dataset. As discussed in § 6, we omit samples where $\mathsf{Acc}(f; x) = 0$ and label the remaining samples based on whether they are consistently solved under perturbation. We then split the dataset into 80%/20% train/test sets and perform binary classification.

### D.2.5 COMPUTATION RESOURCES

The fine-tuning experiments are conducted on 2 NVIDIA A100 GPU cards, each with 80GB of memory. The LLM evaluation experiments can be conducted on one NVIDIA RTX A6000 GPU card with 48 GB of memory.

## E ADDITIONAL EXPERIMENTAL RESULTS

### E.1 MEMORIZATION MEASUREMENT

**Off-the-shelf models** We evaluate Llama3-8B, Phi-3-mini, Phi-3-medium, NuminaMath-7B-CoT, and Deepseek-Math-7B using 0/1-shot Direct/CoT prompting in Fig. 17. The results indicate that these open-source models exhibit poor accuracy on K&K tasks, particularly as the number of people in the K&K puzzles increases. Different prompting methods do not significantly enhance performance. Additionally, the models struggle to consistently solve the K&K prompts under local perturbations, as shown by the memorization scores under statement and lead perturbations.

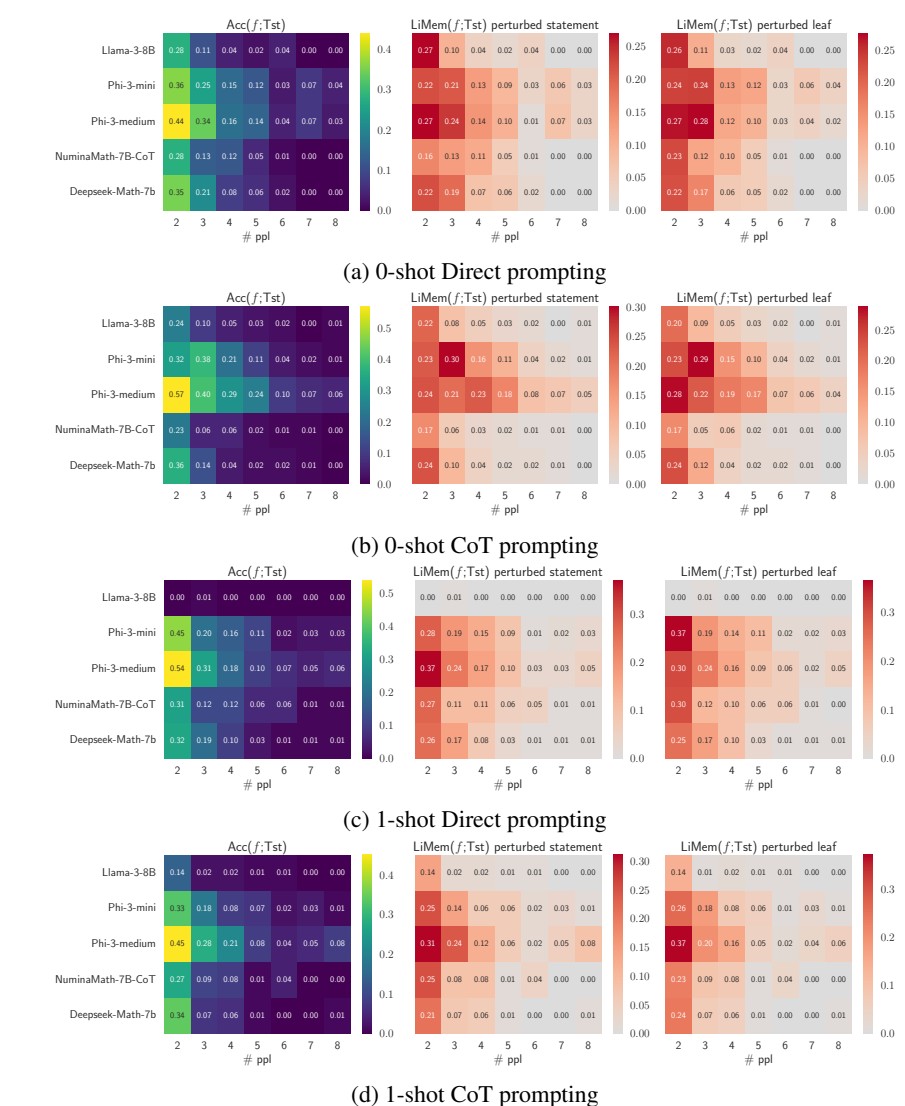

Figure 17: Acc($f$; Tst) and LiMem($f$; Tst) of off-the-shelf models under various prompt formats.

**Fine-tuned models** As shown in Fig. 18, the test accuracy (y-axis) of CoT-FTed or Direct-FTed GPT4o-mini on the unseen test set continues to increase over the epochs, despite that the memoriza-

tion score LiMem($f$; Tr) on training samples also increases. The memorization score LiMem($f$; Tr) under role-flipping is significantly higher than other perturbation, possibly due to an internal bias that knights are truthful.

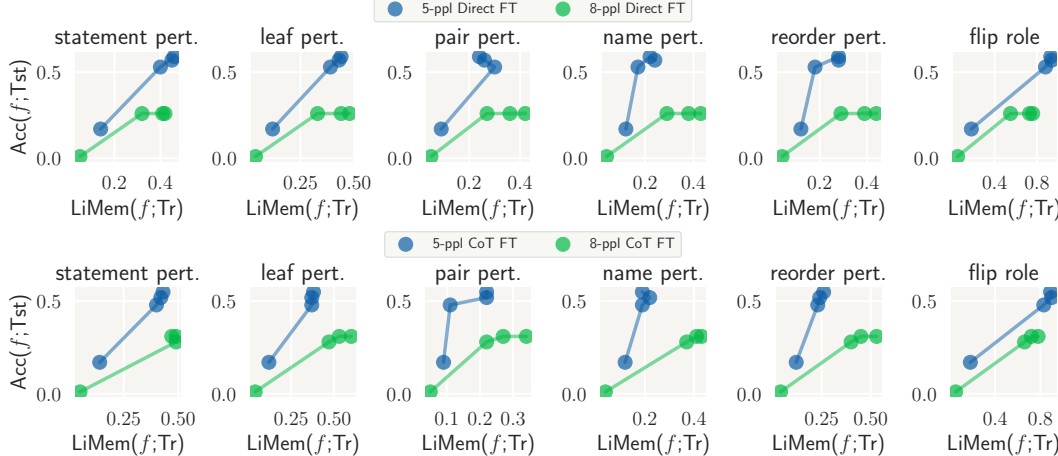

Figure 18: Test accuracy of fine-tuned GPT4o-mini (first row: Direct FT; second row: CoT FT) increase with Acc($f$; Tr), despite that the memorization becomes stronger as reflected by larger LiMem($f$; Tr) under leaf perturbation.

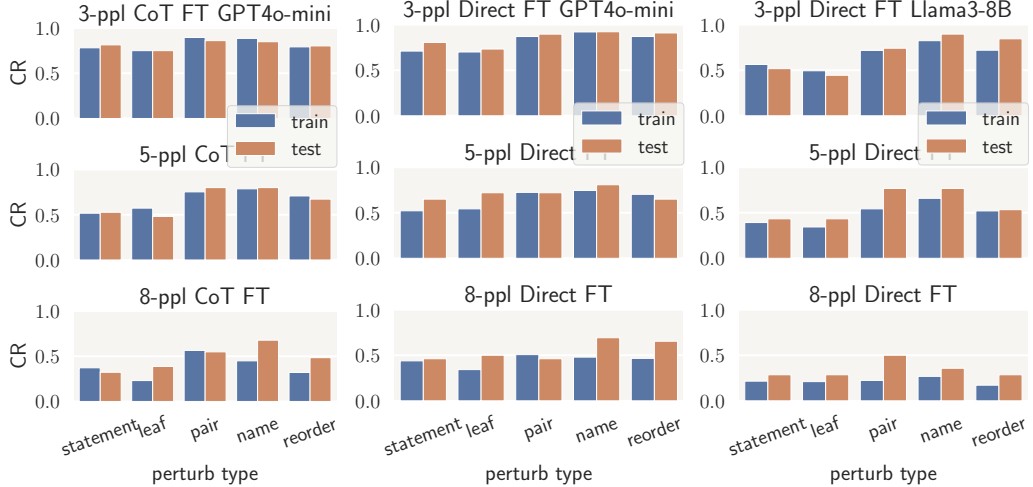

Figure 19: Consistency Ratio (CR ↑) under local perturbations. Fine-tuned LLMs generally demonstrate a higher consistency ratio on solved problems in the test set compared to the train set, particularly for challenging tasks such as 5/8-person puzzles. On the 3-person puzzle task, the consistency ratio between the train and test sets remains comparable.

Table 3: Self-consistency (Wang et al., 2023b) can enhance the accuracy of GPT4o-mini on the easy 2-ppl K&K task, but has limited improvement on 3-ppl task and fails to help on the challenging 8-ppl task, which suggests that the model cannot fundamentally solve such complex problems. Self-consistency also reduces memorization scores on 2-ppl/3-ppl tasks, likely due to its majority voting mechanism that leads to robust reasoning results.

| Method | Test Accuracy | | | Memorization Score | |
|---|---|---|---|---|---|
| | 2-ppl | 3-ppl | 8-ppl | 2-ppl | 3-ppl |
| Direct Prompting | 0.63 | 0.42 | 0.01 | 0.24 | 0.26 |
| Direct Prompting + Self-consistency | 0.74 | 0.43 | 0.02 | 0.20 | 0.22 |

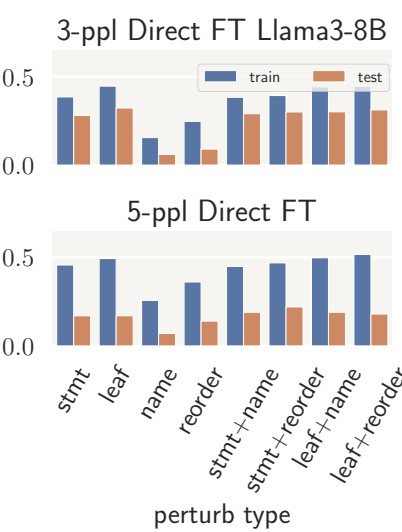

Figure 20: Memorization scores of Directly Fine-Tuned Llama3-8B under various math-level (statement, leaf) and language-level (name, reorder) perturbations. Combining math-level and language-level perturbations progressively can result in higher memorization scores (e.g., leaf + reorder), especially compared to applying language-level perturbations alone.

## E.2 EVALUATION ON REASONING CAPABILITY

### E.2.1 LLAMA3-8B

**Accuracy over epochs** Fig. 21 reports the train and test accuracy (under different evaluation configurations) for the Llama3-8B model fine-tuned on $N$-person tasks across multiple training epochs.

**Transferability** We present the transferability results for the K&K task across different problem sizes and training epochs in Fig. 22 and Fig. 23. Fig. 22 shows the accuracy improvements relative to the baseline with no fine-tuning, while Fig. 23 reports the absolute accuracy values.

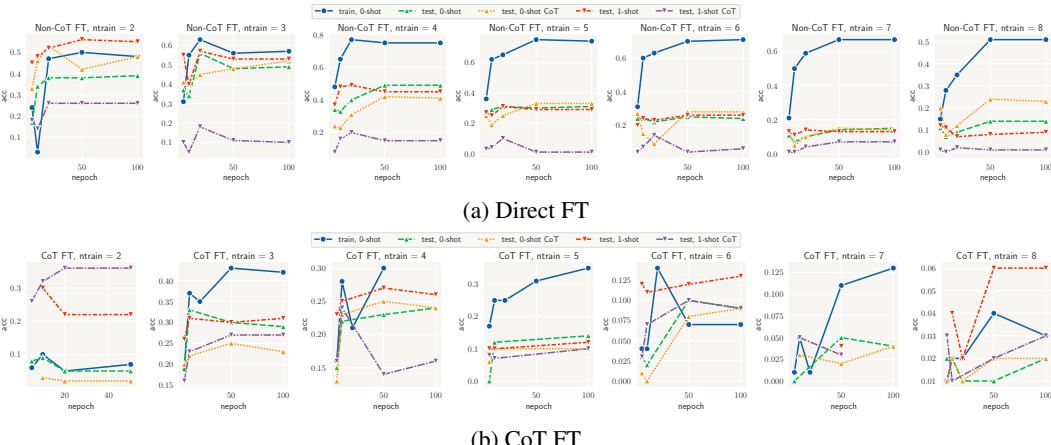

(a) Direct FT

(b) CoT FT

Figure 21: Train and test accuracy (under different evaluation configurations) for the Llama3-8B model fine-tuned on $N$-person tasks across multiple training epochs.

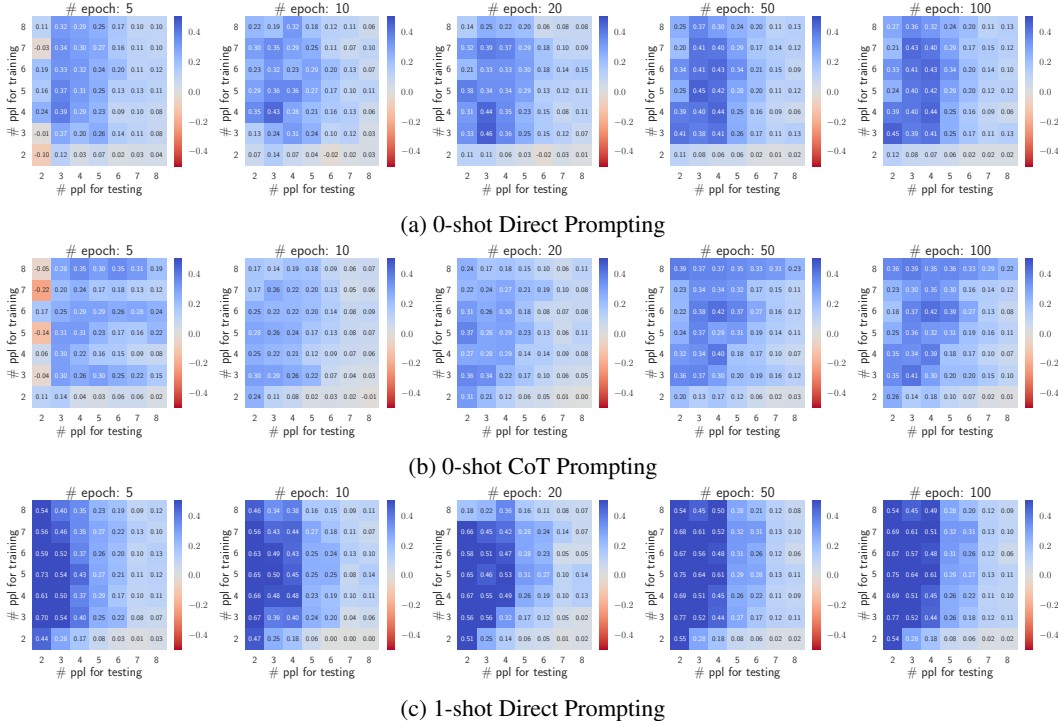

(a) 0-shot Direct Prompting

(b) 0-shot CoT Prompting

(c) 1-shot Direct Prompting

Figure 22: Improvement in test accuracy on $N$-person problems for Llama3-8B fine-tuned on $M$-person problems **with direct FT**, compared to the unfine-tuned model, under various evaluation configurations.

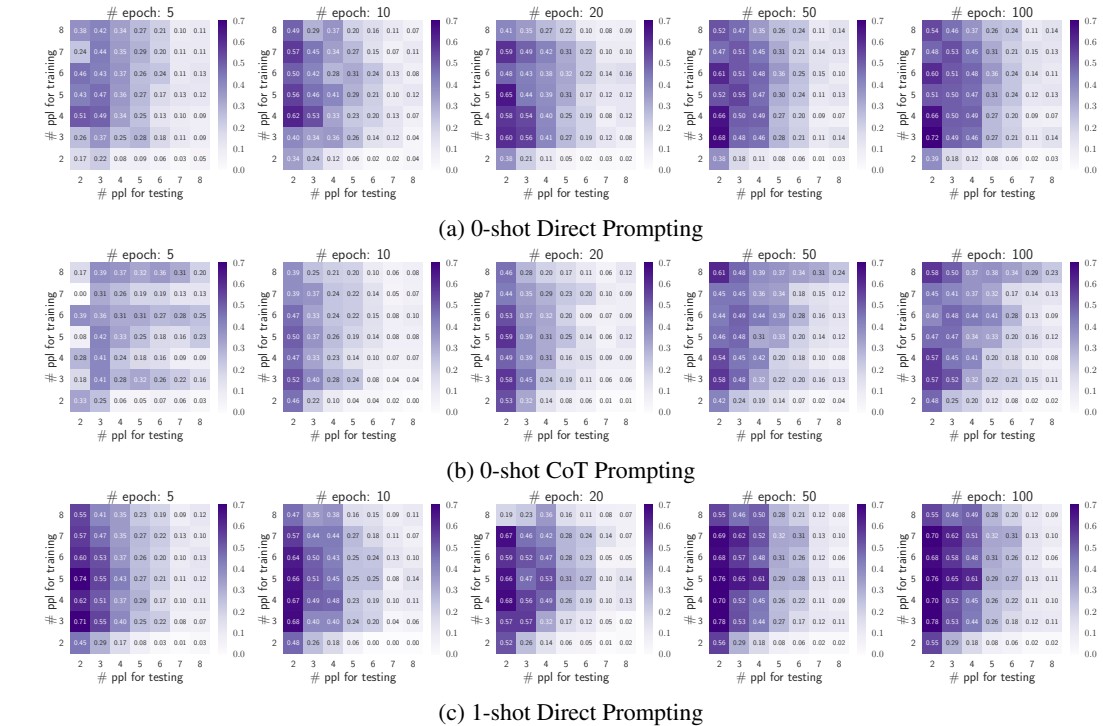

(a) 0-shot Direct Prompting

(b) 0-shot CoT Prompting

(c) 1-shot Direct Prompting

Figure 23: Test accuracy on $N$-person problems for Llama3-8B fine-tuned on $M$-person problems **with direct FT**, under various evaluation configurations.

**Fine-tuning on $10k$ 8-people K&K samples** The results in Fig. 24 shows that $10k$ fine-tuning achieves significantly higher test accuracy than $1k$ fine-tuning on all tasks. Direct FT with $10k$ puzzles shows surprisingly high test accuracy, e.g., 87% accuracy on 3-person tasks, where the un-FTed model has nearly 0 accuracy as shown in Fig. 3. Notably, the models don't see reasoning steps during training and rely solely on memorizing answers. It also suggests that training on the hardest (8-person) tasks helps the model learn certain underlying rules that can be transferred to solve easier tasks.

However, the test accuracy drops for Llama3-8B when Direct FTing on $10k$ samples for overly long epochs, especially evaluated on 2-people K&K task, potentially due to overfitting to the more complicated 8-people training task.

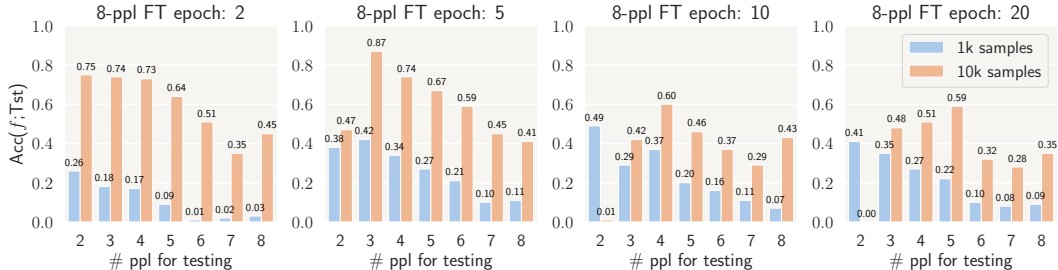

Figure 24: Transferability of Llama3-8B Direct-FTed on $1k/10k$ samples at different epochs.

### E.2.2 GPT4O-MINI

**Accuracy over epochs** Fig. 25 reports the train and test accuracy (under different evaluation configurations) for the GPT4o-mini model fine-tuned on $N$-person tasks across multiple training epochs.

Using the same paradigm for training and evaluation (i.e., Direct FT & Direct Prompting, CoT FT & CoT Prompting) usually achieves the best accuracy for GPT4o-mini on training dataset and test dataset. We focus on 0-shot setting for GPT4o-mini evaluation given its stronger capacity and higher accuracy than Llama3-8B.

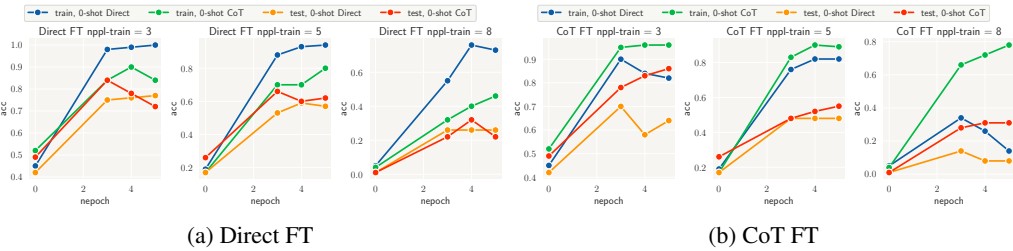

     (a) Direct FT                  (b) CoT FT

Figure 25: Train and test accuracy (under different evaluation configurations) for the GPT4o-mini model fine-tuned on $N$-person tasks across multiple training epochs.

**Transferability** We present the transferability results for the K&K task across different problem sizes and training epochs in Fig. 26 and Fig. 27. Fig. 26 shows the accuracy improvements relative to the baseline with no fine-tuning, while Fig. 27 reports the absolute accuracy values.

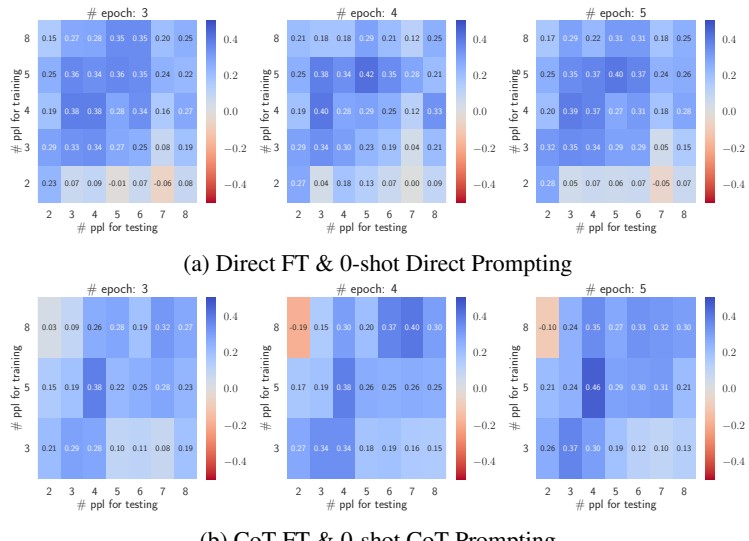

     (a) Direct FT & 0-shot Direct Prompting

     (b) CoT FT & 0-shot CoT Prompting

Figure 26: Improvement in test accuracy on $N$-person problems for GPT4o-mini fine-tuned on $M$-person problems, under two finetuning/evaluation configurations.

**Fine-tuning on $10k$ 8-people K&K samples** We present the transferability results with absolute test accuracy for the K&K task across different 8-people task training sizes and training epochs in Fig. 28. As shown, GPT4o-mini achieves high accuracy on all tasks at early epochs (e.g., 3 epochs). We also find that GPT4o-mini exhibits poor test accuracy on two-person testing puzzles when CoT-FTed on $10k$ 8-people puzzles, unlike the Direct FTed model that have stable performance across all task. In the failure case below, the CoT-FTed GPT4o-mini gets stuck in a loop of listing assumptions and contradictions, resulting in long, repetitive responses without reaching a conclusion.

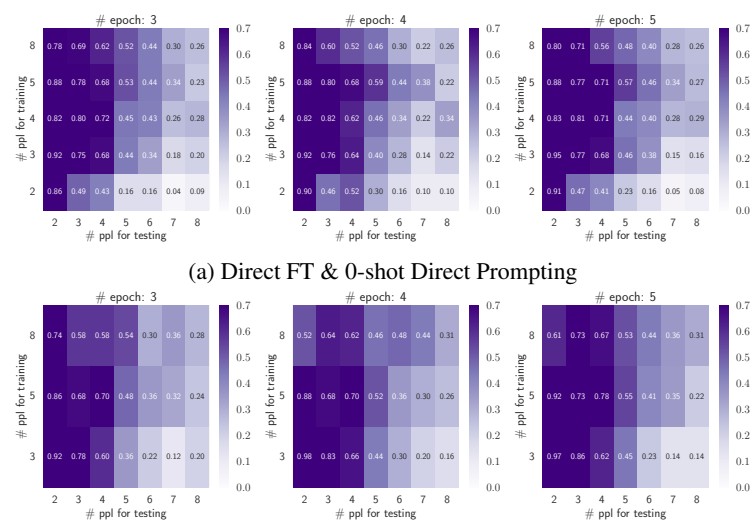

Figure 27: Test accuracy on $N$-person problems for GPT4o-mini fine-tuned on $M$-person problems, under two finetuning/evaluation configurations.

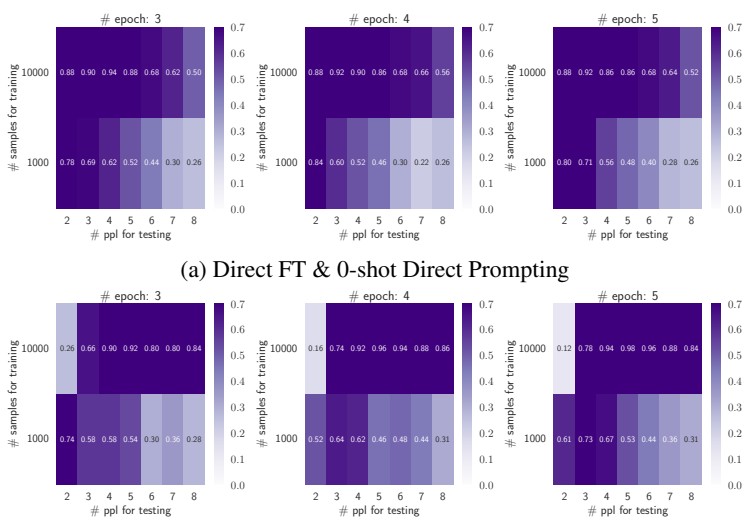

Figure 28: Test accuracy on $N$-person problems for GPT4o-mini fine-tuned on $1k/10k$ 8-people problems, under two finetuning/evaluation configurations.

---

**Failure case of GPT4o-mini CoT FTed on $10k$ 8-people K&K task**

{omitted task instruction}
### Question: A very special island is inhabited only by knights and knaves. Knights always tell the truth, and knaves always lie. You meet 2 inhabitants: Logan, and Olivia. Logan was heard saying, "Logan is a knight". "If Olivia is a knight then Logan is a knight" - Olivia. So who is a knight and who is a knave?
### Answer: Let's think step by step

*// response from GPT4o-mini*
by considering whether each person is lying and if that leads to contradiction. Assume Logan is a knight. No contradiction is found in their claim that Logan is a knight. Assume Olivia is a knight. No contradiction is found in their claim that If Olivia is a knight then Logan is a knight. Assume this leads to no contradiction. No contradiction is found in their claim that If Olivia is a knight then Logan is a knight. This leads to no contradiction in their claim that Logan is a knight. This leads to no contradiction in their claim that If Olivia is a knight then Logan is a knight. This leads to no contradiction in their claim that Logan is a knight. This leads to no contradiction in their claim that If Olivia is a knight then Logan is a knight. This leads to no contradiction in their claim that If Olivia is a knight then Logan is a knight. This leads to no contradiction in their claim that Logan is a knight. This leads to no contradiction in their claim that If Olivia is a knight then Logan is a knight. This leads to no contradiction in their claim that Logan is a knight.

### E.3 DIRECT FT WITH WRONG ANSWERS

### E.3.1 LLAMA3-8B

Fig. 29, Fig. 30 and Fig. 31 show the results of Direct FT with $100\%$, $75\%$ and $50\%$ incorrect answers for the Llama3-8B model across different prompting setups. Consistent with our earlier findings in § 4.3, fine-tuning with incorrect answers still significantly improves K&K performance, especially with 0-shot CoT prompting or 1-shot direct prompting.

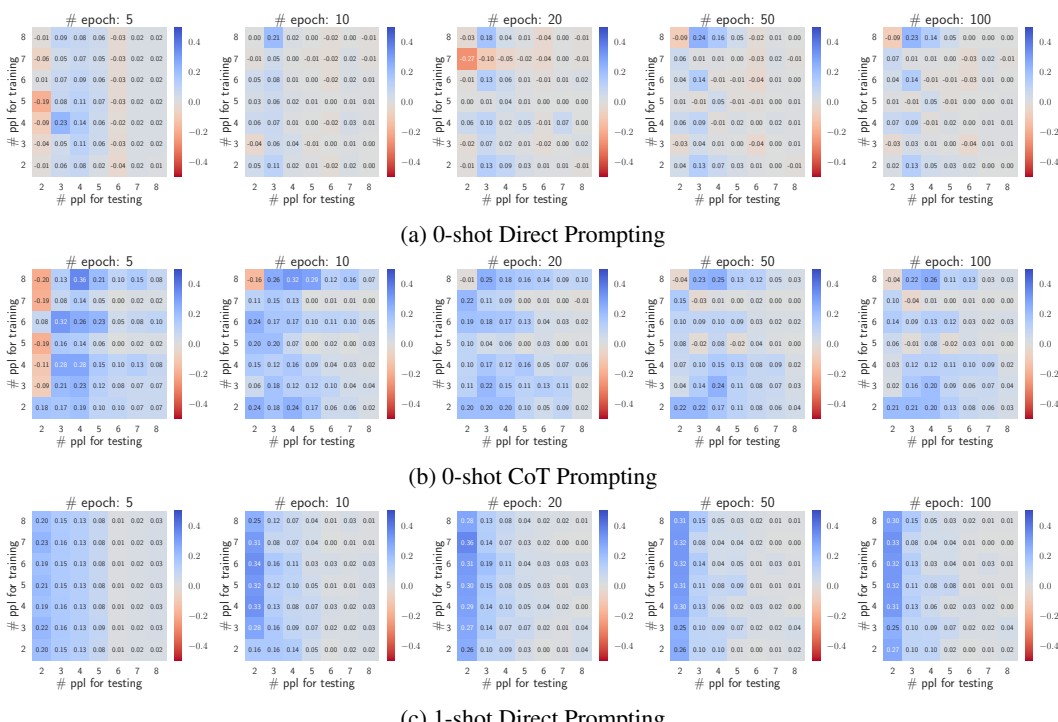

(a) 0-shot Direct Prompting

(b) 0-shot CoT Prompting

(c) 1-shot Direct Prompting

Figure 29: Improvement in test accuracy on $N$-person problems for Llama3-8B fine-tuned on $M$-person problems **with completely wrong answers**, compared to the unfine-tuned model, under various evaluation configurations.

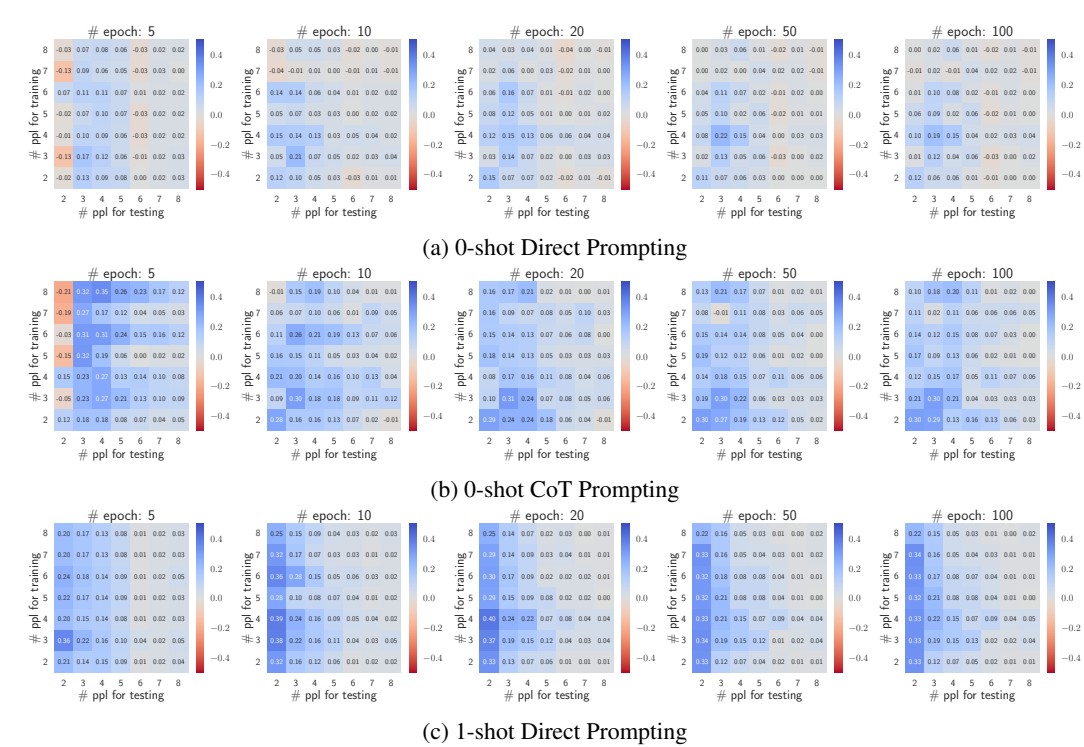

Figure 30: Improvement in test accuracy on $N$-person problems for Llama3-8B fine-tuned on $M$-person problems **with 75% wrong answers**, compared to the unfine-tuned model, under various evaluation configurations.

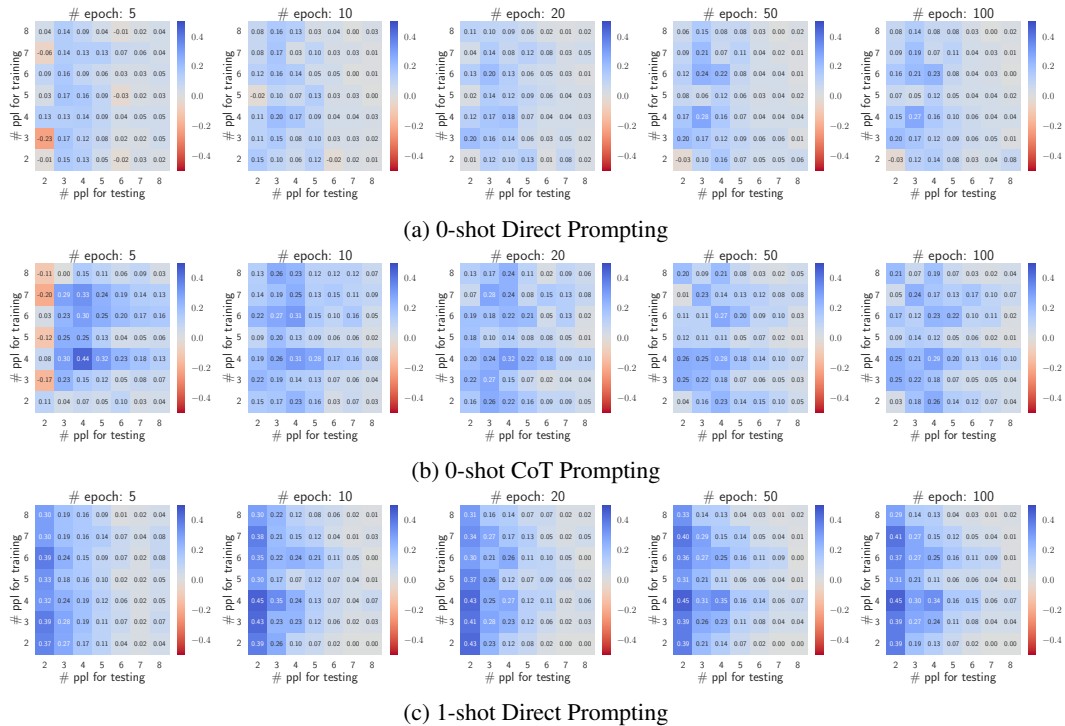

Figure 31: Improvement in test accuracy on $N$-person problems for Llama3-8B fine-tuned on $M$-person problems **with 50% wrong answers**, compared to the unfine-tuned model, under various evaluation configurations.

### E.3.2 GPT4O-MINI

Fig. 32 displays the results of direct fine-tuning using 5-people training K&K puzzles for the GPT4o-mini model, containing varying percentages of incorrect answers in the dataset: $100\%, 75\%, 50\%, 25\%, 0\%$. This is evaluated across different epochs in the five-person puzzle. As noted in § 4.3, when the training dataset includes 50% or fewer samples with incorrect answers, fine-tuning can still enhance K&K's performance across various testing tasks.

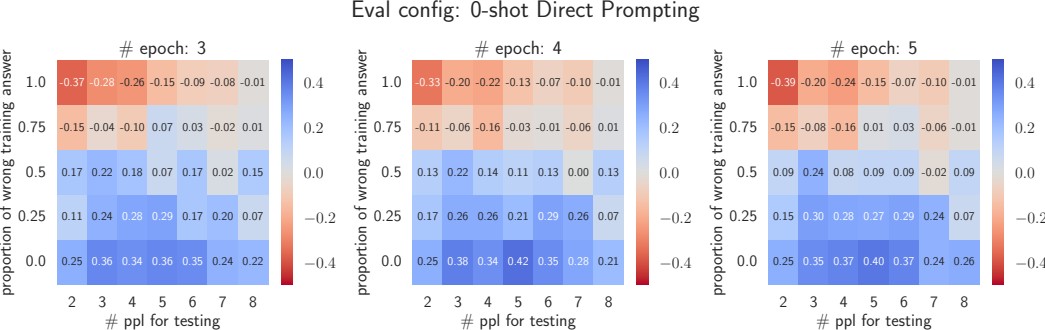

Figure 32: Improvement in test accuracy on $N$-people problems for GPT4o-mini fine-tuned on 5-people problems with different proportion of wrong answers, compared to the unfine-tuned model. Direct FT with 50% wrong answers still improves K&K performance.

### E.4 PROBING

We report the probing accuracy for the un-fine-tuned Llama3-8B model in Fig. 33. As shown, without fine-tuning, the model demonstrates relatively low probing accuracy, with values usually below 90%.

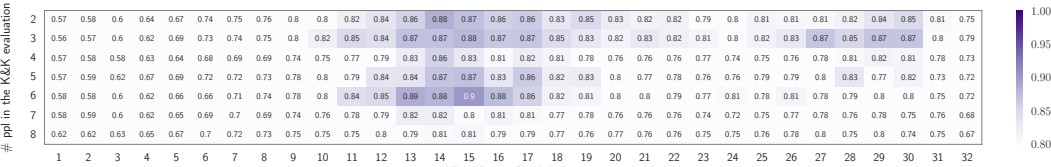

Figure 33: Probing accuracy of K&K puzzles with different number of people in testing puzzles across different layers of the un-finetuned Llama3-8B transformer model.

### E.5 DISTINGUISHING MEMORIZATION FROM REASONING

**Puzzle-based indicators**  Fig. 34 shows the train and test AUC for predicting whether $N$-person puzzles can be consistently solved by a specific model under perturbations, using puzzle-based indicators. The results indicate that length-related features are useful for distinguishing memorization from reasoning. Notably, the test AUC is generally higher for CoT FTed GPT4o-mini compared to Direct FTed GPT4o-mini.

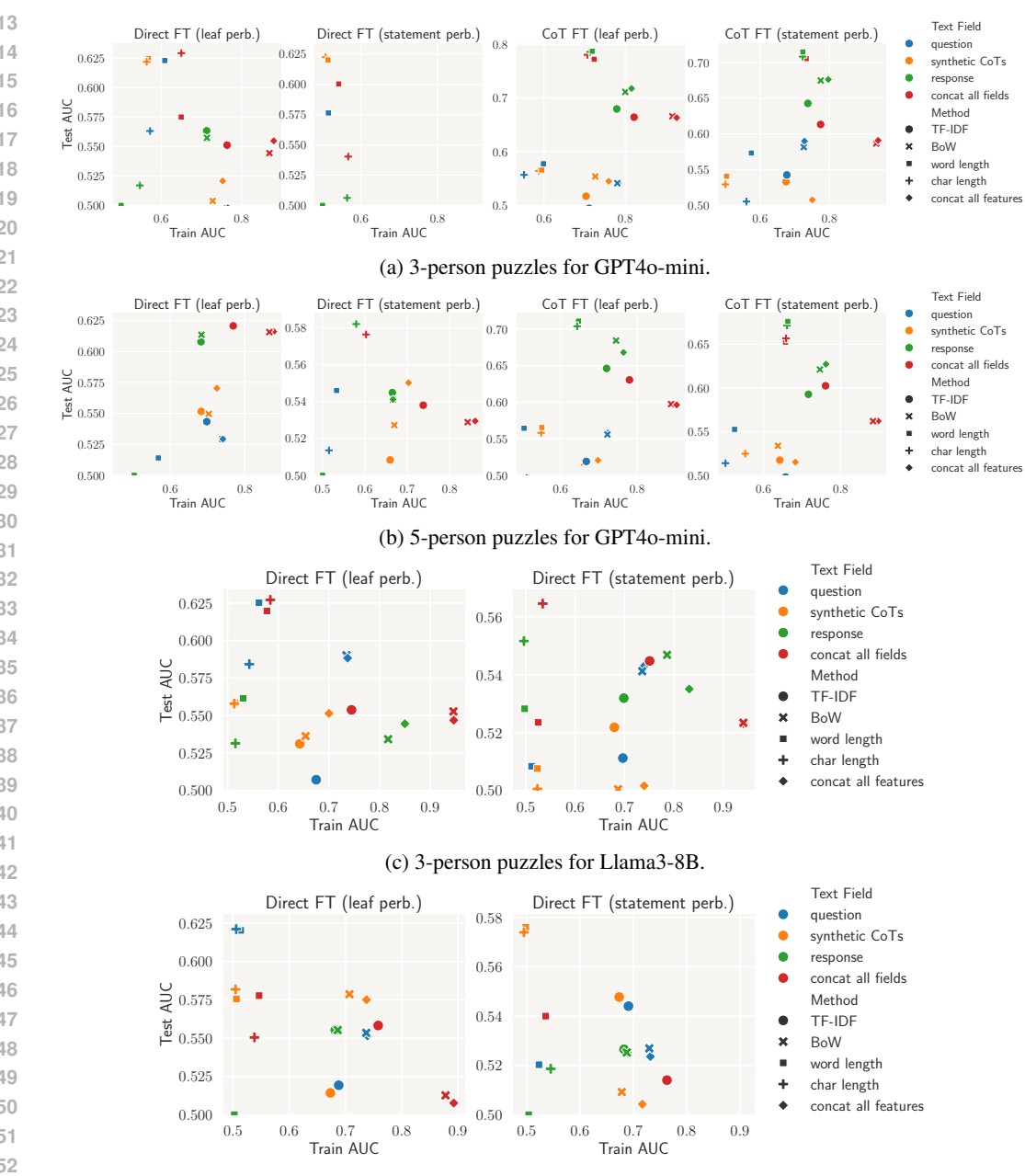

(a) 3-person puzzles for GPT4o-mini.

(b) 5-person puzzles for GPT4o-mini.

(c) 3-person puzzles for Llama3-8B.

(d) 5-person puzzles for Llama3-8B.

Figure 34: AUC for predicting whether $N$-person puzzles can be consistently solved under perturbations based on puzzle-based indicators.

**Model-based indicators**  We report test AUC for classifying puzzles based on whether they are consistently solved under leaf/statement perturbation by the Llama3-8B model Direct-FTed on the 3/5-person task. As shown in Fig. 35, the embeddings across different layers of the fine-tuned

Llama3-8B provide more distinguishable signals for memorized samples than those of the base model.

(a) 3-person puzzles under leaf perturbation.

(b) 3-person puzzles under statement perturbation.

(c) 5-person puzzles under leaf perturbation.

(d) 5-person puzzles under statement perturbation.

Figure 35: Test AUC for predicting whether $N$-person puzzles can be consistently solved under perturbations by Direct-FTed Llama3-8B models.

