# OpenReview forum: "On Memorization of Large Language Models in Logical Reasoning"
_ICLR.cc/2025/Conference — Submitted to ICLR 2025_

### Official Review · Reviewer_ofks · 2024-10-17

**Soundness:** 3
**Presentation:** 4
**Contribution:** 2
**Rating:** 5
**Confidence:** 5

**Summary:**

This paper investigates how large language models (LLMs) solve *Knights and Knaves* (K&K) puzzles, aiming to determine whether models rely more on *memorizing similar problems* or on developing *genuine reasoning skills*. The authors introduce a "*memorization score*" metric and evaluate both pre-trained and fine-tuned LLMs, finding that while both models heavily depend on memorization, they also exhibit some reasoning capability despite this reliance. They also provide some experiments including probing inner representation and consistency-or-not indicators, which sheds light on the mechanism of LLM reasoning.

**Strengths:**

1. Their experiments are comprehensive and well-designed, covering both pre-trained models and finetuned models, various difficulty levels of K&K problems, and both open-source and state-of-the-art closed-source models. Particularly, the probing experiments (section 4.2), finetuning with incorrect answers (in section 4.3) and indicators (in section 6) offer valuable insights into to the reasoning mechanism of LLMs.

2. The use fo the K&K puzzle introduces a novel reasoning challenge for LLMs. This benchmark provides a fresh perspective on understanding the reasoning capabilities of these models.

**Weaknesses:**

1. A key question arises regarding **the choice of the K&K puzzles**. It is unclear whether this puzzle represents a practical and significant challenge or if it is simply another interesting problem, among many others like those in BigBench. While the need for automatically generated, answerable, and perturbable questions is acknowledged, the authors should further justify why K&K puzzles are essential to study. For example, demonstrating that many real-world or well-known problems can be translated into K&K puzzles would strengthen the argument.
2. The novelty of the proposed methodology, especially the memorization metric, is somewhat lacking. Perturbation-based scores have been widely used in work focusing on memorization and generalization. For instance, [1] utilizes the AdvGLUE [2] and ANLI [3] benchmarks to test ChatGPT's robustness, both involving word- or semantic-level perturbations. Similar approaches can also be found in works on LLM reasoning such as [4] and [5].
3. The paper does **not clearly distinguish** between *memorization* and *genuine reasoning*. The authors should explicitly define how they are cocepturalizing memorization vs. genuine reasoning in the context of their study. Concepts such as "case-based" or "rule-based" reasoning [6] could be a helpful framework for differentiating these cognitive modes. Additionally, since accuracy is a factor in the memorization score, an increase in this score might reflect either improved accuracy or reduced robustness, making it difficult to discern its true meaning. Please provide mroe discussion about how the memorization score accounts for the potential confound between accuracy and robustness.
4. Certain experiments **lack clear explanations**, and some conclusions appear questionable:

    1. In Section 3.1, the poor performance of off-the-shelf models on K&K tasks seems *unrelated* to the focus on memorization. This also raises concerns about whether K&K puzzles are suitable for testing reasoning-by-memorization, as reasoning ability may be a prerequisite for drawing meaningful conclusions.

    2. The claim in Section 4.1 that "*generalization performance increases with memorization level*" is debatable. Since accuracy is part of the memorization score, it is unsurprising that test accuracy correlates with memorization score on the training set. They could both be the results of increasing training accuracy. To demonstrate that memorization aids generalization, a better comparison would be between test accuracy and memorization score **with equal training accuracy**.

    3. The fine-tuned models were trained for 100 or 5 epochs, which may have led to overfitting (i.e., memorization). Given the large size of these models, even slight overtraining can inflate memorization scores, raising concerns about whether the results provide meaningful insights into practical reasoning capabilities.

    4. The conclusion in Section 4.1 that "*models likely learned to reason K&K puzzles to some extent*" is unclear. Figures 5 and 4 indicate that both memorization score and test accuracy are lower on test samples than on training samples, so is the lower memorization score simply a result of reduced accuracy?

    5. In Section 6, while the indicator experiments are interesting, I noticed that the labeling of "consistently solved" and "not consistently solved" is based on a single perturbation. Given the complexity of the K&K puzzle, the neighborhood of such a question could be quite large, potentially resulting in varied effects from different perturbations. It is possible that some perturbations lead to a sharp decrease in performance, while others may not. This raises the question of whether a binary labeling approach is truly robust and convincing. I suggest exploring whether this method accurately reflects the model's ability to generalize across a broader range of perturbations. For example, the authors can conduct a sensitivity analysis using multiple perturbations per puzzle to assess the robustness of their binary labeling approach.

5. Although the paper provides a deep dive into model behavior, it lacks a concrete conclusion on how LLMs solve these puzzles without chain-of-thought reasoning. Addressing this question, though difficult, would greatly enhance the paper. But I know it is a hard task and I will not detract from the article for the lack of this.

[1] On the Robustness of ChatGPT: An Adversarial and Out-of-distribution Perspective, Wang et. al., 2023. [https://arxiv.org/abs/2302.12095](https://arxiv.org/abs/2302.12095)

[2] Adversarial GLUE: A Multi-Task Benchmark for Robustness Evaluation of Language Models, Wang et. al., 2021. [https://arxiv.org/abs/2111.02840](https://arxiv.org/abs/2111.02840)

[3] Adversarial NLI: A New Benchmark for Natural Language Understanding, Nie et. al., 2019. [https://arxiv.org/abs/1910.14599](https://arxiv.org/abs/1910.14599)

[4] Can LLM Graph Reasoning Generalize beyond Pattern Memorization? Zhang et. al., 2024. [https://arxiv.org/abs/2406.15992](https://arxiv.org/abs/2406.15992)

[5] RUPBench: Benchmarking Reasoning Under Perturbations for Robustness Evaluation in Large Language Models, Wang & Zhao, 2024. [https://arxiv.org/abs/2406.11020](https://arxiv.org/abs/2406.11020)

[6] Case-Based or Rule-Based: How Do Transformers Do the Math? Hu et. al. 2024. [https://arxiv.org/abs/2402.17709](https://arxiv.org/abs/2402.17709)

**Questions:**

1. What is the necessity of focusing on K&K problems (as discussed in con 1)?
2. More explanation is needed to clarify the relationship between the experimental results and the conclusions.
3. A clearer definition of memorization and its distinction from genuine reasoning is needed. Is memorization simply the opposite of generalization or rule-based reasoning? Additionally, the current metric, which multiplies accuracy by $(1 - cr)$, could be refined, as the latter provides more insight into memorization or generalization. The "not solving" behavior should be excluded by setting a threshold (e.g., accuracy > 0.8) to filter out irrelevant cases.
4. Minor suggestions:
    1. The memorization metric, introduced in Figure 1 (Section 1), needs more explanation to help readers understand it. Adding details to the figure's caption or in the text in Section 1 would be helpful.
    2. The "reasoner" is described as sequentially examining each individual and checking for contradictions. However, humans likely employ heuristics to determine the order of examination and may use shortcut strategies. The paper could discuss whether this approach is the best analogy to human reasoning.

---

> ### Author Response · Authors · 2024-11-24
> **Response to Reviewer ofks (Part 1)**
>
> Thank you for your detailed feedback! We address your questions and comments below.
>
> > Q1. A key question arises regarding the choice of the K&K puzzles. It is unclear whether this puzzle represents a practical and significant challenge or if it is simply another interesting problem, among many others like those in BigBench. While the need for automatically generated, answerable, and perturbable questions is acknowledged, the authors should further justify why K&K puzzles are essential to study. For example, demonstrating that many real-world or well-known problems can be translated into K&K puzzles would strengthen the argument.
>
> Thanks for the question. Indeed many real-world or well-known problems can be translated into K&K puzzles as the principle underlying K&K is the [**Boolean satisfiability (SAT) problem**](https://en.wikipedia.org/wiki/Boolean_satisfiability_problem). K&K puzzles are a simple way to express  SAT instances in natural language.  SAT is a fundamental problem, at the core of computer science; e.g., *it was the first problem that was proven to be NP-complete*.   SAT can be reduced to most natural reasoning problems and hence the performance of a model on SAT (i.e., K&K puzzles) can be indicative of its reasoning capabilities.
>
> Specifically, consider a K&K puzzle involving $N$ people, a possible solution assigns a Boolean value to $N$ variables $B_1,B_2,\ldots,B_N$, where the truth value of $B_i$ indicates whether the $i$th person is telling the truth. By definition, the $i$th person is telling the truth if and only if their statement $S_i$ is true. Therefore, a valid solution to a \kk puzzle is a Boolean assignment for $B_1,B_2,\ldots,B_N$ such that the following formula evaluates to true.
> $$(B_1\Leftrightarrow S_1)\wedge(B_2\Leftrightarrow S_2)\wedge\cdots\wedge(B_N\Leftrightarrow S_N).$$
> Here the statement $S_i$​ involves the five most commonly used propositional connectives: and, or, not, imply, and equivalence [1].  This K&K formulation provides a direct link to the SAT problem.
>
> We also want to highlight that many real-world problems can be translated into SAT problems, for instance:
> - *Constraint Satisfaction Problem (CSP)*: Some existing popular LLM logical reasoning benchmarks including ZebraLogic [6] and Einstein’s Puzzle [7] are CSP, which involves assigning values to variables under certain constraints. Any CSP can be reduced to an SAT problem by converting the variables and constraints into propositional variables and formulas [5].
> - *Hardware and Software Verification*: SAT solvers are widely used in model checking to verify that hardware circuits or software programs adhere to specified behavior, identifying logical errors in designs [2].
> -*Cryptography*: SAT solvers model cryptographic functions to detect vulnerabilities  [3].
> - *Mathematical Theorem Proving*: SAT solvers assist in computer-aided proofs, such as discovering unknown Van der Waerden numbers or solving the Boolean Pythagorean triples problem [4]
>
> These connections illustrate that K&K puzzles, as natural language SAT instances,  are essential to study as logical reasoning tasks. We have added the above discussion to Section 2.2 of our revised PDF.
>
>
>
> Reference:
> - [1] Mendelson, E. (2015). Introduction to Mathematical Logic (6th ed.). CRC Press.
> - [2] Biere, Armin; Cimatti, Alessandro; Clarke, Edmund M.; Strichman, Ofer; Zhu, Yunshan (2003). "Bounded Model Checking". Advances in Computers. 58 (2003)
> - [3] Massacci, F., & Marraro, L. (2000). Logical cryptanalysis as a SAT problem. Journal of Automated Reasoning, 24(1-2), 165–203.
> - [4] https://en.wikipedia.org/wiki/SAT_solver
> - [5] Stuart Russell and Peter Norvig, "Artificial Intelligence: A Modern Approach," 3rd Edition, Prentice Hall, 2010.
> - [6] Bill Yuchen Lin, Ronan Le Bras, and Yejin Choi. ZebraLogic: benchmarking the logical reasoning ability of language models, 2024. URL https://hf.co/spaces/allenai/ZebraLogic.
> - [7] Nouha Dziri, Ximing Lu, Melanie Sclar, Xiang Lorraine Li, Liwei Jian, Bill Yuchen Lin, Peter West, Chandra Bhagavatula, Ronan Le Bras, Jena D. Hwang, Soumya Sanyal, Sean Welleck, Xiang Ren, Allyson Ettinger, Zaïd Harchaoui, and Yejin Choi. Faith and fate: Limits of transformers on compositionality. NeurIPS, 2024.

---

> > ### Comment · Reviewer_ofks · 2024-11-26
> > **Further Questions**
> >
> > 1. **Clarification on the Importance of K&K Problems**
> > The discussion about SAT is helpful, but it's important to note that K&K is only a very limited subset of SAT problems. Currently, your discussion feels somewhat indirect, as it suggests that "K&K is a subset of SAT" and "CSP and verification are also subsets of SAT." I would appreciate a more direct illustration of the importance of the K&K problem. Could you provide references that support the significance of K&K? For example, are there papers that use K&K as a testbed, explaining why it was chosen? References from other fields, such as cognitive science or philosophy, might also be relevant. Additionally, I noticed that you mention ZebraLogic and Einstein's Puzzle in response to review 7GXF, suggesting that similar SAT problems have been proposed before. Could you elaborate on the differences between K&K and these problems? Why is it necessary to study K&K specifically rather than these alternatives?
> >
> > 2. **Clarification of Perturbation Method and Innovation**
> > Thank you for your detailed explanation of your proposed perturbation method. I understand that your perturbations serve as a measure of robustness. However, the key difference between your work and previous studies lies in the definition of robustness. (Please correct me if I'm wrong.) For example, prior work focused on robustness at the linguistic level, while you focus on 'mathematical-level' robustness. However, as mentioned in [1], both advGLUE and ANLI include deeper, manually annotated attacks, not just word- or sentence-level perturbations. Though not identical, this can be compared to your "mathematical-level" perturbations. That said, I still feel that the discussion of innovation in this regard is insufficient. If you haven't emphasized this aspect of your innovation, I believe it’s acceptable, but it may benefit from further clarification.
> >
> > 3. **Confusion Between 'Memorization & Genuine Reasoning' vs. 'Fitting & Generalization'**
> > It seems that the definition of 'genuine reasoning' here is essentially what is typically understood as generalization, as you mention performance on "*novel unseen test data differing from training data*". Therefore, I still don’t understand the distinction between your terms 'memorization & genuine reasoning' and the conventional terms 'fitting & generalization.' Furthermore, while you state that '*memorization and generalization are not opposites*,' the term 1-CR in the score you define seems to reflect a negative consideration of generalization. When the model’s generalization performance is high, CR will be high, and thus LiMem will be low. Therefore, in your definition, memorization appears to be opposite to generalization.
> >
> >     I suggest that the authors carefully reconsider this issue: when you define memorization, what is its opposite? I believe multiplying the two factors leads to a vague definition. As you've stated, the opposite of memorization is a mix of two factors: either poor learning or reliance on reasoning. However, these two factors have distinct natures—one is what we want, the other is what we don’t. Mixing them creates ambiguity, making it difficult to interpret low-score results. Once you try to explain trends in LiMem, we need additional tools to understand the meaning behind these changes. This diminishes the significance of LiMem as proposed. I believe these factors should be separated for clarity.
> >
> > 4. **Reorganization of Experiments Based on CR**
> > I believe Figure 19 is clearer than Figure 5, and I suggest that you reorganize your experiments using only CR. However, the results in Figure 19 seem less significant than those in Figure 5, which deepens my concern. The confusion between the two factors, 'robustness' and 'performance,' has unnecessarily exaggerated the surprising nature of the results. I believe robustness is the true point of concern when discussing 'memorization,' as we typically don’t focus on whether a model relies on memorization until it has demonstrated some performance. If the model’s higher LiMem on the training set is due to improved performance, rather than significantly lower CR, then there is nothing particularly surprising about this conclusion.
> >
> > 5. **Clarification of Response to Q5**
> > The answer to Q5 is well-explained, but I believe these two points are more suitable for the main text. I suggest you include a paragraph in the main body to explain these points, rather than concluding with "off-the-shelf models do not perform well on K&K problems."

---

> > > ### Comment · Reviewer_ofks · 2024-11-26
> > > **Further questions (Part 2)**
> > >
> > > 6. **Further Clarification on Overfitting in Llama-8B Training**
> > > The answer to Q7 has addressed my concern about overfitting, but I still wonder why training a Llama-8B model for 100 epochs resulted in continued increases in test accuracy. I am surprised by this, as when I train LLMs, 100 epochs usually lead to severe overfitting. Could you provide more details about your training process, such as the number of training samples, learning rate, learning rate schedule, or any other settings you believe were crucial in avoiding overfitting? Additionally, could you provide the learning curve of the model?
> > >
> > > **Overall Comments**:
> > >
> > > I highly appreciate the authors' efforts, and several of my concerns have been addressed. However, the current paper still heavily relies on metrics that are not clearly explained, and the explanation of the motivation for studying the K&K problem remains indirect. As such, I will maintain my current rating and look forward to the authors providing further clarifications on these issues.
> > >
> > > Additionally, as a personal suggestion, I believe reorganizing the experiments and discussions based on separate metrics could significantly improve the paper’s credibility and readability. I am concerned that presenting only part of the results in the appendix will not only lengthen the paper unnecessarily but also force the authors to make strained explanations in order to reconcile with previous conclusions.

---

> > > > ### Author Response · Authors · 2024-11-28
> > > > **Response to Reviewer ofks (Part 1)**
> > > >
> > > > Thank you again for your detailed feedback! We address your questions and comments below.
> > > >
> > > >
> > > >
> > > > > Q1: Clarification on the Importance of K&K Problems and differences between K&K and exisiting logical puzzle datasets
> > > >
> > > > The Knights and Knaves (K&K) problem (i.e., Liar and Truthteller puzzle) is significant in cognitive science, philosophy, math and AI, serving as a key testbed for logical reasoning and problem-solving. It was originally introduced In the book "What is the Name of This Book?" back in 1978 [1] and more versions were available in a later book in 1990 [2]
> > > > Several studies and papers illustrate its importance:
> > > > 1. **The Hardest Logic Puzzle Ever**.  The Hardest Logic Puzzle Ever is a logic puzzle so called by American philosopher and logician George Boolos and published in The Harvard Review of Philosophy in 1996. The puzzle is based on Knights and Knaves puzzles [3]
> > > > 2. **Cognitive Science & Philosophy**: K&K puzzles are discussed in many publications in  Cognitive Science and Philosophy areas to explore paradoxical reasoning with liars and truth-tellers and how such puzzles provide insights into human reasoning and logic [4,5,6,7,8].
> > > > 3. **AI and Logic Reasoning**: K&K puzzles are discussed in AI Magazine in 2017 as a challenging task, which highlights that “even studying and solving only single parts of the proposed challenge would represent an important step forward for artificial intelligence.” [9]
> > > > 4.  **Mathematics and programming**:  K&K puzzles are translated into Boolean programming (0-1 programming) problem to study graph representation [10], and explored in Mathematics for paradoxical self-reference [11].
> > > >
> > > >
> > > >
> > > >
> > > > While Einstein’s Puzzle [12] and ZebraLogic [13], construct puzzles to analyze the reasoning abilities LLMs, they differ significantly in their design and focus. These benchmarks are based on scenarios where, given a list of clues/constraints, one must deduce a unique and correct assignment of values for the rest of the variables, assuming that all clues are accurate.
> > > >
> > > > In contrast, K&K is a deductive reasoning task that evaluates LLMs' ability to infer both the truthfulness of statements and their logical implications (e.g., identifying whether a character is a truth-teller or a liar). Unlike traditional benchmarks, K&K does not provide explicit clues. Instead, it demands suppositional reasoning, which relies heavily on conjecture, inference, and paradoxical reasoning, rather than on direct or sufficient evidence/constraint. This makes K&K fundamentally distinct and more challenging.
> > > >
> > > >
> > > > Reference:
> > > >
> > > > - [1] Raymond M. Smullyan. 1978. What is the Name of This Book?: The Riddle of Dracula and Other Logical Puzzles. Prentice-Hall, Englewood Cliffs, N.J.
> > > > - [2] P.N. Johnson-Laird and Ruth M.J. Byrne. 1990. Metalogical problems: Knights, knaves, and rips. Cognition, 36(1):69–84
> > > > - [3] https://en.wikipedia.org/wiki/The_Hardest_Logic_Puzzle_Ever
> > > > - [4] Paralogical reasoning: Evans, Johnson-Laird, and Byrne on liar and truth-teller puzzles. *Cognition*, Volume 36, Issue 3, September 1990, Pages 291-314. https://doi.org/10.1016/0010-0277(90)90061-N
> > > > - [5]  Reasoning with knights and knaves: A discussion of Rips. *Cognition*, Volume 36, Issue 1, July 1990, Pages 85-90. -https://doi.org/10.1016/0010-0277(90)90055-O
> > > > - [6] A general method of solving Smullyan's puzzles. *Logic and Logical Philosophy*, Volume 4 (1996), 97–103. https://www.marianotomatis.it/blog/materiale/kolany.pdf
> > > > - [7]  Sorting the Liars from the Truth Tellers: The Benefits of Asking Unanticipated Questions on Lie Detection. *Applied Cognitive Psychology*, Appl. Cognit. Psychol., 27: 107-114. [https://doi.org/10.1002/acp.2879](https://doi.org/10.1002/acp.2879)
> > > > - [8]  Reasoning About Agent Types and the Hardest Logic Puzzle Ever. *Minds & Machines* 23, 123–161 (2013). https://doi.org/10.1007/s11023-012-9287-x
> > > > - [9] Solving Mathematical Puzzles: A Challenging Competition for AI. *AI Magazine*, 38(3), 83-96. https://ojs.aaai.org/aimagazine/index.php/aimagazine/article/view/2736
> > > > - [10] Boolean programming, truth-teller-liar puzzles and related graphs. *ITI* 2003., Cavtat, Croatia, 2003, pp. 663-668, https://ieeexplore.ieee.org/abstract/document/1225419
> > > > - [11] Truth-Teller–Liar Puzzles with Self-Reference. *Mathematics* 2020, 8(2), 190. https://www.mdpi.com/2227-7390/8/2/190
> > > > - [12] Nouha Dziri, Ximing Lu, Melanie Sclar, Xiang Lorraine Li, Liwei Jian, Bill Yuchen Lin, Peter West, Chandra Bhagavatula, Ronan Le Bras, Jena D. Hwang, Soumya Sanyal, Sean Welleck, Xiang Ren, Allyson Ettinger, Zaïd Harchaoui, and Yejin Choi. Faith and fate: Limits of transformers on compositionality. NeurIPS, 2024.
> > > > - [13] Bill Yuchen Lin, Ronan Le Bras, and Yejin Choi. ZebraLogic: benchmarking the logical reasoning ability of language models, 2024. URL https://hf.co/spaces/allenai/ZebraLogic.

---

> ### Author Response · Authors · 2024-11-24
> **Response to Reviewer ofks (Part 2)**
>
> > Q2: The novelty of the proposed methodology, especially the memorization metric, is somewhat lacking. Perturbation-based scores have been widely used in work focusing on memorization and generalization. For instance, [1] utilizes the AdvGLUE [2] and ANLI [3] benchmarks to test ChatGPT's robustness, both involving word- or semantic-level perturbations. Similar approaches can also be found in works on LLM reasoning such as [4] and [5].
>
> Thanks for the comment. We acknowledge that adding perturbation is a general principle for evaluating model robustness. However, we would like to highlight the following distinctions between our work and prior studies [1-5]:
> - **Novel Data Generation Framework for Mathematical Perturbations**:
> While prior works such as [4] focus predominantly on language-level perturbations (e.g., lexical, syntactic, and semantic changes), our methodology introduces a framework that can generate *mathematical-level perturbations*, particularly for evaluating model reasoning capability. These perturbations are not only systematically controllable in terms of type and difficulty but also target the underlying mathematical structure of problems. This is also more challenging than language-level perturbation because the perturbed problem need to be solved to find the **new groundtruth answer** for evaluation. This allows us to evaluate model generalization to new math problems in a way that is beyond the scope of purely linguistic perturbations.
> - **Emphasis on Natural Perturbations**: Unlike *adversarial* approaches such as typos and distractions [1, 2] or human-annotated examples specifically designed to exploit model weaknesses [3] that target worst-case test scenarios, our framework focuses on *natural robustness*. The perturbations we generate—across both language and mathematics—reflect scenarios commonly encountered in natural K&K QA contexts.
> - **Applicability**: While [5] evaluates graph reasoning through perturbations in graph patterns (e.g., training on connectivity queries but testing on shortest-path queries), our focus is on boolean satisfiability problems (SAT) using K&K tasks, potentially provide a broader context for assessing logical reasoning capabilities in LLMs.
>
> The results in Figure 5 show that mathematical-level perturbations induce significantly larger performance declines compared to language-level perturbations. This suggests that while models may adapt to superficial linguistic variations, they struggle to address deeper math structure changes inherent in logical reasoning tasks. This observation underscores memorization over genuine reasoning,  a key insight of our study.
>
>
> > Q3:   The paper does not clearly distinguish between memorization and genuine reasoning. The authors should explicitly define how they are conceptualizing memorization vs. genuine reasoning in the context of their study. Concepts such as "case-based" or "rule-based" reasoning [6] could be a helpful framework for differentiating these cognitive modes…..  A clearer definition of memorization and its distinction from genuine reasoning is needed. Is memorization simply the opposite of generalization or rule-based reasoning?
>
> Thanks for the valuable comment.  Memorization and generalization are not direct opposites, as they target different aspects of model performance.  As shown in Section 4, a model that memorizes training data can still generalize well to both in-distribution and out-of-distribution test data.
>
> While memorization can be evaluated on **training data** (in our work, using *small, locally perturbed* training samples),  defining and measuring “*genuine reasoning*” is more difficult as it depends on **novel unseen test data** differing from training data.
> We empirically capture the genuine reasoning from several perspectives:
> - Evaluating perturbed testing data
> - Evaluating in-distribution testing data with the same K&K difficulty level.
> - Evaluating out-of-distribution testing data with the different K&K difficulty levels.
> - Probing model's internal representations with a different QA task — a dataset consisting of correct/incorrect K&K statements.
>
> For example, in our transferability study, the model demonstrates capabilities beyond "case-based reasoning" by solving puzzles with unseen difficulty levels. However, we do not claim this constitutes "rule-based reasoning," as 100% transferability accuracy is not achieved. Based on these findings, we hypothesize that the model may develop useful internal computational graphs (e.g., dependencies among different characters in K&K) that aid transferability, while also relying on structured shortcuts that remain partially inaccurate.

---

> ### Author Response · Authors · 2024-11-24
> **Response to Reviewer ofks (Part 3)**
>
> > Q4:  Since accuracy is a factor in the memorization score, an increase in this score might reflect either improved accuracy or reduced robustness, making it difficult to discern its true meaning. Please provide mroe discussion about how the memorization score accounts for the potential confound between accuracy and robustness…… The current metric, which multiplies accuracy by (1−cr), could be refined, as the latter provides more insight into memorization or generalization. The "not solving" behavior should be excluded by setting a threshold (e.g., accuracy > 0.8) to filter out irrelevant cases.
>
> Thanks for the insightful comment.  Regarding the relationship between the memorization score LiMem, accuracy, and robustness,  assume the accuracy increases but the consistency ratio remains the same. Because it is a ratio (that remains the same), an increased accuracy means the (absolute) number of inconsistent samples under perturbation is now increased. As a result, this still reflects more memorization.
>
> We design the two terms based on intuition from judging human behaviors. We combine them into a single metric to make it easier to consume.   As clarified in L140–147: A high LiMem score indicates memorization. However, a low LiMem score could indicate either solving by reasoning or not solving (random guessing), which requires a separate check on the accuracy and robustness terms to differentiate between the two cases.
>
> Following the reviewer’s suggestion, we now report the **consistency ratio** (i.e., robustness) in Appendix Figure 19. Fine-tuned LLMs generally demonstrate a higher consistency ratio (i.e., more robust) on solved problems in the test set compared to the train set, particularly for challenging tasks such as 5/8-person puzzles. On the 3-person puzzle task, the consistency ratio between the train and test sets remains comparable. The consistency ratio generally is higher in easy tasks than in hard tasks.
>
> We thank the reviewer for suggesting setting a threshold, and we will add it to our revision.
>
>
> > Q5: In Section 3.1, the poor performance of off-the-shelf models on K&K tasks seems unrelated to the focus on memorization. This also raises concerns about whether K&K puzzles are suitable for testing reasoning-by-memorization, as reasoning ability may be a prerequisite for drawing meaningful conclusions.
>
> We agree with the reviewer that to achieve a high memorization score, the model has to demonstrate sufficient accuracy on the original problems in the first place, underscoring that high accuracy is a prerequisite to studying meaningful memorization.
>
> Our evaluation of off-the-shelf models on K&K is still related to our focus on memorization given the different performance under easy/hard K&K tasks :
> - **Potential memorization on easy K&K tasks**: We observe relatively high accuracy on easy tasks along with non-trivial memorization scores. For instance, Claude 3.5-sonnet achieves ACC = 0.63 and a memorization score LiMem = 0.33 on 3-person puzzles. Regarding the sources of memorization, we clarify that while our puzzles are randomly generated (e.g., using random language expressions and mathematical structures), Knights and Knaves (K&K) is a well-known classical puzzle type. Existing instances of such puzzles and related materials are available online [1,2], and it is possible that they were included in the pretraining data of these models, which is unknown to us.  In addition, we investigate the open-source pretraining datasets, by utilizing the [WIMBD tool](https://wimbd.apps.allenai.org/) to analyze the occurrence of popular names (“Alice”, “Bob”) combined with different roles (e.g., "knight," "knave," etc.). The below table revealed non-trivial occurrences, suggesting that materials resembling K&K puzzles may have been part of the pretraining data.
> - **Poor performance on hard K&K tasks validates our dataset for studying memorization via fine-tuning**: The hard tasks in our dataset present a significant challenge for even most advanced models (e.g., GPT-4o, Claude 3.5-sonnet, Gemini-1.5-Pro), with accuracy dropping below 11% on 8-person puzzles. This indicates that the *harder portions of our benchmark (e.g., long and complex puzzles) are likely contamination-free, justifying their use in fine-tuning experiments* described in Section 3.2. The challenging nature of these tasks provides a *clean and controlled* setting to study memorization via explicit fine-tuning.
>
> | Statement | Dolma | The PILE | C4 | Oscar | OpenWebText |
> |---|---|---|---|---|---|
> | "Alice is a knave" | 13 | 6 | 2 | 1 | 0 |
> | "Alice is a knight" | 23 | 8 | 6 | 1 | 0 |
> | "Bob is a knave" | 11 | 8 | 0 | 1 | 0 |
> | "Bob is a knight" | 53 | 9 | 22 | 5 | 0 |
> | "Charlie is a knave" | 3 | 0 | 0 | 0 | 0 |
> | "Charlie is a knight" | 10 | 1 | 2 | 0 | 0 |
>
> Reference:
> - [1] https://philosophy.hku.hk/think/logic/knights.php
> - [2] https://dmackinnon1.github.io/knaves/

---

> ### Author Response · Authors · 2024-11-24
> **Response to Reviewer ofks (Part 4)**
>
> > Q6: The claim in Section 4.1 that "generalization performance increases with memorization level" is debatable. Since accuracy is part of the memorization score, it is unsurprising that test accuracy correlates with memorization score on the training set. They could both be the results of increasing training accuracy. To demonstrate that memorization aids generalization, a better comparison would be between test accuracy and memorization score with equal training accuracy.
>
> Thanks for the suggestion. We acknowledge that achieving equal training and testing accuracy is challenging due to the complexities of training dynamics. However, to address this concern, we have separately reported the consistency ratio in response to Q4, which we hope provides additional clarity.
>
> > Q7: The fine-tuned models were trained for 100 or 5 epochs, which may have led to overfitting (i.e., memorization). Given the large size of these models, even slight overtraining can inflate memorization scores, raising concerns about whether the results provide meaningful insights into practical reasoning capabilities.
>
> Thanks for the comment. We clarify that the chosen # epoch is meaningful due to following reasons:
> - *Deliberate Induction of Memorization for Controlled Study*: Training the models for a large # epochs was a deliberate decision to induce memorization within a controlled setup based on K&K dataset. While the exact number of epochs or data used for pretraining off-the-shelf models is unavailable, existing literature reveals that common reasoning benchmarks are often contaminated (memorized). This motivates our controlled experimental design, where we simulate a similar memorization-prone environment using proposed K&K dataset. By inducing memorization with a large # epochs, we aim to systematically study how such conditions affect reasoning performance, thereby gaining insights into the interplay between memorization and reasoning.
> - *# Epoch*: We use 5 epochs for GPT4o-mini.  While 100 is the maximal # epoch for fine-tuning Llama3-8B, we report the memorization score at epoch 50 throughout the paper (as mentioned in Appendix D.2.2.),  as this is the point at which the models typically converge as shown in Figure 19. We now clarified this in the main paper in our revised PDF.
> - *Train/Test Accuracy Trends*: As observed in Figure 4, both train and test accuracies for Llama3-8B (up to 50 epochs) and gpt4omini (up to 5 epochs) continue to increase, and not yet achieve 100% accuracy for challenging KK tasks like 5-ppl puzzles and 8-ppl puzzles,  indicating that the models had not entirely achieved overfitting at these points. This supports the practicality of the chosen # epoch.
>
> > Q8:   The conclusion in Section 4.1 that "models likely learned to reason K&K puzzles to some extent" is unclear. Figures 5 and 4 indicate that both memorization score and test accuracy are lower on test samples than on training samples, so is the lower memorization score simply a result of reduced accuracy?
>
> Thanks for the comment. We acknowledge that indeed a low memorization score LiMem alone is not enough to conclude that models learn to reason K&K puzzles as it could indicate either reasoning or random guessing ( L140–147 ). We conducted further analysis to study the models’s reasoning ability, including generalization across different #ppl puzzles, and the probing tests. We fixed the statement in Section 4.1  (L334-338) to reflect this point.
>
>
> > Q9 In Section 6, while the indicator experiments are interesting, I noticed that the labeling of "consistently solved" and "not consistently solved" is based on a single perturbation….I suggest exploring whether this method accurately reflects the model's ability to generalize across a broader range of perturbations.
>
> Thanks for the comment. We report results under statement perturbation and leaf perturbation in Appendix Figure 33.

---

> ### Author Response · Authors · 2024-11-24
> **Response to Reviewer ofks (Part 5)**
>
> > Q10.  Although the paper provides a deep dive into model behavior, it lacks a concrete conclusion on how LLMs solve these puzzles without chain-of-thought reasoning. Addressing this question, though difficult, would greatly enhance the paper. But I know it is a hard task and I will not detract from the article for the lack of this.
>
> Thank you for the thoughtful comment. We hypothesize that the model may develop implicit internal computational graphs among the $N$ characters in the $N$-ppl K&K task, which enables it to solve these puzzles without explicitly outputting a chain-of-thought reasoning process. While we acknowledge the challenge of verifying this hypothesis directly, we consider it an important direction for future work. A similar hypothesis is explored in [1,  Section 4.2], where the authors demonstrate that models acquire implicit skills, such as learning all-pair dependencies, after pretraining on grade-school math problems.
>
> [1] Tian Ye, Zicheng Xu, Yuanzhi Li, and Zeyuan Allen-Zhu. Physics of language models: Part 2.1, grade-school math and the hidden reasoning process. arXiv preprint arXiv:2407.20311, 2024
>
> > Q11. The memorization metric, introduced in Figure 1 (Section 1), needs more explanation to help readers understand it. Adding details to the figure's caption or in the text in Section 1 would be helpful.
>
> Thanks for the suggestions. We’ve add more explanation for the metric to Figure 1’s cations and Section 1 L70-72 in revised PDF.
>
>
> > Q 12. The "reasoner" is described as sequentially examining each individual and checking for contradictions. However, humans likely employ heuristics to determine the order of examination and may use shortcut strategies. The paper could discuss whether this approach is the best analogy to human reasoning.
>
> Thanks for the valuable comment. We indeed optimized the order of examination for our reasoner design exactly as the reviewer
> suggested. The details are in Appendix  C.3.
>
> Specifically, we note that when explaining the reasoning steps for K&K puzzles, human or off-the-shelf LLMs rarely use the brute-force assignment search approach adopted in our Solver. Instead, they tend to examine the statement from each person sequentially, construct a *partial* assignment for the people examined so far, and backtrack when a contradiction is found. We design our Reasoner following the same procedure. We maintain a queue of people to be examined next, and a partial assignment of knight/knave for people that have been examined so far. More details can be found in Appendix  C.3.

---

> ### Author Response · Authors · 2024-11-28
> **Response to Reviewer ofks (Part 2)**
>
> > Q2 prior work focused on robustness at the linguistic level, while you focus on 'mathematical-level' robustness. However, as mentioned in [1], both advGLUE and ANLI include deeper, manually annotated attacks, not just word- or sentence-level perturbations. Though not identical, this can be compared to your "mathematical-level" perturbations. That said, I still feel that the discussion of innovation in this regard is insufficient. If you haven't emphasized this aspect of your innovation, I believe it’s acceptable, but it may benefit from further clarification.
>
>
> Thanks for the comment.  We argue that language-level perturbations, even when involving manually annotated attacks as in advGLUE and ANLI ([1]), fundamentally differ from the mathematical-level perturbations introduced in our study.
> These approaches are not directly comparable due to critical differences in purpose, methodology, and scope.
>
> In [1], advGLUE and ANLI target natural language understanding tasks, such as sentiment analysis and textual entailment. The primary goal in these cases is to craft adversarial perturbations that lead to misclassification without altering its ground truth.
>
> In contrast, our perturbation method operates at a mathematical level in logical reasoning tasks and modifies **both the problem and the ground-truth answer**. These mathematical perturbations ensure that the perturbed puzzle has a *distinctly different solution* compared to the original puzzle, while remaining superficially similar and maintaining a comparable difficulty level. This is guaranteed by the Perturber, Reasoner, and Solver components (lines 186–192 ). This approach provides a direct evaluation of the models’ understanding of the underlying mathematical principles.
>
> Thus, our method serves a different purpose and employs a fundamentally different framework compared to the adversarial strategies studied in [1], which focus on classification tasks. By addressing logical reasoning robustness through mathematical-level perturbations, our work contributes a novel perspective under the score of reasoning, distinct from advGLUE and ANLI.
>
>
>
> > Q3-4:  Acc and CR should be separated for clarity. Reorganization of Experiments Based on CR
>
>
> Thank you for the comment. In the revised PDF, we will update the figures. We will update Figure 5 into a scatter plot as shown in [this anonymous link](https://ibb.co/GWd6503) where the x-axis represents clean accuracy, the y-axis represents the inconsistency ratio (1 - CR) under local perturbations, and the color spectrum corresponds to the memorization score.
> Our analysis shows that fine-tuned LLMs generally achieve higher clean accuracy (x-axis) but exhibit greater inconsistency under perturbations (y-axis) on the training set compared to the test set. This behavior is associated with a higher memorization score (indicated by color spectrum).
>
> Please let us know if this revision addresses the reviewer’s concerns.
>
>
>
> > Q5: Clarification of Response to Q5
>
>
> Thanks for the suggestion. We will add the discussion in lines 230 - 244 accordingly as shown in [this anonymous link](https://ibb.co/BfHQpbP).
>
> > Q6: Could you provide more details about your training process, such as the number of training samples, learning rate, learning rate schedule, or any other settings you believe were crucial in avoiding overfitting? Additionally, could you provide the learning curve of the model?
>
> Thanks for the comment.
>
> Training Details:
> - We fine-tune the models for each N-people task separately, with ntrain = 1000 for 3 ≤ N ≤ 8, and ntrain = 200 for 2-people task due to the limited number of combinations, as indicated in Line 255
> - Llama3-8B fine-tuning details are presented in D.2.2: we used LoRA fine-tuning with the fol-
> lowing hyperparameters: a batch size of 4, gradient accumulation steps of 8, and 5e-5
> learning rate. The LoRA configuration was set as follows: rank r = 32, scaling factor $\alpha$ = 32,
> and dropout rate 0.05. We relied on the [Supervised Fine-tuning Trainer](https://huggingface.co/docs/trl/en/sft_trainer) from the `trl` library without employing additional mechanisms to mitigate overfitting.
>
> Learning Curves:
> - We report the training and testing accuracy over epochs in Appendix figure 21 (also in this [anonymous link](https://ibb.co/D562bBb))
> - We report the training loss in [this anonymous link](https://ibb.co/hgcM8px)
>
> We think it might be because the challenging nature of the K&K logical reasoning tasks inherently demands more epochs for convergence.
>
> Please also let us know if there are other questions, and we look forward to the discussion with the reviewers to further improve our paper. Thank you!

---

> > ### Comment · Reviewer_ofks · 2024-11-29
> > **Thank you for your further explanation!**
> >
> > 1. The current introduction to the K&K problem effectively conveys its importance. However, I believe reference [9] should be incorporated into this section. In my view, these references would provide valuable context for readers, as they demonstrate the connection between the problem and human cognitive processes, which aligns with the goals of large language models (LLMs). In summary, I find the current explanation of the K&K problem's importance sufficient, but I recommend integrating the additional references into either the introduction or the related works section in the next revision.
> > 2. Figure 5 has shown significant improvement. However, the difference in inconsistency ratios seems marginal. Is the main conclusion that "fine-tuning improves accuracy but maintains the same inconsistency ratio"? If this is the case, it may not be a particularly surprising conclusion.
> > 3. As a minor suggestion, I feel that using 100 (or 50) epochs with only a few thousand training samples may not be the common case. In more practical cases, the training samples are more but less training epoches are used. Could you consider running the experiments with more samples but fewer epochs in the next version to see if the conclusion is the same?
> >
> > Now, I believe it is a stronger paper with all of your efforts. I think the new content makes the paper better and I have raised my score, however, I am still concerned about some of the potentially confusing conclusions (especially about LiMem) that prevent me from further improving the score. I seriously suggest that the authors consider removing that content, and separating the analysis from the perspectives of "performance" and "consistency" will not affect the contribution and conclusions of the paper.

---

> > > ### Author Response · Authors · 2024-12-02
> > > **Response to ofks**
> > >
> > > Thank you for your valuable comments!
> > >
> > > 1. We will incorporate the suggested references into the introduction section in our revision.
> > > 2. Regarding the revised Figure 5, all evaluations are conducted on the fine-tuned models (i.e., no comparison between pre- and post-fine-tuning). The figure highlights several key conclusions:
> > > - The inconsistency ratio on the training set is generally higher than on the test set, indicating greater memorization.
> > > - Inconsistency under math-level perturbations is higher than under language-level perturbations.
> > > - Harder tasks (e.g., the 8-person puzzle) exhibit higher inconsistency compared to easier tasks (e.g., the 3-person puzzle).
> > > 3. We will run additional experiments with more samples but fewer epochs, as per your suggestion.
> > >
> > > Additionally, we will update Figures 3, 5, and 6 to scatter plots, separating "accuracy" (x-axis) and "inconsistency under perturbations" (y-axis), consistent with the style of the [revised Figure 5](https://ibb.co/GWd6503).
> > >
> > > Thank you again for your feedback!

---

### Official Review · Reviewer_7GXF · 2024-10-27

**Soundness:** 3
**Presentation:** 3
**Contribution:** 2
**Rating:** 6
**Confidence:** 3

**Summary:**

This paper explores how large language models (LLMs) balance memorization and reasoning in solving logical tasks. Authors propose a benchmark Knights and Knaves (K&K) puzzles, the authors investigate whether LLMs rely on memorizing training examples or develop genuine reasoning skills. While fine-tuning models improved performance on known puzzles, their accuracy dropped with slight modifications, indicating reliance on memorization. However, fine-tuning also enhanced generalization, suggesting a mix of memorization and reasoning. The study introduces the Local Inconsistency-based Memorization Score (LiMem) to quantify memorization in these tasks.

**Strengths:**

1. Novel Quantification of Memorization: The paper introduces a new metric, the Local Inconsistency-based Memorization Score (LiMem), which provides a structured way to measure the extent of memorization versus reasoning in large language models (LLMs), a valuable contribution to understanding model behavior.

2. Thorough Empirical Analysis: The paper conducts an in-depth evaluation of models under various conditions, including fine-tuning, perturbation tests, and cross-difficulty transferability. This comprehensive approach offers significant insights into the models’ generalization and reasoning capabilities.

3. Insight into Model Generalization: The findings on how fine-tuning impacts both memorization and generalization offer valuable insights for future research on improving reasoning in LLMs.

**Weaknesses:**

1. Limited Task Scope: The paper focuses solely on logical reasoning, particularly the Knights and Knaves puzzles. While this allows for deep analysis, it limits the generalizability of the conclusions. Experiments on other reasoning domains, such as mathematical reasoning or different types of logical reasoning, would strengthen the paper's claims and make the results more broadly applicable.

2. Lack of Surprising Results: The finding that reasoning abilities improve with memorization is not particularly novel or surprising. While the paper conducts detailed analyses, it does not present clear guidance on how to leverage this insight for practical improvements. Standard fine-tuning with Chain-of-Thought (CoT) prompting, which the paper highlights, is already a well-known approach. The study would benefit from more innovative methods that build upon these findings.

3. Insufficient Baselines: The paper evaluates a narrow set of models and approaches. Including a broader range of baseline algorithms, particularly reinforcement learning (RL)-based models or other alternative reasoning frameworks, would provide more context for the performance of LLMs and help assess whether memorization is unique to certain models or training methods.

**Questions:**

1. Your findings suggest that reasoning improves with memorization, but the practical value of this insight is unclear. How do you envision this result being used in real-world applications or for model improvement?

2. Consider diversifying the tasks by including other logical or mathematical reasoning benchmarks (e.g., SAT solvers, arithmetic reasoning, or more complex logic puzzles). This would help demonstrate that your findings generalize beyond K&K and strengthen the paper’s conclusions.

3. Adding comparisons with other learning paradigms (e.g., RLs, decision transformers, or symbolic models) could broaden the understanding of how different models handle reasoning and memorization, especially in dynamic or less static environments than the K&K puzzles.

---

> ### Author Response · Authors · 2024-11-24
> **Response to Reviewer 7GXF (Part 1)**
>
> Thank you for your valuable comments! We address your questions and comments below.
>
> > Q1: Limited Task Scope: The paper focuses solely on logical reasoning, particularly the K&K puzzles. While this allows for deep analysis, it limits the generalizability of the conclusions. Experiments on other reasoning domains, such as mathematical reasoning or different types of logical reasoning, would strengthen the paper's claims and make the results more broadly applicable.
>
> Thanks for the valuable suggestion.  We clarify that many existing reasoning datasets, such as those for mathematical reasoning or other logical reasoning tasks, **do not easily allow for automatic local perturbations (especially for math-level perturbations) and new solution generation**, which is an important part of our memorization study. This limitation motivated us to propose a dataset specifically designed to dynamically generate puzzles & solutions & synthetic CoTs and apply configurable perturbations with controllable perturbing components and difficulty levels.
>
> Furthermore, we highlight that  **K&K puzzles are representative of SAT problems**, which are NP-complete. This classification implies that they capture the complexity inherent in a wide range of natural reasoning tasks [1]. Additionally, some existing logical reasoning benchmarks including ZebraLogic [2] and Einstein’s Puzzle [3] are Constraint Satisfaction Problem (CSP), which involves assigning values to variables under certain constraints. Any CSP can be reduced to an SAT problem by converting the variables and constraints into propositional variables and formulas [4].   This connection shows the relevance of our focus on K&K puzzles as a foundational basis for studying logical reasoning behaviors in LLMs.
>
> Reference:
> - [1] Armin Biere, Marijn Heule, Hans van Maaren, and Toby Walsh (Eds.), "Handbook of Satisfiability," IOS Press, 2009.
> - [2] Stuart Russell and Peter Norvig, "Artificial Intelligence: A Modern Approach," 3rd Edition, Prentice Hall, 2010.
> - [3] Bill Yuchen Lin, Ronan Le Bras, and Yejin Choi. ZebraLogic: benchmarking the logical reasoning ability of language models, 2024. URL https://hf.co/spaces/allenai/ZebraLogic.
> - [4] Nouha Dziri, Ximing Lu, Melanie Sclar, Xiang Lorraine Li, Liwei Jian, Bill Yuchen Lin, Peter West, Chandra Bhagavatula, Ronan Le Bras, Jena D. Hwang, Soumya Sanyal, Sean Welleck, Xiang Ren, Allyson Ettinger, Zaïd Harchaoui, and Yejin Choi. Faith and fate: Limits of transformers on compositionality. NeurIPS, 2024.

---

> ### Author Response · Authors · 2024-11-24
> **Response to Reviewer 7GXF (Part 2)**
>
> > Q2: Lack of Surprising Results: The finding that reasoning abilities improve with memorization is not particularly novel or surprising. While the paper conducts detailed analyses, it does not present clear guidance on how to leverage this insight for practical improvements….  How do you envision this result being used in real-world applications or for model improvement?
>
> We thank the reviewer for the feedback. While we acknowledge that the relationship between memorization and generalization has been explored (e.g., mostly in classification tasks), our study makes the following novel contributions specific to reasoning tasks for LLMs and provide insights for practical model improvements:
> - **Relevance to Dataset Contamination in Benchmarking**. Contamination of training data with benchmark test sets is a pervasive issue in LLMs evaluation, especially for popular reasoning datasets. This often leads to inflated performance metrics that obscure a model's true reasoning capabilities. Our study is motivated by the need to address this issue and systematically analyze how contamination, when controlled and quantified, influences reasoning and problem-solving abilities. The contamination-controlled design of our benchmark enables rigorous evaluation.
> - **Transferability Analysis**: Under our controlled setup, our findings demonstrate that, in reasoning tasks, memorization of easier examples—stemming from fine-tuning—can significantly improve a model's ability to solve harder tasks. Similarly, finetuning on hard tasks can help solve easier tasks. This insight provides practical guidance for designing training strategies: fine-tuning simpler logical reasoning tasks can serve as an effective method for preparing models for more complex tasks.
> - **Leveraging Direct Fine-Tuning in Absence of CoT**. We show that direct fine-tuning without CoT can help models acquire general problem-solving skills from question-answer pairs. This is particularly useful for tasks where generating CoT annotations is resource-intensive or infeasible (e.g., due to the lack of human expertise to generate step-by-step reasoning). The results highlight a novel avenue for efficiently training models in real-world scenarios where detailed CoT data is unavailable.
> - **Probing model internals**: Our probing analysis identifies the transformer blocks most relevant for learning K&K skills and shows that solving harder K&K tasks requires more computation, with task-relevant information shifting to later blocks. This potentially provides insights for model developers: fine-tuning specific blocks may optimize performance for complex reasoning tasks.
> - **Robustness to Low-Quality Data**:  Models fine-tuned on wrong answers or wrong CoT steps remain robust, indicating that high-quality data is not strictly necessary for fine-tuning in certain scenarios. This has direct implications for real-world applications where perfect data quality cannot be guaranteed, enabling broader applicability of fine-tuning methods.
>
>
> We hope these clarifications address concerns about the practical relevance of our results. We would be happy to discuss further if reviewers have additional feedback.

---

> ### Author Response · Authors · 2024-11-24
> **Response to Reviewer 7GXF (Part 3)**
>
> > Q3: Standard fine-tuning with Chain-of-Thought (CoT) prompting, which the paper highlights, is already a well-known approach. The study would benefit from more innovative methods that build upon these findings.
>
> Thanks for the feedback. In fact, our study primarily emphasizes analyzing **Direct FT** rather than CoT FT, as discussed in L299-309. Direct FT, which trains models using only question-answer pairs without detailed reasoning steps, is intuitively more challenging for LLMs, as it requires the model to infer reasoning procedures independently.
> Surprisingly, as shown in Fig. 5, we did not observe Direct FTed GPT-4-mini models exhibiting significantly higher memorization scores than CoT FTed ones.  The results in Sec. 4.1 show that models can learn to reason through K&K puzzles effectively even from question-answer pairs alone. Our findings highlight that Direct FT can be a viable approach for developing reasoning capabilities for logical reasoning problems, especially in cases where CoT data is unavailable or expensive to produce.
>
> To further explore what the model learns through Direct FT, we conducted *innovative analyses* including *Probing Analysis*(Sec. 4.3) and *Ablation Study with Incorrect Data Fine-Tuning* (Sec. 4.3).
>
>
> For **CoT FT**, we also performed novel analysis with low-quality CoT data fine-tuning in Section 5. We introduce incorrect CoT annotations including shuffled reasoning steps or one wrong step. One wrong CoT step mimics carefulness annotators that make small mistake. Interestingly, we found that models could still generalize and learn to reason effectively, demonstrating robustness to low-quality CoT annotations.
>
> Additionally, in Section 6, we propose **puzzle-based and model-based indicators** to distinguish samples solved via reasoning versus memorization. These indicators provide new insights into these two capabilities in LLMs.
>
> Lastly, our *contributions extend beyond fine-tuning*: we developed an innovative K&K **data generation framework** that supports creating new puzzles and systematically perturbing existing puzzles at various difficulty levels and computing new solutions. This framework is not only essential for our study but also serves as a resource for advancing future research on logical reasoning and memorization in LLMs.
>
>
>
> > Q4: Insufficient Baselines: The paper evaluates a narrow set of models and approaches. Including a broader range of baseline algorithms, particularly reinforcement learning (RL)-based models or other alternative reasoning frameworks, would provide more context for the performance of LLMs and help assess whether memorization is unique to certain models or training methods… Adding comparisons with other learning paradigms (e.g., RLs, decision transformers, or symbolic models) could broaden the understanding of how different models handle reasoning and memorization, especially in dynamic or less static environments than the K&K puzzles.
>
> Thank you for the thoughtful suggestion. Could the reviewer kindly suggest specific RL-based models or alternative reasoning frameworks that would be particularly relevant for comparison? We would be happy to include those in our revised manuscript.

---

> > ### Comment · Reviewer_7GXF · 2024-11-26
> >
> > Thanks for your reply!
> >
> > Thank you for highlighting the main contributions of the paper and the significance of your tasks. I agree that SAT questions are a common category in reasoning. However, the generalization of the paper's conclusions (such as "Leveraging Direct Fine-Tuning in the Absence of CoT," "Robustness to Low-Quality Data," etc.) should be validated through other tasks; otherwise, the universality of the conclusions may be questioned. I appreciated the thorough analysis the authors have conducted in the paper. My main concern lies with the generalization of the paper's conclusions. I will increase my score accordingly.

---

> > > ### Author Response · Authors · 2024-11-28
> > > **Response to Reviewer 7GXF**
> > >
> > > Thank you again for your valuable feedback and positive rating. Your support is vital to us.
> > >
> > > Indeed, while our dataset is a representative SAT logical reasoning task, the conclusions about the effectiveness of direct FT and robustness of low-quality data have not been verified for other types of reasoning tasks (e.g., mathematical reasoning). We will make it clear in our revision. Extending our work to include broader reasoning tasks is an exciting direction for future research, and we are grateful for your suggestion.

---

### Official Review · Reviewer_XCvy · 2024-11-04

**Soundness:** 1
**Presentation:** 2
**Contribution:** 1
**Rating:** 5
**Confidence:** 4

**Summary:**

The paper proposes a memorisation metric (LiMem) to quantify the extent of memorisation vs reasoning exhibited by language models when solving logical reasoning tasks. The metric is based on measuring inconsistency when a model solves a locally perturbed version of a logical reasoning puzzle. The paper also proposes logical reasoning benchmark based on the Knights and Knaves puzzles, which enables the memorisation study and could be useful for future research on logical reasoning in language models.

## Main Experiments
- They test  on set of eight open and closed source models. They compare the scores with and without perturbations across various parameters.
- They also run experiments by fine-tuning models on variations of the knight and knave puzzles, They claim that fine tuning leads to better memorisation and reasoning on various modes of difficulties.

**Strengths:**

- Overall, I believe this paper is well written and easy to understand.
- The authors have explained, their assumptions and experimental setup clearly.
- Main figure explains most of experimental gist at glance.

**Weaknesses:**

## Limitations
- **State space with perturbations**: As the problem space is limited in terms of number of people, depth and width,  Only maximum of 8, 2, 2 respectively. These limited dimensions make it relatively easy for models to interpolate the entire problem space with perturbations, potentially inflating perceived generalisation.
- **Limited Evaluation** : The authors analyze only 8 models, yet they refer to it as a benchmark, which limits its claim to be benchmark. A more comprehensive evaluation across diverse models is necessary, particularly with a focus on distinguishing performance in terms of memorization versus reasoning—an analysis notably missing in the paper.
- Boilerplate memorization issues have been raised by other studies (e.g., Sai et al. [1]) that address similar patterns of template memorisation. [1] work reaffirms that slight variations in variable names do not disrupt memoization. It can also in turn explains strong perfomance of fine-tuned models, with small number of finetuning samples.
- Recent analyses, like that by Saparov and He [2], have highlights LMs' reasoning abilities in Chain-of-Thought (CoT) contexts. 1-shot performant doesn't seems to reduce performance for various perturbations as opposed to 0-shot.

Please cite missing relavant work that explore memorization and formal reasoning.
[1] P. U. Sai et al., “Recite, Reconstruct, Recollect: Memorization in LMs as a Multifaceted Phenomenon,” arXiv.org, 2024. https://arxiv.org/abs/2406.17746.
[2] A. Saparov and H. He, “Language Models Are Greedy Reasoners: A Systematic Formal Analysis of Chain-of-Thought,” arXiv:2210.01240 [cs],
[3] L. Pan et al "LOGIC-LM: Empowering Large Language Models with Symbolic Solvers for Faithful Logical Reasoning" https://arxiv.org/pdf/2305.12295

‌

**Questions:**

I have highlighted fundamental limitations in previous section. I have a few questions regarding implementation and experimental details:

- The paper seems to omits key fine-tuning details, such as whether standard SFT or LoRA was used, any quantization techniques, configurations or other factors that could significantly impact the observed memorisation and reasoning behaviours.

- Although the study mentions language-level and mathematical-level perturbations, it does not examine the effect of progressively stacking these perturbations. A more thorough exploration here could offer insights into model robustness and reasoning depth.

-  It seems logical to consider test-time inference techniques like self-refine, majority voting or self-consistency, which could enhance reasoning results. Such experiments would help demonstrate robustness.

---

> ### Author Response · Authors · 2024-11-24
> **Response to Reviewer XCvy (Part 1)**
>
> Thank you for your valuable feedback! We address your questions and comments below.
>
> > Q1: State space with perturbations: As the problem space is limited in terms of number of people, depth and width, Only maximum of 8, 2, 2 respectively. These limited dimensions make it relatively easy for models to interpolate the entire problem space with perturbations, potentially inflating perceived generalisation.
>
> Thank you for the comment. We would like to clarify that the problem space is, in fact, extremely large (~10^24). Below, we calculate the number of possible combinations for the n-ppl=8, depth=2, width=2 configuration:
>
> - **Leaf Nodes**: A leaf node can represent the statement “X is lying” or “X is telling the truth.” With 8 individuals, this results in 16 possible combinations for a single leaf node.
> - **Branching Nodes**: Ignoring the ‘not’ operator for simplicity and considering only the logical operators ‘and,’ ‘or,’ ‘imply,’ and ‘equivalence,’ each branching node has a width of 2 (as specified). With a depth of 2, the children of each branching node are always leaf nodes. Thus, a branching node with width-2 and depth-2 has: 4 operators × 16^2 combinations of two child leaf nodes = 1024 possible combinations.
> - **Total Problem Space**: With 8 individuals, each making a statement, the total problem space becomes: 1024^8 ≈ 10^24 combinations.
>
> Even though a large portion of these combinations may not yield valid solutions, the resulting problem space remains enormous.
>
>
> > Q2 Limited Evaluation : The authors analyze only 8 models, yet they refer to it as a benchmark, which limits its claim to be benchmark. A more comprehensive evaluation across diverse models is necessary, particularly with a focus on distinguishing performance in terms of memorization versus reasoning—an analysis notably missing in the paper.
>
> Thanks for the comment. We mainly focus on LLMs that are shown to perform competitively on common reasoning benchmarks.  Following the comment, we additionally evaluate 6 models: Gemini-1.5-Flash-002, Gemini-1.5-Pro-002, Gemma-2-9b, Llama-3.1-8B-Instruct, Qwen2-Math-7B-Instruct,  Qwen2.5-Math-7B-Instruct.
>
> We report results for a total of 14 models in Figure 3 of the revised manuscript. Notably, while Qwen2-Math and Llama-3.1-8B-Instruct are competitive among the open-source models, they exhibit large performance inconsistency under perturbation (e.g., 0.3 memorization score in 2-ppl K&K task under perturbed leaf setting), highlighting the limitations in their robust reasoning abilities and potential memorization.
>
>
> > Q3. Boilerplate memorization issues have been raised by other studies (e.g., Sai et al. [1]) that address similar patterns of template memorisation. [1] work reaffirms that slight variations in variable names do not disrupt memoization. It can also in turn explains strong performance of fine-tuned models, with small number of finetuning samples.
>
> Thanks for the comment. While prior studies have shown that slight variations in variable names or surface-level language do not significantly disrupt memorization, we focus more specifically on **math-level small, local perturbations** (i.e., statement and leaf perturbations). As shown in Fig 5 and Fig 6, math-level perturbations lead to significantly larger memorization scores (i.e., performance drops) compared to language-level perturbations (e.g., changing people's names, role pair names, reordering). These math-level perturbations involve slightly altering the underlying math structures, rather than just modifying variable names or linguistic expressions. This is also more challenging than language-level perturbation because the perturbed problem needs to be solved to find the **new ground-truth answer** for evaluation, which is enabled by our dynamic K&K generation pipeline. Our study goes beyond template memorization by challenging the models' understanding of the underlying mathematical principle.
>
>
>
> > Q4. Recent analyses, like that by Saparov and He [2], have highlights LMs' reasoning abilities in Chain-of-Thought (CoT) contexts. 1-shot performant doesn't seems to reduce performance for various perturbations as opposed to 0-shot.
>
> We thank the reviewers for highlighting the relevant analyses. We did evaluate 1-shot prompting, CoT prompting and 1-shot CoT prompting  (deferred to appendix due to space limit).
> In Appendix Figure 17, we report the performance of models under these 1-shot/CoT prompting and direct prompting. Our results show that models with high accuracy, such as Phi-3-medium on 3-ppl puzzles (accuracy = 54%), also exhibit a substantial performance inconsistency under perturbations, as indicated by their memorization scores (LiMem = 37%).
> This suggests that even with 1-shot and CoT prompting, models are sensitive to perturbations in K&K puzzles, highlighting the K&K dataset's value for assessing robustness and reasoning in diverse scenarios.

---

> > ### Author Response · Authors · 2024-11-24
> > **Response to Reviewer XCvy (Part 2)**
> >
> > > Q5 Please cite missing relavant work that explore memorization and formal reasoning. [1] P. U. Sai et al., “Recite, Reconstruct, Recollect: Memorization in LMs as a Multifaceted Phenomenon,” arXiv.org, 2024. https://arxiv.org/abs/2406.17746. [2] A. Saparov and H. He, “Language Models Are Greedy Reasoners: A Systematic Formal Analysis of Chain-of-Thought,” arXiv:2210.01240 [cs], [3] L. Pan et al "LOGIC-LM: Empowering Large Language Models with Symbolic Solvers for Faithful Logical Reasoning" https://arxiv.org/pdf/2305.12295
> >
> > Thanks for the comment. In our revision, we discussed these relevant works in Section 7 related work and also in Appendix B extended related work (due to space limit).
> >
> >
> > > Q6 The paper seems to omits key fine-tuning details, such as whether standard SFT or LoRA was used, any quantization techniques, configurations or other factors that could significantly impact the observed memorisation and reasoning behaviours.
> >
> > Thank you for the valuable suggestion. We included fine-tuning the details in Appendix D.2.2 and have now added further clarification in the revised PDF.
> >
> > For Llama fine-tuning, we used LoRA fine-tuning with standard hyperparameters: a batch size of 4,  gradient accumulation steps of 8, and 5e-5 learning rate.  The LoRA configuration was set as follows: rank $r = 32$, scaling factor $\alpha = 32$, and dropout rate $0.05$. No quantization techniques were used.
> > Please let us know if there are any other details or clarifications needed.
> >
> > > Q7  Although the study mentions language-level and mathematical-level perturbations, it does not examine the effect of progressively stacking these perturbations. A more thorough exploration here could offer insights into model robustness and reasoning depth.
> >
> > Thank you for the insightful comment. We note that the memorization score under mathematical perturbations has been high, as demonstrated in Figures 3, 5, and 6, supporting our hypothesis that the models could rely on memorization to solve the puzzles instead of robust reasoning.
> >
> > Following your suggestion, we explored the combination of math-level  perturbations with language-level perturbations in Appendix E.1 Figure 20. Memorization scores of Directly Fine-Tuned Llama3-8B under various math-level (statement, leaf) and language-level (name, reorder) perturbations. Combining math-level and language-level perturbations progressively can result in higher memorization scores (e.g., leaf + reorder), especially compared to applying language-level perturbations alone.
> >
> > > Q8 It seems logical to consider test-time inference techniques like self-refine, majority voting or self-consistency, which could enhance reasoning results. Such experiments would help demonstrate robustness.
> >
> >
> > Thank you for the suggestion. While test-time inference techniques are orthogonal to our study of memorization, which is introduced during training, we followed this recommendation to evaluate self-consistency. Specifically, we tested self-consistency using gpt-4o-mini, a strong model that outperforms open-source models on K&K tasks (as shown in Table 3). For the default direct prompting method, we used greedy sampling with a temperature of 0. For self-consistency, we applied temperature-based sampling with a temperature of 0.7, generating 40 samples to compute the results.
> >
> > The results in below table show that while self-consistency improves accuracy on the simpler 2-ppl task, it provides only marginal improvement on the 3-ppl task and fails entirely on the more challenging 8-ppl task. This highlights a fundamental limitation of the model in solving complex problems. Additionally, we report the memorization scores for the 2-ppl and 3-ppl tasks, which demonstrate that self-consistency reduces memorization scores, likely due to its majority voting mechanism that leads to robust reasoning results.
> >
> > | Method | Test Accuracy ↑ | | | Memorization Score ↓ | |
> > | --- | --- | --- | --- | --- | --- |
> > | | 2-ppl | 3-ppl | 8-ppl | 2-ppl | 3-ppl |
> > | Direct Prompting | 0.63 | 0.42 | 0.01 | 0.24 | 0.26 |
> > | Direct Prompting + Self-consistency | 0.74 | 0.43 | 0.02 | 0.20 | 0.22 |

---

> > > ### Comment · Reviewer_XCvy · 2024-12-01
> > > **Thanks for explanation**
> > >
> > > Thank you for your clarifications. As highlighted by other reviewers as well, I see some issues with the current work:
> > >
> > > - The findings of the paper (finetuning, reasoning steps), while reasonable and intuitive, do not seem to provide new insights into model capabilities or propose recipes that could be effectively transferred to other problems.
> > >
> > > - As noted by the authors, most models appear robust with respect to LiMem scores in the 1-shot regime. If models were purely memorizing, this would not be the case.
> > >
> > > - I remain unconvinced about the fundamentals and necessity of the LiMem scores. It is unclear whether this metric adequately distinguishes between memorization and generalization.
> > >
> > >
> > > Given these concerns, I believe the contributions of the paper need further refinement to strengthen their impact. Likewise, I am increasing the score to represent my beliefs.

---

> ### Author Response · Authors · 2024-12-02
> **Response to Reviewer XCvy**
>
> Thank you for your comments and for raising the score! Please find our response to your questions below:
>
> 1. Thanks for the feedback and we would like to emphasize that the findings of our work provide several practical insights for model development. Specifically, our contributions are motivated by the dataset contamination issues in benchmarking, and we show that under high memorization, the model exhibits transferability for easier/harder logical reasoning tasks. We also show the effectiveness of direct fine-tuning in the absence of Chain-of-Thought training data, provide interpretability analysis through probing model internals to better understand decision-making processes, and analyze model's robustness under low-quality or noisy data.
>
> 2. We would like to clarify a potential misunderstanding. In our original response, we stated that  "even with 1-shot and CoT prompting, models **are sensitive** to perturbations in K&K puzzles."  As shown in Figure 17, for instance, Phi-3-medium exhibits a high memorization score of 0.37 under lead perturbation on the 2-person task, which corresponds to a significant performance drop.
>
>
> 3. We clarify that high LiMem scores effectively capture high memorization cases by reflecting two important characteristics observed in human behavior: high accuracy on previously seen problems and low consistency when problems are slightly perturbed. This underscores the fundamental importance and necessity of LiMem scores for capturing memorization in LLM reasoning tasks.
>
> Please feel free to let us know if there are additional questions or points requiring clarification.   Thank you for your time and feedback.

---

### Official Review · Reviewer_1uBu · 2024-11-04

**Soundness:** 2
**Presentation:** 3
**Contribution:** 2
**Rating:** 5
**Confidence:** 4

**Summary:**

This study examines how LLMs balance memorization and reasoning in solving logical reasoning tasks, using a benchmark based on Knights and Knaves (K&K) puzzles. Findings reveal that while fine-tuning enhances LLMs' generalization abilities, it also leads to heavy memorization. The models perform well on familiar tasks but struggle with slight variations, suggesting a nuanced interplay between memorization and genuine reasoning skills in LLMs.

**Strengths:**

- The paper attempts to reveal the nuanced relationship between memorization and reasoning, contributing to a deeper understanding of LLM capabilities and limitations.

- Perturbation tests offer methods to assess LLMs' reasoning abilities independently of memorization.

**Weaknesses:**

- The definition of "memorization" is vague. Is that the opposite of "generalization"? Why do we need to create a new term (and even a new metric) compared to the traditional term in machine learning research?

- Following the question above, I think the definition of memorization score is too arbitrary and may be misleading. What's the meaning of multiplication of accuracy and CR? For example, (ACC=0.2, CR=0.2) and (ACC=0.8 and CR=0.8) will produce the same score. Do these two results have the same level of memorization under your definition? It's also very counter-intuitive that "off-the-shelf models" show signs of "memorization" when solving these puzzles, even though they are never trained on this it. The name and the motivation in the Introduction left an impression that it's a metric to reflect the generalization gap. However, this doesn't seem to be the case, since a model that has never been fitted on the dataset is also likely to have a memorization score > 0.

- Though the authors attempt to "distinguish memorization from reasoning" with rich experiments, they are all based on vague and probably problematic definitions of "memorization", which makes the results not as insightful.

- A rich line of research on "grokking" [1, 2] might be very relevant to the research problem in this paper.

[1] Grokking: Generalization Beyond Overfitting on Small Algorithmic Datasets

[2] Grokked Transformers are Implicit Reasoners: A Mechanistic Journey to the Edge of Generalization

**Questions:**

N/A

---

> ### Author Response · Authors · 2024-11-24
> **Response to Reviewer 1uBu (Part 1)**
>
> Thank you for your thoughtful feedback! We address your questions and comments below.
>
> > Q1. The definition of "memorization" is vague. Is that the opposite of "generalization"? Why do we need to create a new term (and even a new metric) compared to the traditional term in machine learning research?
>
> Memorization and generalization are not opposites, as they target different aspects of model performance. Memorization is typically assessed on **training data** (in our work, using *small, locally perturbed* training samples), while generalization is evaluated on novel **test data**.  As shown in Section 4, a model that memorizes training data can still generalize well to both in-distribution and out-of-distribution test data.
>
> Modern foundation models, such as LLMs, differ from traditional ML. Unlike typical ML settings (e.g., classification) where the training and test sets can be explicitly separated, LLMs are trained on internet-scale data where the exact training data is often unknown. This makes it challenging to construct clean test sets to measure generalization accurately. Due to potential overlap between training and test data, concerns about benchmark saturation arise, where models may have memorized the same/similar problems during training and thus achieve high accuracy on testing benchmarks.  In this context, a notion of "memorization" is especially important to underscore the difficulties in accurately assessing model reasoning capabilities.
>
> To address this challenge, we introduce a memorization metric designed to detect signs of memorization under various local perturbations and to quantify this phenomenon.
>
> > Q2.   I think the definition of memorization score is too arbitrary and may be misleading.  What's the meaning of multiplication of accuracy and CR? For example, (ACC=0.2, CR=0.2) and (ACC=0.8 and CR=0.8) will produce the same score. Do these two results have the same level of memorization under your definition?
>
> Under the LiMem definition, these two results indeed reflect the **same low level** of memorization, as both produce the same score (LiMem = 0.2 × (1-0.2) = 0.8 × (1-0.8)=0.16). However, the implications of these cases are different, as clarified in lines 144-147: Low LiMem score could indicate either solving by reasoning (ACC=0.8 and CR=0.8) or not solving (e.g., random guessing) (ACC=0.2, CR=0.2).
>
> A low LiMem score can only indicate “solving by reasoning” if we separately check that accuracy is high, as seen in the (ACC=0.8, CR=0.8) case.
>
> > Q3: It's also very counter-intuitive that "off-the-shelf models" show signs of "memorization" when solving these puzzles, even though they are never trained on this it. The name and the motivation in the Introduction left an impression that it's a metric to reflect the generalization gap. However, this doesn't seem to be the case, since a model that has never been fitted on the dataset is also likely to have a memorization score > 0.
>
> We appreciate the comment and would like to clarify the potential sources of memorization observed in off-the-shelf models. While the puzzles we generate are randomly constructed (e.g., random language expression and math structures), Knights and Knaves (K&K) is a well-known classical puzzle type. Existing instances of such puzzles, along with related materials, are available online [1,2] and may have been included in the training data for off-the-shelf models. Since the exact training sets for these models are not disclosed, it is plausible that some degree of exposure to K&K-like problems or related text has occurred during pretraining.
>
> In addition, we conducted a search using existing open-source datasets. Specifically, we utilized the [WIMBD tool](https://wimbd.apps.allenai.org/) to analyze the occurrence of popular names (“Alice”, “Bob”) combined with different roles (e.g., "knight," "knave," etc.) in these datasets. The results, summarized in the table below, suggest that K&K types of materials could be included in pretraining data, indicating a potential source for memorization in off-the-shelf models.
>
> | Statement | Dolma | The PILE | C4 | Oscar | OpenWebText |
> |---|---|---|---|---|---|
> | "Alice is a knave" | 13 | 6 | 2 | 1 | 0 |
> | "Alice is a knight" | 23 | 8 | 6 | 1 | 0 |
> | "Bob is a knave" | 11 | 8 | 0 | 1 | 0 |
> | "Bob is a knight" | 53 | 9 | 22 | 5 | 0 |
> | "Charlie is a knave" | 3 | 0 | 0 | 0 | 0 |
> | "Charlie is a knight" | 10 | 1 | 2 | 0 | 0 |
>
> References
> - [1] https://philosophy.hku.hk/think/logic/knights.php
>  - [2] https://dmackinnon1.github.io/knaves/

---

> ### Author Response · Authors · 2024-11-24
> **Response to Reviewer 1uBu (Part 2)**
>
> > Q4: "distinguish memorization from reasoning" are all based on vague and probably problematic definitions of "memorization", which makes the results not as insightful.
>
> Thanks for the comment. We clarify that the definition of memorization is valid in response to Q1-Q2.
>
> Additionally, for "distinguish memorization from reasoning",  we note that our analysis at this stage is based on a *sample-level binary score*— whether a sample is consistently solved under perturbation or not, as mentioned in  in L459-462.  Specifically, we consider *1-point dataset*, and only study **samples for which the model predicts correctly (ACC=1), to rule out the possibility of “not solving”**, and thus provide direct insights into “solving by reasoning” v.s . “solving by reasoning ” for each sample.
>
> > Q5: A rich line of research on "grokking" [1, 2] might be very relevant to the research problem in this paper.
>
> We thank the reviewer for pointing out the related work. We acknowledge that the grokking phenomenon are very interesting, which is first identified by [1] on a small algorithmic dataset where validation accuracy suddenly improves from random chance to near-perfect generalization long after severe overfitting.  Recently [2] observed grokking in the domain of complex knowledge-based tasks, showing that implicit reasoning over parametric knowledge emerges only after extensive overfitting.
>
> In this work, we observe a related phenomenon but through the lens of memorization. Through novel (math &language-level) perturbation tests, transferability, and probing analyses, we verify that LLM reasoning skills emerge alongside memorization.  Furthermore, our investigation focuses on logical reasoning, offering new insights into how LLMs acquire logical reasoning skills.
>
> We added the above discussion in the extended related work in Appendix B (due to space limit).

---

> ### Comment · Reviewer_1uBu · 2024-11-25
>
> > Under the LiMem definition, these two results indeed reflect the same low level of memorization, as both produce the same score (LiMem = 0.2 × (1-0.2) = 0.8 × (1-0.8)=0.16). However, the implications of these cases are different, as clarified in lines 144-147: Low LiMem score could indicate either solving by reasoning (ACC=0.8 and CR=0.8) or not solving (e.g., random guessing) (ACC=0.2, CR=0.2).
>
> Why do you believe they should have the same level of memorization? This explanation only states that these two cases are the same under your definition, but does not justify why these two cases **should** be the same from the principle. The definition of this metric is still rather arbitrary to me.
>
> > ...the potential sources of memorization observed in off-the-shelf models...
>
> I don't think data leakage can explain this problem. Let's just imagine we can remove all K&K puzzles in the pertaining data. Do you think the LLM trained this way will have a LiMem score of 0?
>
> I would assume it can still learn certain "true reasoning abilities" from the pertaining, so the accuracy will not be 0. There would be some randomness in the prediction, so it may not always consistently solve all cases, which makes CR < 1. Therefore, the LiMem would still be > 0.
>
> If the logic above is correct, I think that means the definition of your metric is not really reflective of "memorization".
>
> > ... only study samples for which the model predicts correctly (ACC=1), to rule out the possibility of “not solving”...
>
> I think that makes much more sense than LiMem which mixes accuracy and CR... However, the main body of this paper is still based on the LiMem, and it's necessary to provide a convincing explanation of this metric.

---

> > ### Author Response · Authors · 2024-11-25
> >
> > > Why do you believe they should have the same level of memorization? This explanation only states that these two cases are the same under your definition, but does not justify why these two cases **should** be the same from the principle. The definition of this metric is still rather arbitrary to me.
> >
> > We would like to clarify two things: (1) in our response, we were acknowledging that the two cases indeed leads to the same LiMem score, therefore the same level of memorization according to our metric. But we were not claiming that these two cases are the same from the principle. (2) we emphasized the "**low** level of memorization" to clarify that a **low** LiMem score is an indicator of **a lack of memorization**. In the example suggested by the reviewer, a low (0.16) LiMem could either be due to low accuracy or high robustness.
> >
> > The primary purpose of the LiMem metric is to capture **cases of memorization (high LiMem score)** , inspired by human behavior. We acknowledge that a low LiMem score can have multiple explanations and there is no formal principle to properly quantify which of the two cases should be interpreted as "more level of memorization": (ACC=0.2, CR=0.2) (ACC=0.8 and CR=0.8). As a result, while we use high LiMem to capture memorization (Sec 3), when studying the relation between memorization and reasoning (Sec 4, 5), we complement the low LiMem measurement with additional evidences such as cross-difficult-level generalization and hidden state probing. In summary, memorization is a vague term which motivates us to quantify it in the context of reasoning, and we hope this clarification highlights both the intent and the limitations of the metric. We will include this discussion in the paper.
> >
> >
> > > I don't think data leakage can explain this problem. Let's just imagine we can remove all K&K puzzles in the pertaining data. Do you think the LLM trained this way will have a LiMem score of 0?
> > I would assume it can still learn certain "true reasoning abilities" from the pertaining, so the accuracy will not be 0. There would be some randomness in the prediction, so it may not always consistently solve all cases, which makes CR < 1. Therefore, the LiMem would still be > 0.
> > > If the logic above is correct, I think that means the definition of your metric is not really reflective of "memorization".
> >
> > We agree that several factors, such as token bias, order bias, influence the consistency ratio (CR) observed in LLM behavior. These factors may not necessarily reflect  memorization.  As a result, even in the absence of K&K data leakage, a model may achieve non-trivial accuracy while exhibiting some inconsistency, leading to a non-zero LiMem score.
> >
> > However, we believe that a **high LiMem score** remains an effective indicator of memorization. As illustrated in Figure 5, achieving a memorization score of 0.5, with a training accuracy of 0.75 after fine-tuning (as shown in Figure 4), requires the model to have an inconsistency ratio of at least 0.67. This high level of inconsistency strongly suggests that the model is memorizing and failing to maintain performance on most originally solved problems under perturbations.
> >
> > We acknowledge the reviewer’s concerns. Setting a threshold for accuracy to only consider high LiMem score cases may help to more clearly interpret the LiMem score. We will make it clear in revision.

---

### Author Response · Authors · 2024-11-24
**Revision Summary**

We sincerely thank all reviewers for their constructive feedback and suggestions, which are very helpful to us. We are encouraged that the reviewers found our work (1) propose novel memorization metric (Reviewer 7GXF), (2) novel data generation and perturbation methods (Reviewer 1uBu, Reviewer ofks),  (3) and provide deeper understanding of LLM reasoning (Reviewer 1uBu, Reviewer 7GXF), (4) the results are well-design and extensive  (Reviewer 7GXF, Reviewer ofks) and (5) the writing and presentation are clear (Reviewer XCvy).

Following the reviewers’ suggestions, we added more experiments/discussions, and we addressed the questions in the response to each reviewer. Below is a summary of our new experimental results and discussions in the revised PDF:

- **Figure 3, More LLMs**: we evaluate 6 additional models that have competitive reasoning capability: Gemini-1.5-Flash-002, Gemini-1.5-Pro-002, Gemma-2-9b, Llama-3.1-8B-Instruct, Qwen2-Math-7B-Instruct, Qwen2.5-Math-7B-Instruct.  (Reviewer XCvy)
- **Appendix E.1 Figure 20, Combined perturbations**: we study the effect of progressively stacking language-level and mathematical-level perturbations. (Reviewer XCvy)
- **Appendix  E.1 Table 3, Self-consistency**: we report the accuracy and memorization score when using test-time inference technique self-consistency.(Reviewer XCvy)
- **Appendix  E.1 Figure 19, Consistency ratio**: we report the consistency ratio on the train/test set. (Reviewer ofks)
- **Section 7 & Appendix B Related Work**: we discuss more related work suggested by the reviewers. (Reviewer 1uBu, XCvy)
- **Figure 1 and Section 1 Introduction**: we add more explanation regarding the memorization score.  (Reviewer ofks)
- **Section 2.2**: we highlight that the principle underlying K&K is the Boolean satisfiability (SAT) problem, which is essential to study for evaluating LLM logical reasoning capability. (Reviewer 7GXF, ofks)

Please also let us know if there are other questions, and we look forward to the discussion with the reviewers to further improve our paper. Thank you!

---

### Meta-Review · Area_Chair_aGuu · 2024-12-21

**Metareview:**

Whether LLMs learn to reason or its perceived reasoning power is rooted in its ability to memorize huge space of potential answers is an important question to understand the mechanism of LLMs. Because of its importance, lots of prior work exists. This paper proposes a "perturbation-based" method to quantify LLMs' memorization ability. Memorization definitely plays a role in reasoning. If a model knows nothing, it definitely cannot reason. Because of this, a method to quantify such that we can understand to what extend the memorization plays the dominant role is an important attempt. However, main concerns are two-folded. Firstly, the novelty of this metric, and to what extent we can trust this metric. Perturbation-base methods are not particularly novel conceptually, and a comprehensive inclusion of reference is needed. Furthermore, due to how the state space is perturbed, the state space is not enough to fully convince reviewers the validity of this metric. Secondly, I would not particularly argue for whether the results/conclusion need to be surprising or not. But in some sense, we get a hand-wavy definition of "memorization" and "genuine" reasoning, which makes some discussion less concrete. I would encourage the authors to make a more clarified definition, make a more distinct case to separate the two, and include more results on a larger state space and models.

**Additional Comments On Reviewer Discussion:**

More than one reviewers raised the concern about the distinction between "memorization" and "genuine reasoning". The concern still remains after the rebuttal. The soundness of the experiments (e.g., concern about limited model evaluations) which were raised by all reviewers, were somewhat addressed by the authors, and 1 or 2 two reviewers raised the score accordingly. However, due to the limitation of the state space, reviewers still found the results not convincing enough.

---

### Decision · Program_Chairs · 2025-01-22

Reject